# Are First-Order Diffusion Samplers Really Slower?
# A Fast Forward-Value Approach

**Yuchen Jiao** [\* 1]  **Na Li** [\* 2]  **Changxiao Cai** [3]  **Gen Li** [1]

## Abstract

Higher-order ODE solvers have become a standard tool for accelerating diffusion probabilistic model (DPM) sampling, motivating the widespread view that first-order methods are inherently slower and that increasing discretization order is the primary path to faster generation. This paper challenges this belief and revisits acceleration from a complementary angle: beyond solver order, the placement of DPM evaluations along the reverse-time dynamics can substantially affect sampling accuracy in the low-neural function evaluation (NFE) regime. We propose a novel training-free, first-order sampler named Forward DPMSolver (F-DPMSolver), whose leading discretization error has the opposite sign to that of DDIM. Algorithmically, the method approximates the forward-value evaluation via a cheap one-step lookahead predictor. We provide theoretical guarantees showing that the resulting sampler provably approximates the ideal forward-value trajectory while retaining first-order convergence. Empirically, across standard image generation benchmarks, the proposed sampler consistently improves sample quality under the same NFE budget and can be competitive with, and sometimes outperform, state-of-the-art higher-order samplers. Overall, the results suggest that the placement of DPM evaluations provides an additional and largely independent design angle for accelerating diffusion sampling. Our code is available at https://github.com/Na-Li66/F-DPMSolver.

---

[\*]Equal contribution [1]Department of Statistics and Data Science, Chinese University of Hong Kong, Shatin, New Territories, Hong Kong [2]College of Information Science and Electronic Engineering, Zhejiang University, Hangzhou, China [3]Department of Industrial and Operations Engineering, University of Michigan, Ann Arbor, USA. Correspondence to: Gen Li <genli@cuhk.edu.hk>.

*Proceedings of the 43$^{rd}$ International Conference on Machine Learning*, Seoul, South Korea. PMLR 306, 2026. Copyright 2026 by the author(s).

## 1. Introduction

Diffusion probabilistic models (DPMs) have rapidly emerged as a leading paradigm in modern generative modeling, achieving state-of-the-art performance across a wide range applications in generative AI (Ho et al., 2020; Song et al., 2020b; Song & Ermon, 2019; Song et al., 2020a; Dhariwal & Nichol, 2021). Rooted in principles from non-equilibrium thermodynamics (Sohl-Dickstein et al., 2015), DPMs generate complex data by learning to reverse a gradual noise-injection process, transforming simple noise into structured, high-fidelity outputs. This transformation is achieved through a denoising process guided by pre-trained neural networks that approximate the score functions. These models have shown remarkable success in a wide range of generative tasks, including image synthesis (Rombach et al., 2022; Ramesh et al., 2022; Saharia et al., 2022), audio generation (Kong et al., 2021), video generation (Villegas et al., 2022), and molecular design (Hoogeboom et al., 2022), underscoring their versatility and impact. See e.g., Yang et al. (2023); Croitoru et al. (2023); Lai et al. (2025) for overviews of recent development.

Sampling from a pre-trained DPM typically proceeds by discretizing either the diffusion SDE (e.g., Ho et al. (2020); Nichol & Dhariwal (2021); Bao et al. (2022)) or the diffusion ODE (e.g., Song et al. (2020a); Lu et al. (2022a)) associated with the DPM. DPMs, parameterized by large neural networks, are commonly trained as *noise prediction models*: given a noisy sample and a time step, they are trained to predict the noise added to the original clean data at that time in the forward process. Since subtracting this noise estimate from the noisy sample yields an estimate of the clean data, equivalently, these models learn to predict the clean data. As a result, high-quality generation relies on how effectively we discretize an diffusion SDE/ODE and leverage the pre-trained DPMs.

Most existing diffusion-based samplers first choose a sequence of time steps for the iterations and then iterate backward in time. At each step, the pre-trained DPM is evaluated at the current iterate and time to produce an estimate of the clean data, which is then used to update the iterate. In particular, a first-order discretization of the diffusion ODE and SDE leads to the deterministic DDIM (Song et al., 2020a)

and the DDPM (Ho et al., 2020) samplers, respectively.

A major drawback of DDIM and DDPM is their slow convergence, often requiring tens to thousands of iterations to generate high-quality samples. Since each iteration involves a neural function evaluation (NFE) of the pre-trained DPM, sampling speed is often the main bottleneck in deployment. One can expect that the slow sampling convergence is largely due to the discretization error incurred by the sampling schemes. This has motivated a line of accelerated samplers (Lu et al., 2022a;b; Zhao et al., 2023) that use higher-order discretization schemes of the diffusion ODE to reduce discretization error and improve sampling speed.

The practical success of higher-order samplers has lead to a widespread belief: the order of a diffusion sampler (the order of the underlying discretization scheme) is the key to speed, and higher-order samplers are inherently faster than first-order ones. This belief has driven substantial effort towards more refined diffusion ODEs solvers.

In this work, we challenge this common belief by asking two fundamental questions:

1. *Is the order of a diffusion sampler truly the bottleneck limiting sampling speed?*

2. *If not, can a carefully designed first-order sampler reach comparable (or better) performance?*

### 1.1. Our contributions

Motivated by these questions, we revisit the conventional wisdom that acceleration in diffusion-based sampling necessarily requires increasing the discretization order.

We start from a simple but under-explored idea: a *forward-value* discretization of the diffusion ODE. Given the current iterate and time, the idealized forward-value update would evaluate the DPM at quantities corresponding the *next* time. We observe that such a scheme can perform effectively even in the extremely low-NFE regime, which motivates us to design a practical sampler that approximates the forward-value update. Specifically, we propose a *first-order* sampler named Forward DPMSolver (F-DPMSolver), which predicts these next-step quantities on the fly: at each iteration, we first generate a rough estimate using a vanilla sampler (e.g., DDIM), and then combine the current iterate with a DPM evaluation at this estimate to produce the next iterate. We then theoretically prove that the discretization error of our proposed sampler is of the same order as the deterministic DDIM, both scaling $\widetilde{\Theta}(1/M)$ where $M$ is the number of iterations.

We validate the effectiveness of the proposed sampling procedure through extensive experiments using pre-trained DPMs on standard image generation benchmarks, including the CIFAR-10 (Krizhevsky et al., 2009), ImageNet (Deng et al., 2009), FFHQ (Karras et al., 2019), and LSUN (Yu et al., 2015) dataset. Despite being first-order, our approach achieves substantial improvements over FID scores (Heusel et al., 2017), in comparison with higher-order samplers including DPMSolver-2, DPMSolver-3 (Lu et al., 2022b), and UniPC-3 (Zhao et al., 2023); see Section 5 for details.[1]

These results convey a surprising insight: the order of a diffusion sampler may not be the decisive factor governing practical sampling efficiency. While the focus of this paper is not to advocate our proposed samplers, they reveal a complementary and largely orthogonal lever to classical order analysis—*where* and *how* the DPM is evaluated and combined across time steps can matter just as much (and sometimes more) than the nominal solver order.

## 2. Other related work

**Training-free/based acceleration.** Existing acceleration strategies for diffusion sampling can be broadly divided into *training-free* and *training-based* approaches. Training-free methods reuse a fixed pre-trained DPM and speed up sampling solely by modifying the sampling update rule, making them broadly applicable to off-the-shelf DPMs. The methods discussed in the introduction belong to this category. In contrast, training-based strategies introduce an additional stage of training (e.g., distillation (Luhman & Luhman, 2021; Salimans & Ho, 2022; Meng et al., 2023) and consistency models (Song et al., 2023)). They aim to shorten sampling trajectories by adapting pre-trained models into related architectures, at the cost of additional training computation. Since our focus is training-free acceleration, we next summarize related work from both practical and theoretical viewpoints.

**Training-free acceleration: practice.** Most practical training-free accelerations are driven by higher-order numerical discretizations of the reverse-time SDE/ODE. For ODE-based samplers, a prominent line of work exploits the structure of the diffusion ODE to design higher-order solvers, including DPM-Solver++ (Lu et al., 2022a;b), exponential-integrator-based methods (Zhang & Chen, 2022), and predictor-corrector-based schemes such as UniPC (Zhao et al., 2023). For SDE-based samplers, acceleration is comparatively less explored due to the intrinsic difficulty of SDE discretization, but notable progress includes stochastic Improved Euler's methods (Jolicoeur-Martineau et al., 2021) and stochastic Adams methods (Xue et al., 2024).

---

[1]In our experiments, we do not incorporate certain implementation tricks (e.g., projecting to the valid pixel range or reducing the solver order in the final steps), which may lead to minor discrepancies with results reported in the prior works.

**Training-free acceleration: theory.** Convergence theory has been established for a wide range of diffusion samplers (Chen et al., 2022; Lee et al., 2023; Chen et al., 2023a; Li et al., 2023; Chen et al., 2023b; Huang et al., 2024a; Benton et al., 2023; Li & Yan, 2024; Li et al., 2025a), mostly focusing on standard DDPM and DDIM. For provable acceleration, most existing results likewise concentrate on higher-order discretization and on controlling the resulting discretization error for the reverse-time dynamics (Li & Cai, 2024; Li & Jiao, 2024; Jiao & Li, 2024; Huang et al., 2024a;b; Li et al., 2024; Yu & Yu, 2025; Li et al., 2025b; Gupta et al., 2024; Wu et al., 2024). Beyond designing improved solvers, another line of work accelerates generation via parallel sampling that implements multiple denoising steps in parallel (Shih et al., 2023; Chen et al., 2024; Gupta et al., 2024). In addition to continuous diffusion models, sampling acceleration has also been studied for discrete diffusion models (Li & Cai, 2025; Zhao & Cai, 2026; Cai & Li, 2026; Chen et al., 2025; Dmitriev et al., 2026).

## 3. Problem setup

In this section, we review basics of DPM sampling.

**Forward process.** Consider a forward process $(\boldsymbol{x}_t)_{0 \le t \le T}$ in $\mathbb{R}^d$ whose marginal distribution at time $t \in [0, T]$ satisfies

$$\boldsymbol{x}_t \stackrel{\mathrm{d}}{=} \alpha_t \boldsymbol{x}_0 + \sigma_t \boldsymbol{z}, \tag{1}$$

where $\boldsymbol{x}_0 \sim q_0$ and $\boldsymbol{z} \sim \mathcal{N}(\boldsymbol{0}, \boldsymbol{I}_d)$ are generated independently. Let $q_t$ be the law or density of $\boldsymbol{x}_t$. The functions $\alpha_t, \sigma_t > 0$, called *noise schedules*, are chosen to ensure that $q_T \approx \mathcal{N}(0, \tilde{\sigma}^2 \boldsymbol{I}_d)$ and that the signal-to-noise (SNR) ratio $\alpha_t/\sigma_t$ is strictly decreasing in time $t$.

Such a forward process can be realized by the following SDE

$$\mathrm{d}\boldsymbol{x}_t = f(t)\boldsymbol{x}_t \, \mathrm{d}t + g(t) \, \mathrm{d}\boldsymbol{\omega}_t, \quad \boldsymbol{x}_0 \sim q_0, \tag{2}$$

where $(\boldsymbol{\omega}_t)_{t \ge 0}$ is a standard Brownian motion in $\mathbb{R}^d$, and the functions $f(t)$ and $g(t)$ are defined by

$$f(t) = \frac{\mathrm{d}\log\alpha_t}{\mathrm{d}t} \quad \text{and} \quad g^2(t) = \frac{\mathrm{d}\sigma_t^2}{\mathrm{d}t} - 2\sigma_t^2 \frac{\mathrm{d}\log\alpha_t}{\mathrm{d}t}.$$

For more details, see e.g., Kingma et al. (2021).

**Reverse process.** According to classical results on time-reversal of SDEs (Anderson, 1982), the process in (2) has an associated reverse-time process that runs backward in time from $T$ to 0, governed by the SDE

$$\mathrm{d}\boldsymbol{x}_t = \big[f(t)\boldsymbol{x}_t - g^2(t)\boldsymbol{s}_t^\star(\boldsymbol{x}_t)\big]\,\mathrm{d}t + g(t)\,\mathrm{d}\bar{\boldsymbol{\omega}}_t, \quad \boldsymbol{x}_T \sim q_T,$$

where $(\bar{\boldsymbol{\omega}}_t)_{t \ge 0}$ is standard reverse Brownian motion. Here, $\boldsymbol{s}_t^\star(\cdot) := \nabla \log q_t(\cdot)$ represents the score function associated with the marginal distribution $q_t$ of the forward process

at time $t$. Instead of running this reverse SDE, an alternative is to use a deterministic ODE process, known as the *probability flow ODE*, given by

$$\dot{\boldsymbol{x}}_t = f(t)\boldsymbol{x}_t - \frac{1}{2}g^2(t)\boldsymbol{s}_t^\star(\boldsymbol{x}_t), \quad \boldsymbol{x}_T \sim q_T. \tag{3}$$

The forward process and these two reverse processes share the same marginal distribution; see e.g., Song et al. (2020b) for details. Since $q_T \approx \mathcal{N}(0, \tilde{\sigma}^2 \boldsymbol{I}_d)$ is easy to sample from, both reverse processes enable sampling from the target distribution $q_0$, provided that the score functions $\boldsymbol{s}_t^\star(\cdot)$ can be accurately estimated.

**Noise prediction model.** For any $t \in [0, T]$, the score function $\boldsymbol{s}_t^\star(\cdot)$ associated with $q_t$ satisfies

$$\boldsymbol{s}_t^\star(\cdot) = \underset{s(\cdot)\,:\,\mathbb{R}^d \to \mathbb{R}^d}{\arg\min} \, \mathbb{E}\Big[\big\|\boldsymbol{s}(\alpha_t\boldsymbol{x}_0 + \sigma_t\boldsymbol{z}) + \boldsymbol{z}/\sigma_t\big\|_2^2\Big],$$

where $\boldsymbol{x}_0 \sim q_0$ and $\boldsymbol{z} \sim \mathcal{N}(\boldsymbol{0}, \boldsymbol{I}_d)$ are generated independently. This motivates the use of a large neural network $\boldsymbol{\varepsilon}_\theta(\boldsymbol{x}, t)$, parameterized by $\theta$, to approximate the scaled score function $-\sigma_t \boldsymbol{s}_t^\star(\boldsymbol{x})$. The parameter $\theta$ is optimized by minimizing

$$\int_0^T \omega(t) \, \mathbb{E}_{\boldsymbol{x}_0 \sim q_0, \boldsymbol{z} \sim \mathcal{N}(\boldsymbol{0}, \boldsymbol{I}_d)} \Big[\big\|\boldsymbol{\varepsilon}_\theta(\alpha_t\boldsymbol{x}_0 + \sigma_t\boldsymbol{\varepsilon}, t) - \boldsymbol{z}\big\|_2^2\Big] \, \mathrm{d}t,$$

where $\omega(t) > 0$ is a weight function. Since $\boldsymbol{\varepsilon}_\theta(\boldsymbol{x}_t, t)$ can be viewed as a predictor for the Gaussian noise $\boldsymbol{\varepsilon}$ used in generating $\boldsymbol{x}_t$ (through $\boldsymbol{x}_t = \alpha_t\boldsymbol{x}_0 + \sigma_t\boldsymbol{z}$, where both $\boldsymbol{x}_0 \sim q_0$ and $\boldsymbol{z} \sim \mathcal{N}(\boldsymbol{0}, \boldsymbol{I}_d)$ are not observed), it is known as the *noise prediction model*.

**Diffusion ODE.** Given a pre-trained noise prediction model $\boldsymbol{\varepsilon}_\theta(\boldsymbol{x}, t)$, we can instantiate the probability flow ODE (3) by substituting the unknown score function $\boldsymbol{s}_t^\star(\boldsymbol{x})$ with $-\boldsymbol{\varepsilon}_\theta(\boldsymbol{x}, t)/\sigma_t$. With the Gaussian initialization $\boldsymbol{x}_T \sim \mathcal{N}(0, \tilde{\sigma}^2 \boldsymbol{I}_d)$, this yields the *diffusion ODE*:

$$\dot{\boldsymbol{x}}_t = f(t)\boldsymbol{x}_t + \frac{g^2(t)}{2\sigma_t}\boldsymbol{\varepsilon}_\theta(\boldsymbol{x}_t, t), \quad \boldsymbol{x}_T \sim \mathcal{N}(0, \tilde{\sigma}^2 \boldsymbol{I}_d). \tag{4}$$

Sampling from the target distribution $q_0$ can then be achieved by numerically solving this diffusion ODE backward from $T$ to 0. By applying a time reparameterization $\lambda(t) := \log(\alpha_t/\sigma_t)$ (which is strictly decreasing on $[0, T]$) and its inverse $t(\lambda)$, the solution $\boldsymbol{x}_t$ to the diffusion ODE (4) at any time $t < s$ can be expressed as

$$\boldsymbol{x}_t = \frac{\alpha_t}{\alpha_s}\boldsymbol{x}_s - \alpha_t \int_{\lambda(s)}^{\lambda(t)} \mathrm{e}^{-\lambda} \boldsymbol{\varepsilon}_\theta\big(\boldsymbol{x}_{t(\lambda)}, t(\lambda)\big) \, \mathrm{d}\lambda. \tag{5}$$

For a detailed derivation, see Lu et al. (2022b).

**Diffusion ODE solvers.** In this paragraph, we summarize several popular diffusion samplers that are based on solving the diffusion ODE (5). We first discretize the time horizon from $T$ to 0 into $(M+1)$ time steps and solve backward from $T$ to 0:

$$T = t_0 > t_1 > \cdots > t_{M-1} > t_M = 0. \qquad (6)$$

Starting from an initial state $\boldsymbol{x}_{t_0}$ sampled from a Gaussian distribution, the goal is to sequentially compute $\boldsymbol{x}_{t_1}, \ldots, \boldsymbol{x}_{t_M}$ that approximate the exact ODE solution at the corresponding times, given the initial condition $\boldsymbol{x}_{t_0}$ at time $T$. The key challenge is to achieve accurate approximation using only a limited number of evaluations of the noise prediction model $\boldsymbol{\varepsilon}_\theta(\boldsymbol{x}, t)$. For notional convenience, we use the abbreviation

$$\boldsymbol{\varepsilon}_t := \boldsymbol{\varepsilon}_\theta(\boldsymbol{x}_t, t)$$

whenever it is clear from context.

- **Deterministic DDIM.** The deterministic DDIM (Song et al., 2020a), which can be interpreted as a first-order solver, admits the following update rule:

$$\boldsymbol{x}_{t_i} = \frac{\alpha_{t_i}}{\alpha_{t_{i-1}}} \boldsymbol{x}_{t_{i-1}} - \left( \frac{\alpha_{t_i} \sigma_{t_{i-1}}}{\alpha_{t_{i-1}}} - \sigma_{t_i} \right) \boldsymbol{\varepsilon}_{t_{i-1}}. \quad (7)$$

- **Higher-order ODE solvers.** A $p$-th order ODE solver constructs a $p$-th order approximation to the diffusion ODE (5) for $p \geq 2$. A general form is given as follows (see e.g., Lu et al. (2022a)):

$$\boldsymbol{x}_{t_i} = \text{ODESolver}p(\boldsymbol{x}_{t_{i-1}}, \boldsymbol{\varepsilon}_{t_{i-1}}, \cdots, \boldsymbol{\varepsilon}_{t_{i-p}}) \qquad (8)$$

$$:= \frac{\alpha_{t_i}}{\alpha_{t_{i-1}}} \boldsymbol{x}_{t_{i-1}} - \alpha_{t_i} e^{-\lambda_{t_{i-1}}} \sum_{k=0}^{p-1} \frac{\varphi_k(\lambda_{t_i} - \lambda_{t_{i-1}})}{k!} \boldsymbol{y}_{k+1},$$

where the functions $\varphi_k : \mathbb{R} \to \mathbb{R}$, for $k = 0, \cdots, p-1$, are defined by

$$\varphi_0(x) = 1 - e^{-x},$$

$$\varphi_k(x) = \int_0^x e^{-\lambda} \lambda^k \, d\lambda = k\varphi_{k-1}(x) - x^k e^{-x}, \ k \geq 1,$$

and $\boldsymbol{y}_k^\top$, for $k \in [p]$, is the $k$-th row of the solution $\boldsymbol{Y}$ to

$$\boldsymbol{A}\boldsymbol{Y} = \boldsymbol{B}, \ \boldsymbol{A} = \left[ \frac{(\lambda_{t_{i-j}} - \lambda_{t_{i-1}})^{k-1}}{(k-1)!} \right]_{1 \leq j,k \leq p} \in \mathbb{R}^{p \times p},$$

$$\boldsymbol{B} = [\boldsymbol{\varepsilon}_{t_{i-j}}^\top]_{1 \leq j \leq p} \in \mathbb{R}^{p \times d}.$$

In practice, one typically takes $p = 2$ or 3.

- **UniPC.** UniPC (Zhao et al., 2023) follows a predictor-corrector design: it first generates a corrected intermediate state and then performs the actual update using

a (multi-step) ODE solver. In our notation, one can view UniPC-p as using the $p$-order solver to compute an intermediate corrected iterate

$$\boldsymbol{x}_{t_{i-1}}^{\text{cor}} = \text{ODESolver}p(\boldsymbol{x}_{t_{i-2}}^{\text{cor}}, \boldsymbol{\varepsilon}_{t_{i-1}}, \cdots, \boldsymbol{\varepsilon}_{t_{i-p}}), \ (9a)$$

where ODESolver$p$ is defined in (8), and $\boldsymbol{\varepsilon}_{t_{i-k}}$ is evaluated at $\boldsymbol{x}_{t_{i-k}}$. Subsequently, it updates $\boldsymbol{x}_{t_i}$ by a $q = (p-1)$-order rule:

$$\boldsymbol{x}_{t_i} = \text{ODESolver}q(\boldsymbol{x}_{t_{i-1}}^{\text{cor}}, \boldsymbol{\varepsilon}_{t_{i-1}}, \cdots, \boldsymbol{\varepsilon}_{t_{i-q}}). \quad (9b)$$

To conclude this paragraph, we emphasize that all the above ODE solvers can be interpreted as approximate the integral in (5) by using the *right-endpoint* evaluation of the noise prediction model $\boldsymbol{\varepsilon}_\theta(\boldsymbol{x}_{t_{i-1}}, t_{i-1})$.

**Notation.** For vector $\boldsymbol{x}$, we denote by $\|\boldsymbol{x}\|_2$ or $\|\boldsymbol{x}\|$ its $\ell_2$ norm. For matrix $\boldsymbol{A}$, we denote by $\|\boldsymbol{A}\|$ its spectral norm. In the rest of the paper, we use $\lambda(t)$ or $\lambda_t$ interchangeably to denote the mapping from the parameter $t$ to $\lambda$. For random vector $\boldsymbol{x}_t$ defined in (1), we let $q_t$ denote its probability density function. For function $f(M)$, we write $f(M) = O(M^{-k})$ if there exists some constant $C > 0$ such that $|f(M)| \leq CM^{-k}$ for all $M \geq 1$. Similarly, we denote $f(M) = \Omega(M^{-k})$ if $|f(M)| \geq cM^{-k}$ holds for some constant $c > 0$, and $f(M) = \Theta(M^{-k})$ if both $f(M) = O(M^{-k})$ and $f(M) = \Omega(M^{-k})$ hold. Finally, we say $f(M) = o(M^{-k})$ if $\lim_{M \to \infty} M^k f(M) = 0$.

## 4. Main results

In this section, we present our proposed first-order forward-value sampler along with its convergence guarantee.

### 4.1. Algorithm

**Motivation.** As discussed in Section 3, diffusion sampling can be interpreted as numerically solving the diffusion ODE (5). The essential challenge lies in accurately approximating the integral term involving the noise prediction model $\boldsymbol{\varepsilon}_\theta$ when we only have access to its evaluations at finite time steps.

A natural and straightforward approach, given the initial condition $\boldsymbol{x}_s$ at time $s$, is to use the noise prediction model $\boldsymbol{\varepsilon}_\theta(\boldsymbol{x}_s, s)$ evaluated at the backward endpoint $s$ to approximate the integrand $\boldsymbol{\varepsilon}_\theta(\boldsymbol{x}_{t(\lambda)}, x_{t(\lambda)})$ for all $\lambda \in [\lambda(t), \lambda(s)]$. Indeed, this first-order, backward-value discretization leads to the widely used DDIM sampler:

$$\boldsymbol{x}_t \approx \frac{\alpha_t}{\alpha_s} \boldsymbol{x}_s - \alpha_t \boldsymbol{\varepsilon}_\theta(\boldsymbol{x}_s, s) \int_{\lambda(s)}^{\lambda(t)} e^{-\lambda} \, d\lambda$$

$$= \frac{\alpha_t}{\alpha_s} \boldsymbol{x}_s - \left( \frac{\alpha_t}{\alpha_s} \sigma_s - \sigma_t \right) \boldsymbol{\varepsilon}_\theta(\boldsymbol{x}_s, s).$$

Recall from Section 3 that a noise prediction model $\varepsilon_\theta(\boldsymbol{x}_t, t)$ seeks to predict the noise component in the noisy data $\boldsymbol{x}_t = \alpha_t \boldsymbol{x}_0 + \sigma_t \boldsymbol{z}$ where $\boldsymbol{x}_0$ is the clean data and $\boldsymbol{z}$ is the unobserved noise. It is often convenient to consider the associated *data prediction model*:

$$\boldsymbol{\mu}_\theta(\boldsymbol{x}_t, t) := \frac{1}{\alpha_t}\big(\boldsymbol{x}_t - \sigma_t \varepsilon_\theta(\boldsymbol{x}_t, t)\big), \qquad (10)$$

which can be interpreted as predicting the underlying clean data $\boldsymbol{x}_0$ from the noisy input $\boldsymbol{x}_t$. This interpretation is particularly transparent at $t = 0$, since $\sigma_0 = 0$ and $\alpha_0 = 1$, so that $\boldsymbol{\mu}_\theta(\boldsymbol{x}_0, 0) = \boldsymbol{x}_0$, i.e., it directly outputs the clean data $\boldsymbol{x}_0$. With this definition, we can rewrite the DDIM update by substituting $\varepsilon_\theta(\boldsymbol{x}_s, s) = \big(\boldsymbol{x}_s - \alpha_s \boldsymbol{\mu}_\theta(\boldsymbol{x}_s, s)\big)/\sigma_s$, which yields

$$\boldsymbol{x}_t = \frac{\sigma_t}{\sigma_s}\boldsymbol{x}_s + \Big(\alpha_t - \frac{\sigma_t}{\sigma_s}\alpha_s\Big)\boldsymbol{\mu}_\theta(\boldsymbol{x}_s, s). \qquad (11)$$

In light of its intuitive interpretation, let us revisit the discretization of the diffusion ODE from the perspective of the data prediction model $\boldsymbol{\mu}_\theta$. Since the update effectively holds the DPM fixed over a finite interval, it inevitably introduces discretization error, which in turn governs the convergence rate of the sampler. This raises a natural question: can we design a more accurate approximation scheme by changing how (and where) the data predictor is evaluated? To this end, let us fix an end time $T$ and consider two simple and extreme scenarios regarding the number of iterations ( equivalently, the number of discretization steps) $M$.

- *Iterations sufficiently few.* Let us begin with the case where the number of iterations takes a small value $M = 1$. In this case, the initial time $s$ corresponds to $s = T$ and the end time $t$ corresponds to $t = 0$. Thus, if we replace the integrand $\varepsilon_\theta(\boldsymbol{x}_{t(\lambda)}, t(\lambda))$ with the forward value $\varepsilon_\theta(\boldsymbol{x}_0, 0)$, the resulting first-order, forward-value discretization becomes

$$\frac{\alpha_0}{\alpha_T}\boldsymbol{x}_T - \alpha_0 \varepsilon_\theta(\boldsymbol{x}_0, 0)\int_{\lambda(T)}^{\lambda(0)} e^{-\lambda}\, d\lambda$$

$$= \frac{\alpha_0}{\alpha_T}\boldsymbol{x}_T - \Big(\frac{\alpha_0}{\alpha_T}\sigma_T - \sigma_0\Big)\varepsilon_\theta(\boldsymbol{x}_0, 0)$$

$$= \frac{\sigma_0}{\sigma_T}\boldsymbol{x}_T + \Big(\alpha_0 - \frac{\sigma_0}{\sigma_T}\alpha_T\Big)\boldsymbol{\mu}_\theta(\boldsymbol{x}_0, 0),$$

where in the second line, we rewrite the expression in terms of the data prediction model $\boldsymbol{\mu}_\theta$ as in the previous derivation in (11). Now, recognizing that $\boldsymbol{\mu}_\theta(\boldsymbol{x}_0, 0) = \boldsymbol{x}_0$, $\sigma_0 = 0$, and $\alpha_0 = 1$, we find that

$$\frac{\sigma_0}{\sigma_T}\boldsymbol{x}_T + \Big(\alpha_0 - \sigma_0\frac{\alpha_T}{\sigma_T}\Big)\boldsymbol{\mu}_\theta(\boldsymbol{x}_0, 0) = \boldsymbol{x}_0,$$

implying that the forward-value approximation exactly recover the clean data without any discretization error.

**Algorithm 1** F-DPMSolver

---

1: **Input:** Noise prediction model $\varepsilon_\theta(\boldsymbol{x}, t)$, time grid $t_0 > t_1 > \cdots > t_M$, initialization $\boldsymbol{x}_{t_0}$.
2: Set $\boldsymbol{\mu}_\theta(\boldsymbol{x}_t, t) \leftarrow \big(\boldsymbol{x}_t - \sigma_t \varepsilon_\theta(\boldsymbol{x}_t, t)\big)/\alpha_t$.
3: **for** $i = 1, \ldots, M$ **do**
4:     Construct a lookahead estimate $\widehat{\boldsymbol{x}}_{t_i}$ for $\boldsymbol{x}_{t_i}$, using only information up to step $i - 1$ (e.g., one-step DDIM from $(\boldsymbol{x}_{t_{i-1}}, t_{i-1})$).
5:     Update

$$\boldsymbol{x}_{t_i} \leftarrow \frac{\sigma_{t_i}}{\sigma_{t_{i-1}}}\boldsymbol{x}_{t_{i-1}} - \Big(\frac{\sigma_{t_i}\alpha_{t_{i-1}}}{\sigma_{t_{i-1}}} - \alpha_{t_i}\Big)\boldsymbol{\mu}_\theta(\widehat{\boldsymbol{x}}_{t_i}, t_i).$$

6: **Output:** Generated sample $\boldsymbol{x}_{t_M}$.

---

- *Iterations sufficiently many.* Now consider the opposite case where the number of iterations $M$ approaches to infinity. In this case, one can expect that the discretization error is negligible regardless of whether we use the forward value $\varepsilon_\theta(\boldsymbol{x}_t, t)$ or the backward value $\varepsilon_\theta(\boldsymbol{x}_s, s)$ to approximate the integrand $\varepsilon_\theta(\boldsymbol{x}_{t(\lambda)}, t(\lambda))$ over the interval $[\lambda(s), \lambda(t)]$. Hence, the forward-value approximation also yields accurate results.

Taken together, these two extremes suggest that the forward-value discretization is, in principle, a plausible alternative to the standard backward-value choice (used by DDIM) — it is exact in the one-step case ($M = 1$) and becomes indistinguishable from the backward-value discretization as $M \to \infty$ when the step size vanishes. The practically relevant regime, however, is neither of these limits but rather a small number of iterations, where discretization error dominates and DPM evaluation placement matters. Figure 3 in Appendix A tests the performance of the forward-value discretization in finite number of iterations on CIFAR-10 and ImageNet dataset with resolution $64 \times 64$. Our observation suggests that for general $M$, forward-value evaluations can be systematically more accurate than the backward-value choice. This motivates the central question we study: *when $M$ is small, how accurate is the first-order, forward-value discretization, and can its potential advantage be exploited in a practical way?*

**First-order forward-value sampler.** Capitalizing on the above observation, we propose a practical first-order forward-value sampler named F-DPMSolver. In light of the above motivation, we find it more convenient to express the algorithm in terms of the data prediction model $\boldsymbol{\mu}_\theta$ defined in (10), which is computed deterministically from the pre-trained noise prediction model $\varepsilon_\theta$ and the known noise schedule.

For iteration $i = 1, \ldots, M$, we first form a one-step lookahead estimate $\widehat{\boldsymbol{x}}_{t_i}$ of the next state using only information

available up to step $i - 1$. We then evaluate the DPM at this lookahead estimate, $\boldsymbol{\mu}_\theta(\widehat{\boldsymbol{x}}_{t_i}, t_i)$, and combine it with the current iterate $\boldsymbol{x}_{t_{i-1}}$ to compute the next iterate $\boldsymbol{x}_{t_i}$ via

$$\boldsymbol{x}_{t_i} = \frac{\sigma_{t_i}}{\sigma_{t_{i-1}}}\boldsymbol{x}_{t_{i-1}} - \left(\frac{\sigma_{t_i}\alpha_{t_{i-1}}}{\sigma_{t_{i-1}}} - \alpha_{t_i}\right)\boldsymbol{\mu}_\theta(\widehat{\boldsymbol{x}}_{t_i}, t_i). \quad (12)$$

The complete procedure is summarized in Algorithm 1.

As a note, the lookahead estimate $\widehat{\boldsymbol{x}}_{t_i}$ can be generated by any algorithm that depends only on information available up to iteration $i - 1$. For instance, one may obtain $\widehat{\boldsymbol{x}}_{t_i}$ by taking a single DDIM step (7) from $\boldsymbol{x}_{t_{i-1}}$, and the resulting F-DPMSolver sampler is also called F-DDIM. Alternatively, one can use a higher-order predictor, e.g., the second-order solver (18), applied using $\boldsymbol{x}_{t_{i-1}}$ together with previously computed DPM evaluations such as $\boldsymbol{\varepsilon}_\theta(\widehat{\boldsymbol{x}}_{t_{i-1}}, t_{i-1})$ and $\boldsymbol{\varepsilon}_\theta(\widehat{\boldsymbol{x}}_{t_{i-2}}, t_{i-2})$. F-DPMSolver sampler with $p$-th order prediction is also called F-DPMSolver$p$. Note that for both F-DDIM and F-DPMSolver$p$, we perform only one DPM evaluation per iteration, i.e., $\boldsymbol{\mu}_\theta(\widehat{\boldsymbol{x}}_{t_i}, t_i)$.

### 4.2. Convergence analysis

Now that we have introduced our forward-value sampler, a natural question arises: how does its sampling convergence speed compare with existing diffusion ODE solvers such as DDIM and DPM-Solvers?

To answer this rigorously, let us first introduce the notion of convergence order of a diffusion sample, which allows us to compare the sampling speeds of diffusion samplers formally.

**Definition 1.** For a time grid $\{t_i\}_{i=0}^M$, we say a diffusion sampler has convergence order $k$ if

$$\left\|\boldsymbol{x}_{t_M} - \boldsymbol{x}_{t_M}^\star\right\|_2 = O(1/M^k),$$

where $\{\boldsymbol{x}_{t_i}^\star\}_{i=0}^M$ denotes the exact solution to the diffusion ODE (5) when initialized at $\boldsymbol{x}_{t_0}^\star = \boldsymbol{x}_{t_0}$, i.e., for $i \in [M]$,

$$\boldsymbol{x}_{t_i}^\star = \frac{\alpha_{t_i}}{\alpha_{t_0}}\boldsymbol{x}_{t_0}^\star - \alpha_{t_i}\int_{\lambda_{t_0}}^{\lambda_{t_i}} e^{-\lambda}\boldsymbol{\varepsilon}_\theta\big(\boldsymbol{x}_{t(\lambda)}^\star, t(\lambda)\big)\, d\lambda. \quad (13)$$

**Convergence orders of existing ODE solvers.** With this definition in hand, we now provide convergence guarantees for the diffusion ODE solvers presented in Section 3.

In order to establish convergence results, we make the following assumptions on the time discretization and the pre-trained DPM. To stay consistent with the sampler description in (12), we impose regularity assumptions on the DPM through the associated data prediction model $\boldsymbol{\mu}_\theta$ defined in (10), which has one-to-one correspondence with the noise prediction model $\boldsymbol{\varepsilon}_\theta$.

**Assumption 1.** We assume that the time discretization grid satisfies $\max_{1 \le i \le M}(\lambda_{t_i} - \lambda_{t_{i-1}}) = O(1/M)$ and that the

noise schedule obeys $\alpha_{t_i} \ge \alpha_{t_{i-1}}$ and $\sigma_{t_i} \le \sigma_{t_{i-1}}$ for all $i \in [M]$. In addition, we assume that data prediction function $\boldsymbol{\mu}_\theta(\boldsymbol{x}, t)$ defined in (10) satisfies

(A1) $\|\nabla_{\boldsymbol{x}}\boldsymbol{\mu}_\theta(\boldsymbol{x}, t)\| \le L_x/\alpha_t$ for any $\boldsymbol{x} \in \mathbb{R}^d$ and $t \ge 0$;

(A2) $\|\nabla_{\boldsymbol{x}}^2\boldsymbol{\mu}_\theta(\boldsymbol{x}, t)\| \le H_x$ for any $\boldsymbol{x} \in \mathbb{R}^d$ and $t \ge 0$;

(A3) $\left\|\frac{\partial}{\partial\lambda}\boldsymbol{\mu}_\theta\big(\boldsymbol{x}_{t(\lambda)}^\star, t(\lambda)\big)\right\| \le L_t$ for any $\lambda \in \mathbb{R}$;

(A4) $\left\|\frac{\partial^2}{\partial\lambda^2}\boldsymbol{\mu}_\theta\big(\boldsymbol{x}_{t(\lambda)}^\star, t(\lambda)\big)\right\| \le H_t$ for any $\lambda \in \mathbb{R}$;

(A5) $\left\|\frac{\partial}{\partial\lambda}\nabla_{\boldsymbol{x}}\boldsymbol{\mu}_\theta\big(\boldsymbol{x}_{t(\lambda)}^\star, t(\lambda)\big)\right\| \le H$ for any $\lambda \in \mathbb{R}$.

In words, Assumption 1 requires the time discretization to be sufficiently fine and the pre-trained DPM to be smooth in both the data variable $\boldsymbol{x}$ and the time variable $t$.

According to Lu et al. (2022a, Theorem 3.2) and Zheng et al. (2023, Theorem 3.3), we have the following upper bound on the convergence orders of the diffusion ODE solvers summarized in Section 3.

**Theorem 1** (Convergence order upper bound). *Under Assumption 1, the convergence orders of DDIM (7), the second-order ODE solver (18) and UniPC (19) are 1, 2 and 3, respectively.*

Theorem 1 formalizes the usual intuition: increasing the discretization order of a diffusion ODE solver leads to improved convergence rates.

Moreover, we show that these convergence rates are tight by establishing the following matching lower bound, even in the idealized setting when the true score functions are available (i.e., there is no score matching error). For analytical simplicity, our analysis focuses on $p = 1$ and $p = 2$. The analysis for higher-order follows analogously. The proof is postponed to Appendix C.

**Theorem 2** (Convergence order lower bound). *For any time grid $\{\lambda_{t_i}\}_{i=0}^M$ satisfying $\lambda_{t_i} - \lambda_{t_{i-1}} = O(M^{-1})$, there exists some distribution such that the convergence order of the deterministic DDIM (7) is at most 1. In addition, the convergence order of the second-order ODE solver (18) is at most 2.*

The lower bound indicates that there is no generic free improvement in convergence order for DDIM or standard second-order solvers without changing the underlying approximation strategy.

**Convergence order of forward-value discretization.** Now let us consider the (idealized) first-oder forward-value discretization of the diffusion ODE (5). For each iteration $i = 1, \ldots, M$, given the current iterate $\boldsymbol{x}_{t_{i-1}}^{\text{for}}$ at time $t_{i-1}$,

we compute the next iterate $\boldsymbol{x}_{t_i}^{\text{for}}$ by

$$\boldsymbol{x}_{t_i}^{\text{for}} = \frac{\sigma_{t_i}}{\sigma_{t_{i-1}}}\boldsymbol{x}_{t_{i-1}}^{\text{for}} + \left(\alpha_{t_i} - \frac{\sigma_{t_i}\alpha_{t_{i-1}}}{\sigma_{t_{i-1}}}\right)\boldsymbol{\mu}_\theta(\boldsymbol{x}_{t_i}^{\text{for}}, t_i). \tag{14}$$

For comparison, the deterministic DDIM update (7) can be written in the data-prediction form as

$$\boldsymbol{x}_{t_i}^{\text{bck}} = \frac{\sigma_{t_i}}{\sigma_{t_{i-1}}}\boldsymbol{x}_{t_{i-1}}^{\text{bck}} + \left(\alpha_{t_i} - \frac{\sigma_{t_i}\alpha_{t_{i-1}}}{\sigma_{t_{i-1}}}\right)\boldsymbol{\mu}_\theta(\boldsymbol{x}_{t_{i-1}}^{\text{bck}}, t_{i-1}). \tag{15}$$

The only difference between (14) and (15) is whether the data prediction model $\boldsymbol{\mu}_\theta(\boldsymbol{x}, t)$ is evaluated at the next state/time $(\boldsymbol{x}_{t_i}^{\text{for}}, t_i)$, or the current state/time $(\boldsymbol{x}_{t_{i-1}}^{\text{bck}}, t_{i-1})$. As a note, while the update rule of the naive forward-value discretization (14) is not directly implementable, it serves as a clean lens for understanding how evaluation placement affects discretization error.

Interestingly, it turns out that the first-order discretization errors introduced by these two strategies have exactly opposite signs, as formalized in the following theorem. The proof is postponed to Appendix B.1.

**Theorem 3.** *Under Assumption 1, the idealized forward-value discretization scheme* (14) *has convergence order 1. Moreover, let $\boldsymbol{x}_{t_M}^{\text{for}}$ and $\boldsymbol{x}_{t_M}^{\text{bck}}$ denote the outputs of the forward-value discretization* (14) *and DDIM* (15), *respectively, and let $\boldsymbol{x}_{t_M}^\star$ be the exact ODE solution defined in* (13), *with the common initialization $\boldsymbol{x}_{t_0}^{\text{bck}} = \boldsymbol{x}_{t_0}^{\text{for}} = \boldsymbol{x}_{t_0}^\star$. Then we have*

$$\left\| \boldsymbol{x}_{t_M}^{\text{bck}} - \boldsymbol{x}_{t_M}^\star + \boldsymbol{x}_{t_M}^{\text{for}} - \boldsymbol{x}_{t_M}^\star \right\|_2 = O(1/M^2). \tag{16}$$

Theorem 3 implies that the leading $O(1/M)$ errors of the forward-value and backward-value discretizations cancel to second order:

$$\boldsymbol{x}_{t_M}^{\text{bck}} - \boldsymbol{x}_{t_M}^\star \approx -(\boldsymbol{x}_{t_M}^{\text{for}} - \boldsymbol{x}_{t_M}^\star).$$

The first-order error terms of both methods with respect to the target $x_{t_M}^\star$ share an identical norm and have opposite directions, while only higher-order error terms exhibit divergence.

**Convergence order of our sampler.** Finally, we provide the convergence guarantee of the proposed sampler in Theorem 4 below; with proof postponed to Appendix B.2.

**Theorem 4.** *Suppose that Assumption 1 holds. In addition, assume the the one-step lookahead estimate $\widehat{\boldsymbol{x}}_{t_i}$ satisfies $\|\widehat{\boldsymbol{x}}_{t_i} - \boldsymbol{x}_{t_i}^\star(\boldsymbol{x}_{t_{i-1}}, t_{i-1})\| = o(1/M)$ for all $i \in [M]$, where $\boldsymbol{x}_t^\star(\boldsymbol{x}_{t_{i-1}}, t_{i-1})$ denotes the solution to the diffusion ODE at time $t \leq t_{i-1}$ when initialized at time $t_{i-1}$ with state*

$\boldsymbol{x}_{t_{i-1}}$, *i.e.,*

$$\boldsymbol{x}_t^\star(\boldsymbol{x}_{t_{i-1}}, t_{i-1})$$
$$= \frac{\alpha_t}{\alpha_{t_{i-1}}}\boldsymbol{x}_{t_{i-1}} - \alpha_t \int_{\lambda_{t_{i-1}}}^{\lambda_t} \mathrm{e}^{-\lambda}\boldsymbol{\varepsilon}_{\theta,\lambda}^\star \, \mathrm{d}\lambda, \tag{17}$$

*where $\boldsymbol{\varepsilon}_{\theta,\lambda}^\star = \boldsymbol{\varepsilon}_\theta\big(\boldsymbol{x}_{t(\lambda)}^\star(\boldsymbol{x}_{t_{i-1}}, t_{i-1}), t(\lambda)\big)$. Then the proposed sampler* (12) *satisfies*

$$\left\| \boldsymbol{x}_{t_M} - \boldsymbol{x}_{t_M}^{\text{for}} \right\| = o(1/M),$$

*where $\boldsymbol{x}_t^{\text{for}}$ represents the idealized forward-value iterate defined in* (14) *with the same initialization as $\boldsymbol{x}_t$. In particular, the convergence order of the sampler* (12) *is 1.*

This theorem guarantees that, as long as the one-step lookahead $\widehat{\boldsymbol{x}}_{t_i}$ approximates the target $\boldsymbol{x}_{t_i}^\star(\boldsymbol{x}_{t_{i-1}}, t_{i-1})$ with an error of order $o(1/M)$ (given that the time step $\lambda_{t_{i-1}} - \lambda_{t_i} = O(1/M)$), the implemented method tracks the idealized forward-value trajectory closely enough that it inherits the same first-order rate. This matters because the idealized forward-value discretization is precisely the object that exhibits signed error cancellation with DDIM, as established in Theorem 3. Note that here the condition about $\widehat{\boldsymbol{x}}_{t_i}$ concerns the one-step lookahead error from $t_{i-1}$ to $t_i$, rather than the cumulative error over the full trajectory. Specifically, the target $\boldsymbol{x}_{t_i}^\star(\boldsymbol{x}_{t_{i-1}}, t_{i-1})$ denotes the exact ODE solution at time $t_i$ initialized from $(\boldsymbol{x}_{t_{i-1}}, t_{i-1})$ instead of $(\boldsymbol{x}_{t_0}, t_0)$. Moreover, both F-DDIM and F-DPMSolver2 satisfy this condition naturally.

Even more interesting, despite being only a first-order method, we observe that the sampler (14) converges even faster than the second-order ODE solver (18) when $\widehat{\boldsymbol{x}}_{t_i}$ provides a reasonably accurate proxy for the next-step iterate. This phenomenon conveys a surprising message: the discretization error introduced by the first-order, forward-value scheme (14) is less harmful than that of the backward-value approximation (7), or even the second-order solver (18). This supports the main takeaway suggested by the theorems above: where the DPM is evaluated (and how those evaluations are combined) can matter as much as nominal solver order for practical sampling efficiency.

Finally, we compare our method with predictor-corrector samplers. Classical predictor-corrector samplers are primarily constructed to increase the numerical order of the solver; for example, in methods such as UniPC, the corrector is explicitly designed to achieve a higher-order approximation than the predictor. In contrast, our method does not aim to improve the order of the solver and remains first-order overall (cf. Theorem 4). Instead, it explores a different design axis — namely, the choice of evaluation location of the denoising model — and analyzes how this affects discretization error.

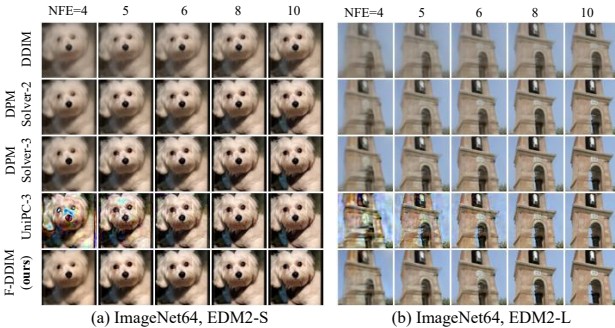

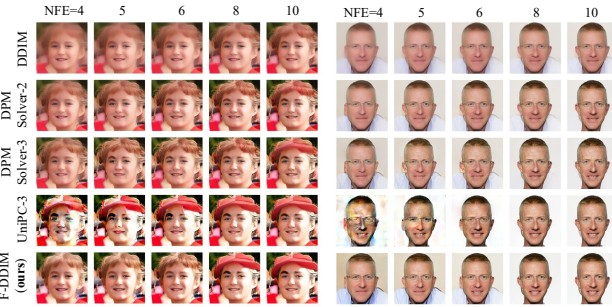

*Figure 1.* Qualitative comparisons between F-DDIM (ours) and DDIM, DPMSolver-2, DPMSolver-3, and UniPC-3. Images are sampled from pre-trained EDM2 with S and L size on ImageNet64 dataset.

*Figure 2.* Qualitative comparisons between F-DDIM (ours) and DDIM, DPMSolver-2, DPMSolver-3, and UniPC-3. Images are sampled from pre-trained latent diffusion model on FFHQ dataset.

## 5. Experiments

In this section, we present extensive experiments to compare the performance of our sampler and high-order samplers.

**Experiment setup.** We conduct experiments on four datasets, employing different pre-trained data prediction models: (1) CIFAR-10 (Krizhevsky et al., 2009) with Diffusion-Based Generative Models (EDM) (Karras et al., 2022); (2) ImageNet dataset (Deng et al., 2009) in $64 \times 64$ resolution with EDM2 in size S and L; (3) ImageNet dataset (Deng et al., 2009) in $512 \times 512$ resolution with EDM2 in size XS and XXL; (4) large scale Scene Understanding (LSUN) dataset (Yu et al., 2015) and the Flickr-Faces-HQ (FFHQ) dataset (Karras et al., 2019) with latent diffusion models using noise predictors (Rombach et al., 2022; Blattmann et al., 2022). For all datasets, we evaluate samplers with a number of NFEs $M \in \{4, 5, 6, 8, 10\}$. [2]

We compare our sampler with DDIM, the second and third-order ODE solvers (denoted by DPMSolver-2 and 3), and the third-order UniPC (i.e., UniPC-3); see Section 3 for their details.

The noise schedules $\alpha_{t_i}$ and $\sigma_{t_i}$ are subsampled from the reference schedules $\alpha_t^{\text{ref}}$ and $\sigma_t^{\text{ref}}$. Specifically, for the reference schedules defined over $M^{\text{ref}}$ steps, the sampler with $M$ iterations adopts $\alpha_{t_i} = \alpha_{t_{i'}}^{\text{ref}}$, $\sigma_{t_i} = \sigma_{t_{i'}}^{\text{ref}}$, where $i' = \text{round}(iM^{\text{ref}}/M)$.

**Qualitative comparisons.** Figures 1–7 provide qualitative comparisons across all considered models and settings, spanning both conditional (Figure 1, 5, 6) and unconditional generation (Figure 2, 4, 7), as well as pixel-space (Figure 1, 4, 5) and latent-space sampling (Figure 2, 6, 7). Across

these settings, our sampler consistently generates superior samples with richer fine-grained details, with the advantage most pronounced at low NFEs. In comparison, DDIM and higher-order solvers (DPMSolver-2/3 and UniPC-3) more often lack similar visual details or show localized artifacts under the same compute budget. Due to space constraints, Figures 4–7 are provided in Appendix A.

**Quantitative results (FID).** Tables 1–5 report FID scores, computed over 50K generated samples, across pixel-space (Tables 1, 4) and latent space models (Tables 2, 5), and for conditional (Tables 1, 2, 4) and unconditional sampling (Tables 1, 5) under varying NFEs, where lower FIDs generally indicate better sample quality. Across all datasets and model sizes, our sampler achieves significantly lower scores than DDIM, and obtains at least comparable performance with all of the high-order algorithms. Particularly, on ImageNet512 and 64 (Tables 2 and 4), our sampler remains the top performer across all tested NFEs and model sizes, with the largest gains at small NFEs, and on CIFAR-10 in Table 1, our sampler achieves the lowest FID throughout the low-to-mid compute regime (up to NFE = 8). Due to space constraints, we defer Tables 4–5 to Appendix A.

Finally, we also validate the effectiveness of our proposed forward-value framework when augmented with higher-order solvers; see Figures 8–9 and Tables 6–7 in Appendix A.3 for details.

**Results with larger NFE.** To show the quality as the number of steps increases, we present results at higher NFEs. In particular, we consider a large model, as it typically requires more sampling steps. We report FIDs on ImageNet512 using the EDM2-XXL model for NFE = $10, 12, 15, 20$, as shown in Table 3. As NFE increases, our method remains comparable to higher-order samplers such as DPMSolver and UniPC across this range, further supporting our claims.

---

[2]For the LSUN and FFHQ datasets, since no public reference statistics are available for computing FID, we calculate them ourselves. As a result, our FID scores may differ from those reported in previous studies.

*Table 1.* The FID of different samplers on CIFAR-10 dataset across varying NFEs, including both unconditional (EDM-uncond) and conditional generation (EDM-cond).

| Model | NFE | High-order Algorithms | | | First-order Algorithms | |
|---|---|---|---|---|---|---|
| | | DPMSolver-2 | DPMSolver-3 | UniPC-3 | DDIM | F-DDIM (**ours**) |
| EDM -uncond | 4 | 43.55 | 33.77 | 111.60 | 66.72 | **25.01** |
| | 5 | 27.86 | 18.20 | 55.14 | 49.62 | **16.04** |
| | 6 | 17.70 | 10.68 | 58.86 | 35.58 | **9.47** |
| | 8 | 9.74 | 5.60 | 9.83 | 22.30 | **4.91** |
| | 10 | 6.52 | 3.88 | **2.86** | 15.67 | 3.46 |
| EDM -cond | 4 | 31.60 | 23.99 | 78.70 | 47.27 | **18.44** |
| | 5 | 20.63 | 13.68 | 41.50 | 35.57 | **12.29** |
| | 6 | 14.04 | 8.66 | 39.27 | 26.78 | **7.57** |
| | 8 | 8.34 | 5.00 | 5.61 | 17.57 | **4.32** |
| | 10 | 5.89 | 3.61 | **2.55** | 13.00 | 3.18 |

*Table 2.* FID comparison of different samplers on ImageNet512 dataset across varying NFEs.

| Model | NFE | High-order Algorithms | | | First-order Algorithms | |
|---|---|---|---|---|---|---|
| | | DPMSolver-2 | DPMSolver-3 | UniPC-3 | DDIM | F-DDIM (**ours**) |
| EDM2 -XS | 4 | 66.23 | 59.35 | 217.23 | 87.81 | **50.96** |
| | 5 | 38.02 | 28.06 | 94.35 | 61.62 | **22.69** |
| | 6 | 28.24 | 16.11 | 30.64 | 53.89 | **14.33** |
| | 8 | 12.19 | 7.01 | 8.51 | 31.17 | **6.18** |
| | 10 | 7.45 | 4.93 | 4.90 | 20.97 | **4.57** |
| EDM2 -XXL | 4 | 68.18 | 60.11 | 62.93 | 91.48 | **52.32** |
| | 5 | 40.61 | 28.41 | 34.02 | 65.26 | **23.56** |
| | 6 | 28.46 | 16.57 | 16.43 | 53.94 | **14.04** |
| | 8 | 12.06 | 6.03 | 5.63 | 32.40 | **5.04** |
| | 10 | 6.59 | 3.60 | 3.30 | 21.48 | **3.06** |

*Table 3.* FID comparison of different samplers on ImageNet512 datasets with EDM2-XXL model across larger NFEs.

| NFE | DPM Solver-2 | DPM Solver-3 | UniPC -3 | DDIM | F-DDIM (**ours**) |
|---|---|---|---|---|---|
| 10 | 6.58 | 3.60 | 3.30 | 21.48 | **3.06** |
| 12 | 4.44 | 2.80 | 2.58 | 15.02 | **2.48** |
| 15 | 3.19 | 2.39 | 2.26 | 9.76 | **2.23** |
| 20 | 2.52 | 2.16 | **2.11** | 5.98 | **2.11** |

## 6. Discussion

In this paper, we have developed a novel, training-free acceleration method for sampling from DPMs. Despite its first-order nature, our method leverages a forward-value discretization strategy that achieves significant empirical improvements over standard first-order samplers like DDIM, and even competes with state-of-the-art higher-order methods. This result challenges the conventional wisdom that higher-order accuracy is necessary for effective acceleration in diffusion sampling, and reveals a distinct lever for acceleration that is orthogonal to classical order analysis.

Moving forward, several extensions appear natural. First, it would be valuable to develop a more rigorous theoretical understanding of the proposed forward-value discretization, including a characterization of why it yields practical gains beyond standard first-order schemes. Second, extending the framework to diffusion SDE sampling is an important direction, where stochasticity introduces additional challenges such as the interaction between discretization and noise injection. Third, it remains to be explored how the method behaves under classifier guidance and, more broadly, classifier-free guidance, where guidance strength can amplify errors and alter the effective dynamics. Finally, it is of interest to investigate applications beyond unconditional/conditional generation, such as leveraging the forward-value principle in diffusion-prior inverse problems (e.g., deblurring, super-resolution, and related reconstruction tasks), where measurement consistency constraints may interact nontrivially with the sampling discretization.

## Acknowledgements

Y. Jiao and G. Li are supported in part by the Chinese University of Hong Kong Direct Grant for Research and the Hong Kong Research Grants Council ECS 2191363 and GRF 2131005. C. Cai is supported in part by the NSF grants DMS-2515333.

## Impact Statement

This paper presents work whose goal is to advance the field of diffusion models. There are many potential societal consequences of our work, none of which we feel must be specifically highlighted here.

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

*Table 4.* The FID of different samplers on ImageNet64 dataset across varying NFEs.

| Model | NFE | High-order Algorithms | | | First-order Algorithms | |
|---|---|---|---|---|---|---|
| | | DPMSolver-2 | DPMSolver-3 | UniPC-3 | DDIM | F-DDIM (**ours**) |
| EDM2 -S | 4 | 27.00 | 21.50 | 50.43 | 39.38 | **20.55** |
| | 5 | 16.14 | 11.16 | 25.88 | 28.11 | **10.70** |
| | 6 | 10.66 | 6.96 | 13.88 | 20.65 | **6.67** |
| | 8 | 5.83 | 3.74 | 5.07 | 12.56 | **3.50** |
| | 10 | 3.98 | 2.71 | 2.79 | 8.77 | **2.56** |
| EDM2 -L | 4 | 29.91 | 23.66 | 49.99 | 43.87 | **22.36** |
| | 5 | 18.16 | 12.57 | 26.91 | 31.41 | **11.98** |
| | 6 | 11.90 | 7.63 | 15.25 | 23.22 | **7.21** |
| | 8 | 6.40 | 3.92 | 5.77 | 14.20 | **3.64** |
| | 10 | 4.26 | 2.71 | 2.71 | 9.85 | **2.51** |

# A. Further experimental results

## A.1. Motivation validation

We valid our motivation in Section 4.1 by implementing the forward-value discretization (12) with the estimation $\widehat{x}_{t_i} \approx x_{t_i}^\star$. Specifically, we compute $\widehat{x}_{t_i}$ by using a 5-step DDIM iteration from $t_{i-1}$ to $t_i$ starting from $x_{t_{i-1}}$. This procedure is intended solely for validating our motivation and is not a practical sampler. The result, presented in Figure 3, demonstrates that the forward-value discretization maintains a low FID across varying values of $M$. This indicates that this scheme performs effectively for any finite number of iterations, extending beyond the one-step and infinite-step limits discussed in Section 4.1.

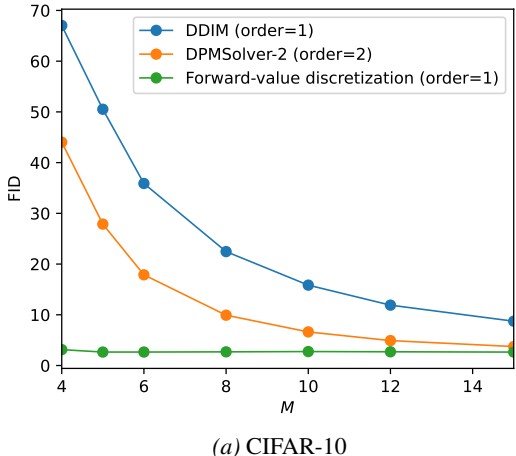
*(a)* CIFAR-10

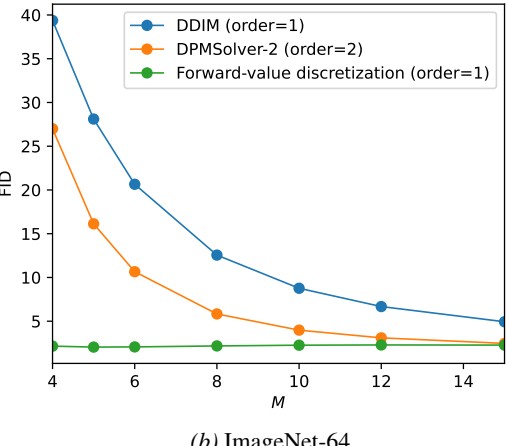
*(b)* ImageNet-64

*Figure 3.* Comparison of deterministic DDIM, second-order DPMSolver (see (18)), and the forward-value discretization on datasets CIFAR-10 (data prediction model: EDM-uncond) and ImageNet-64 (EDM2-S) under various number of iterations $M$.

## A.2. Further results on our sampler

Before presenting additional experimental results, we first detail the specific update formulations for ODESolver-2 and ODESolver-3. Taking $p = 2$ or 3 in (8) leads to the following schemes:

- **Second-order ODE solver:**

$$\begin{aligned}
x_{t_i} &= \mathsf{ODESolver2}(x_{t_{i-1}}, \varepsilon_{t_{i-1}}, \varepsilon_{t_{i-2}}) \\
&:= \frac{\alpha_{t_i}}{\alpha_{t_{i-1}}} x_{t_{i-1}} + \alpha_{t_i}\left[(\mathrm{e}^{-\lambda_{t_i}} - \mathrm{e}^{-\lambda_{t_{i-1}}})\varepsilon_{t_{i-1}} + \phi_1 D_1\right],
\end{aligned} \tag{18a}$$

where we define

$$\boldsymbol{D}_1 := \frac{\boldsymbol{\varepsilon}_{t_{i-1}} - \boldsymbol{\varepsilon}_{t_{i-2}}}{\lambda_{t_{i-1}} - \lambda_{t_{i-2}}} \qquad \text{and} \qquad \phi_1 := (\lambda_{t_i} - \lambda_{t_{i-1}} + 1)\mathrm{e}^{-\lambda_{t_i}} - \mathrm{e}^{-\lambda_{t_{i-1}}}. \tag{18b}$$

- **Third-order ODE solver:**

$$\begin{aligned}
\boldsymbol{x}_{t_i} &= \mathsf{ODESolver3}(\boldsymbol{x}_{t_{i-1}}, \boldsymbol{\varepsilon}_{t_{i-1}}, \boldsymbol{\varepsilon}_{t_{i-2}}, \boldsymbol{\varepsilon}_{t_{i-3}}) \\
&= \frac{\alpha_{t_i}}{\alpha_{t_{i-1}}} \boldsymbol{x}_{t_{i-1}} + \alpha_{t_i}\bigg\{ \left(\mathrm{e}^{-\lambda_{t_i}} - \mathrm{e}^{-\lambda_{t_{i-1}}}\right)\boldsymbol{\varepsilon}_{t_{i-1}} \\
&\qquad + \frac{\phi_1[(\lambda_{t_{i-1}} - \lambda_{t_{i-3}})\boldsymbol{D}_1 - (\lambda_{t_{i-1}} - \lambda_{t_{i-2}})\boldsymbol{D}_2]}{\lambda_{t_{i-2}} - \lambda_{t_{i-3}}} + \frac{\phi_2(\boldsymbol{D}_1 - \boldsymbol{D}_2)}{\lambda_{t_{i-2}} - \lambda_{t_{i-3}}} \bigg\},
\end{aligned} \tag{19a}$$

where in addition to $\boldsymbol{D}_1$ and $\phi_1$ defined in (18b), we further define

$$\boldsymbol{D}_2 := \frac{\boldsymbol{\varepsilon}_{t_{i-1}} - \boldsymbol{\varepsilon}_{t_{i-3}}}{\lambda_{t_{i-1}} - \lambda_{t_{i-3}}} \qquad \text{and} \qquad \phi_2 := (\lambda_{t_i} - \lambda_{t_{i-1}})^2 \mathrm{e}^{-\lambda_{t_i}} + 2\phi_1. \tag{19b}$$

We will use them to implement DPMSolver-2, DPMSolver-3, as well as UniPC-3 presented in (9) with $p = 3$.

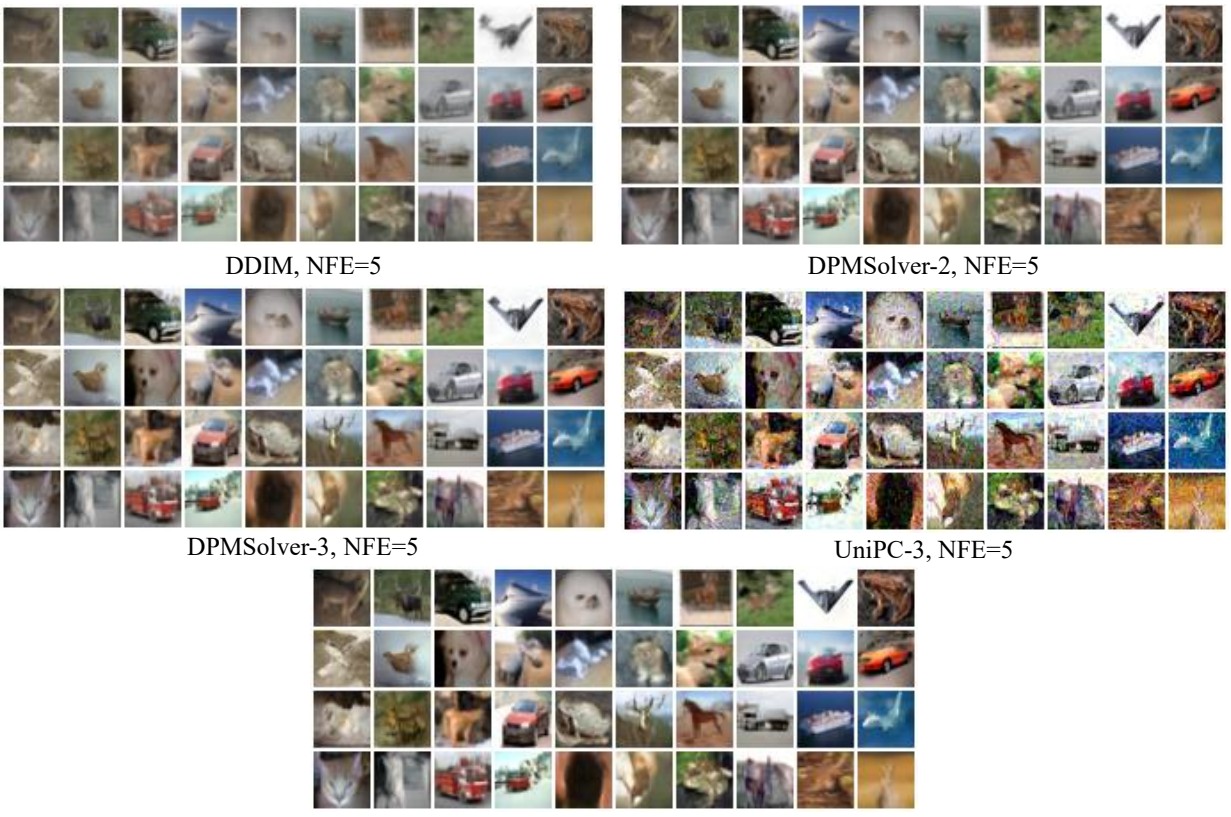

DDIM, NFE=5

DPMSolver-2, NFE=5

DPMSolver-3, NFE=5

UniPC-3, NFE=5

F-DDIM (**ours**), NFE=5

*Figure 4.* Qualitative comparisons between F-DDIM (ours) and DDIM, DPMSolver-2, DPMSolver-3, and UniPC-3. Images are sampled from pre-trained unconditional EDM on CIFAR10 dataset.

As mentioned in Section 5, Tables 4 and 5 present FID results on ImageNet64, FFHQ, and LSUN datasets. Figures 4 and 5 present sampling results on CIFAR-10 dataset, including unconditional and conditional generation with NFE= 5; Figure 6 present sampling results on ImageNet-64 dataset under various NFEs. In all of these settings, our sampler consistently generate much clearer samples with more visual details than the first-order algorithm DDIM, comparable with high-order algorithms DPMSolver-2/3, and avoid localized artifacts appearing in samples generated by UniPC-3. Figure 7 presents

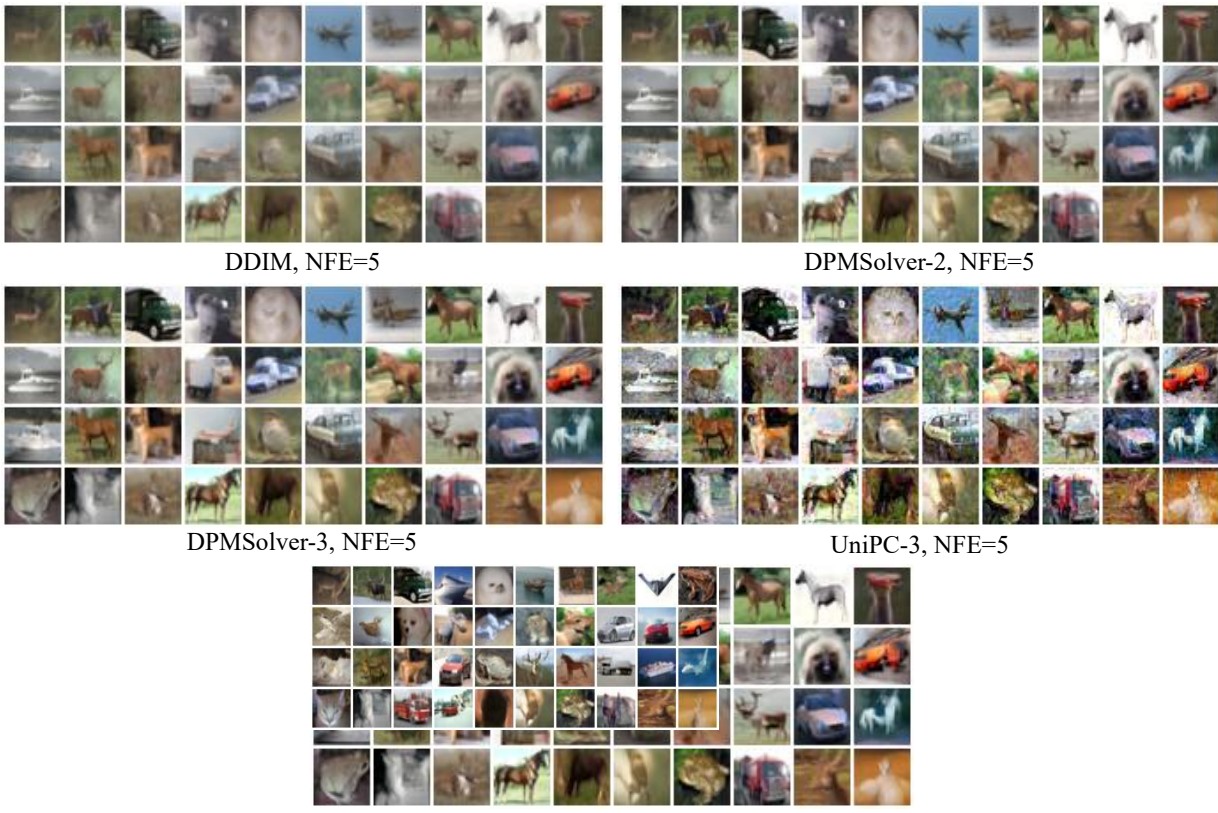

DDIM, NFE=5

DPMSolver-2, NFE=5

DPMSolver-3, NFE=5

UniPC-3, NFE=5

F-DDIM (**ours**), NFE=5

*Figure 5.* Qualitative comparisons between F-DDIM (ours) and DDIM, DPMSolver-2, DPMSolver-3, and UniPC-3. Images are sampled from pre-trained conditional EDM on CIFAR10 dataset.

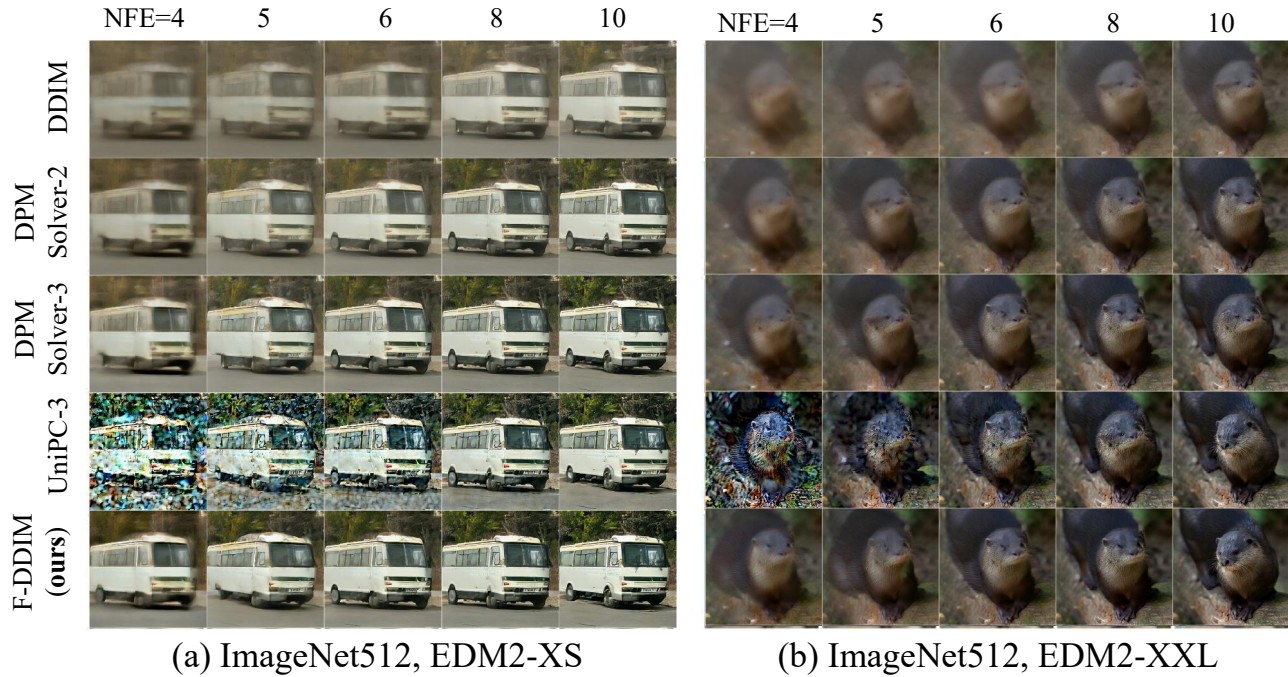

(a) ImageNet512, EDM2-XS

(b) ImageNet512, EDM2-XXL

*Figure 6.* Qualitative comparisons between F-DDIM (ours) and DDIM, DPMSolver-2, DPMSolver-3, and UniPC-3. Images are sampled from pre-trained EDM2 with XS and XXL size on ImageNet512 dataset.

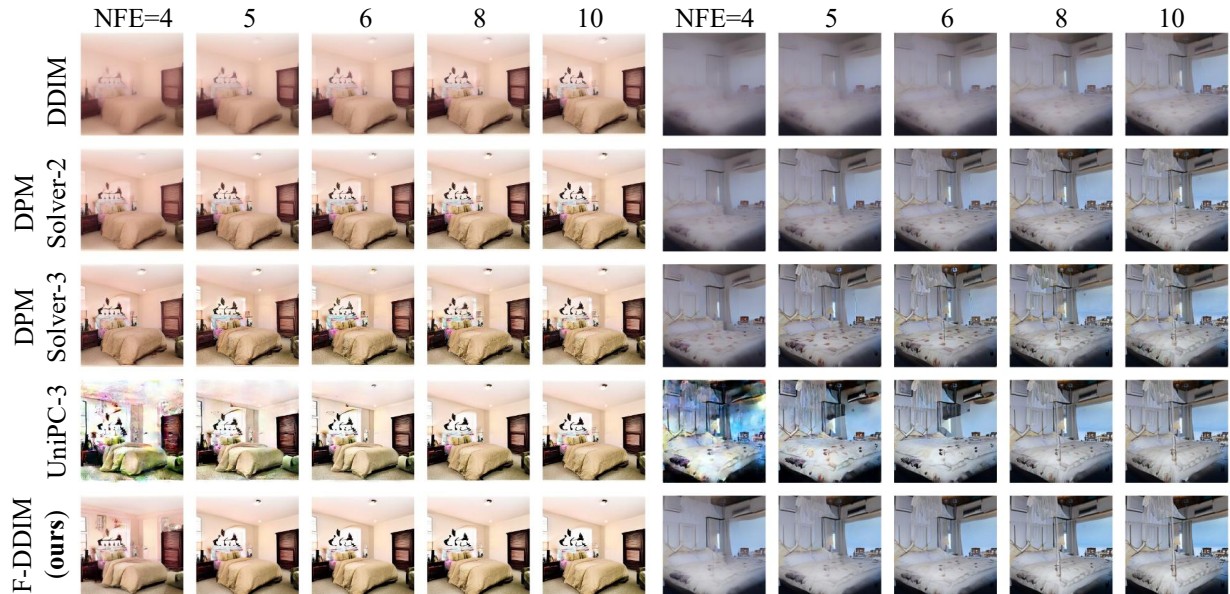

*Figure 7.* Qualitative comparisons between F-DDIM (ours) and DDIM, DPMSolver-2, DPMSolver-3, and UniPC-3. Images are sampled from pre-trained latent diffusion model on LSUN bedroom dataset.

*Table 5.* FID comparison of different samplers on latent diffusion model across varying NFEs.

| Dataset | NFE | High-order Algorithms | | | First-order Algorithms | |
|---------|-----|-------------|-------------|--------|------|---------------|
| | | DPMSolver-2 | DPMSolver-3 | UniPC-3 | DDIM | F-DDIM (**ours**) |
| FFHQ | 4 | 31.26 | **15.67** | 77.43 | 80.30 | 15.79 |
| | 5 | 16.33 | 11.28 | 21.93 | 58.42 | **9.75** |
| | 6 | 10.47 | 11.53 | 9.96 | 43.75 | **7.89** |
| | 8 | 7.19 | 12.08 | 7.37 | 27.05 | **6.80** |
| | 10 | 6.58 | 11.62 | 6.82 | 18.81 | **6.46** |
| LSUN Bedroom | 4 | 18.96 | **10.12** | 121.75 | 70.47 | 12.18 |
| | 5 | 8.66 | 8.52 | 24.74 | 41.01 | **6.98** |
| | 6 | 6.02 | 9.33 | 9.41 | 26.12 | **5.57** |
| | 8 | **4.78** | 9.26 | 5.72 | 13.68 | 4.93 |
| | 10 | **4.49** | 8.41 | 4.83 | 9.00 | 4.66 |

sampling results on LSUN bedroom dataset. Our sampler can generate high-quality samples close to the high-order reference samplers DPMSolver-2/3 and UniPC-3, which are significantly clearer than those generated by DDIM. In addition, we note that iPNDM (Zhang & Chen, 2022; Tong et al., 2025) achieves slightly better performance than our sampler on LSUN Bedroom in Table 5. The results are $50.43, 25.88, 13.88, 5.07, 2.79$ at NFE = $4, 5, 6, 8, 10$, respectively.

### A.3. Experimental results on augmenting with DPMSolver-2

Figures 8 and 9 show that augmenting our sampler with DPMSolver-2 can further improve perceptual fidelity at low NFEs, often yielding cleaner details and more consistent global structure. Notably, even with DPMSolver-2, our sampler remains a first-order algorithm, which means it retains $O(1/M)$ convergence rate as established in Theorems 3 and 4.

Tables 6 and 7 quantify a hybrid variant that augments our sampler with DPMSolver-2. It typically improves FID at low-to-mid NFEs, while at the highest NFE the gain may diminish or reverse depending on the setting or model size.

### A.4. Experimental results on text-to-image models

We have completed the experiments for Stable Diffusion, and the visual results are presented in Figure 10.

In the left subfigure of Figure 10, for the prompt "a desk and chair in an office cubicle", our forward-value sampler

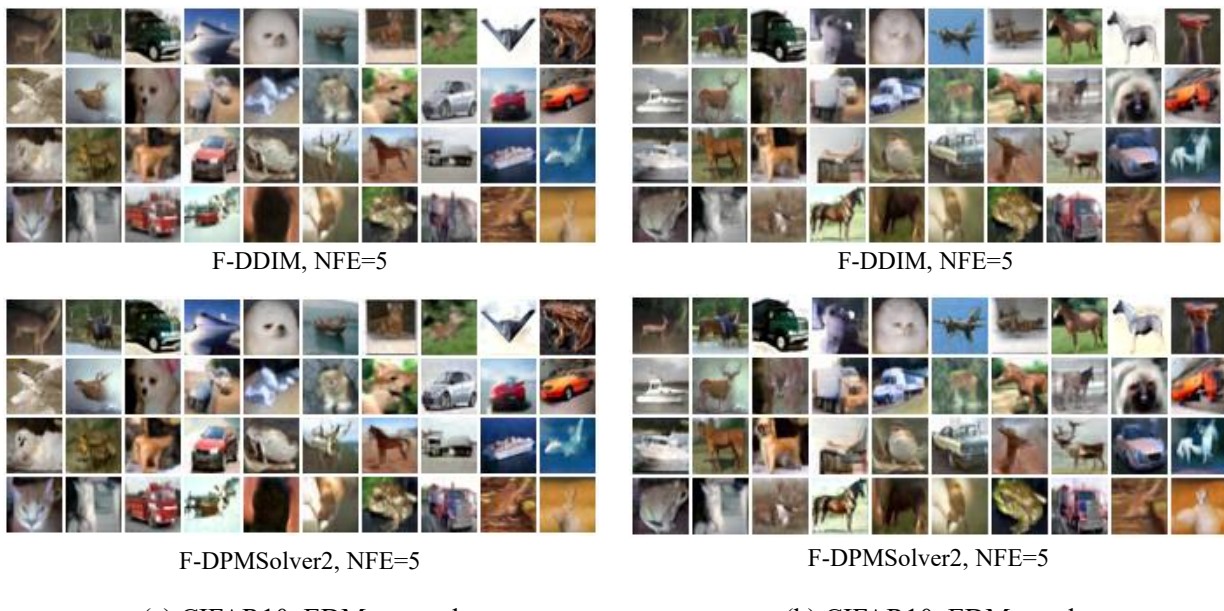

(a) CIFAR10, EDM-uncond          (b) CIFAR10, EDM-cond

*Figure 8.* Qualitative comparisons between our samplers F-DDIM and F-DPMSolver2. Images are sampled from pre-trained EDM model on CIFAR10 dataset.

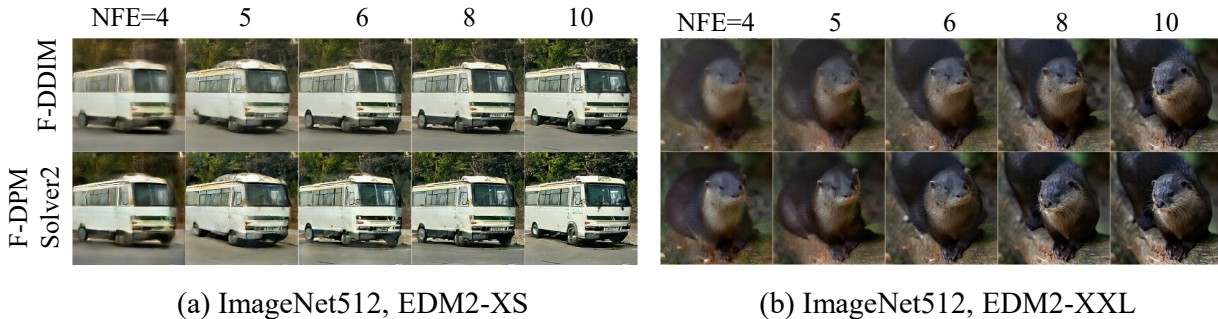

(a) ImageNet512, EDM2-XS          (b) ImageNet512, EDM2-XXL

*Figure 9.* Qualitative comparisons between our samplers F-DDIM and F-DPMSolver2. Images are sampled from pre-trained EDM2 model on ImageNet512 dataset.

*Table 6.* FID comparison between our sampler F-DDIM and the hybrid variant F-DPMSolver2 on CIFAR-10 and ImageNet512 datasets across different NFEs.

| NFE | EDM-uncond | | EDM-cond | |
|---|---|---|---|---|
| | F-DDIM | F-DPMSolver2 | F-DDIM | F-DPMSolver2 |
| 4 | 25.01 | **22.14** | 18.44 | **17.21** |
| 5 | 16.04 | **14.97** | 12.29 | **11.37** |
| 6 | 9.47 | **8.44** | 7.57 | **6.70** |
| 8 | 4.91 | **4.13** | 4.32 | **3.57** |
| 10 | 3.46 | **2.79** | 3.18 | **2.47** |

*Table 7.* FID comparison between our sampler F-DDIM and the hybrid variant F-DPMSolver2 on ImageNet512 datasets across different NFEs.

| NFE | EDM2-XS | | EDM2-XXL | |
|---|---|---|---|---|
| | F-DDIM | F-DPMSolver2 | F-DDIM | F-DPMSolver2 |
| 4 | 50.96 | **47.63** | 52.32 | **47.06** |
| 5 | 22.69 | **20.30** | 23.56 | **18.97** |
| 6 | 14.33 | **11.93** | 14.04 | **10.88** |
| 8 | 6.18 | **6.09** | 5.04 | **4.24** |
| 10 | **4.57** | 4.90 | 3.06 | **2.89** |

successfully generates a chair, whereas the backward-value samplers fail to do so. In the right subfigure of Figure 10, for the prompt "Four tennis players with rackets on a court", our forward-value sampler generates four players, while the backward-value samplers produce only three.

### A.5. Running time comparison

To show a practical speed-up, we make a run-time comparsion between our methods and reference samplers on ImageNet64 dataset with model EDM2-S. To provide a clear comparison across methods, we normalize the runtime by taking DDIM with NFE $= 10$ as a reference (set to 10), and report other methods using the normalized metric

$$\frac{\text{runtime}}{\text{runtime of DDIM} - 10} \times 10.$$

The results are reported in Table 8.

*Table 8.* Running time comparison

| NFE | 4 | 5 | 6 | 8 | 10 |
|---|---|---|---|---|---|
| DDIM | $4 \times 1.18$ | $5 \times 1.12$ | $6 \times 1.32$ | $8 \times 1.04$ | $10 \times 1.00$ |
| DPMSolver-2 | $4 \times 1.17$ | $5 \times 1.10$ | $6 \times 1.07$ | $8 \times 1.02$ | $10 \times 1.00$ |
| DPMSolver-3 | $4 \times 1.59$ | $5 \times 1.47$ | $6 \times 1.16$ | $8 \times 1.09$ | $10 \times 1.05$ |
| UniPC-3 | $4 \times 1.18$ | $5 \times 1.16$ | $6 \times 1.38$ | $8 \times 1.08$ | $10 \times 1.05$ |
| F-DDIM (ours) | $4 \times 1.15$ | $5 \times 1.08$ | $6 \times 1.06$ | $8 \times 1.03$ | $10 \times 0.99$ |
| F-DPMSovler2 (ours) | $4 \times 1.13$ | $5 \times 1.07$ | $6 \times 1.04$ | $8 \times 1.01$ | $10 \times 0.99$ |

## B. Proof of convergence order upper bounds (Theorems 3 and 4)

In this section, we analyze the convergence behavior of the idealized first-order forward-value discretization scheme (14) (Theorem 3) and our practical sampler (12) (Theorem 4).

For ease of presentation, we use the notation $\delta_{t_i} := \lambda_{t_i} - \lambda_{t_{i-1}}$ for each $i \in [M]$ throughout this section. In addition, $\nabla^k \boldsymbol{\mu}(\boldsymbol{x}, t)$ denotes the $k$-th order gradient of $\boldsymbol{\mu}(\boldsymbol{x}, t)$ with respect to $\boldsymbol{x}$.

Finally, we note that the diffusion ODE in (4) can be equivalently written in terms of the data prediction model $\boldsymbol{\mu}_\theta$. Indeed,

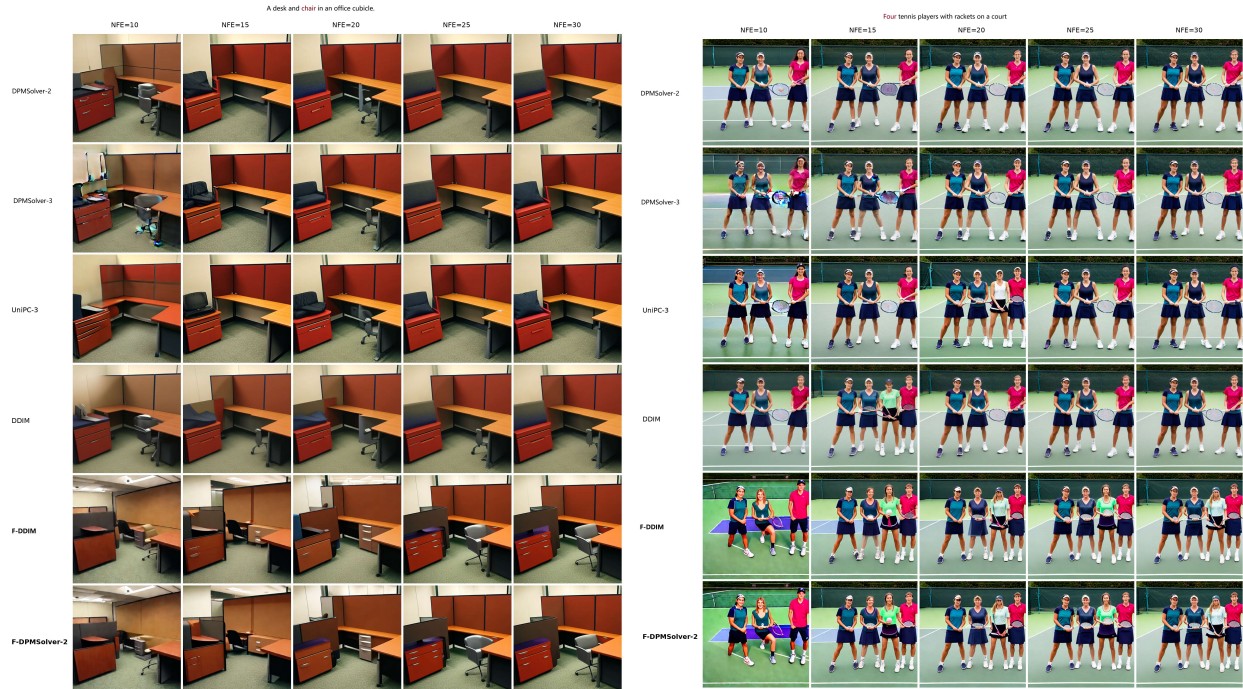

*Figure 10.* Comparisons of text-to-image results between F-DDIM (ours), F-DPMSolver2 (ours) and DDIM, DPMSolver-2, DPMSolver-3, and UniPC-3. The text of the left subfigure is "A desk and chair in an office cubicle", and the text of the right subfigure is "Four tennis players with rackets on a court". Images are sampled from the stable-diffusion-v1-5. The first three rows correspond to the higher-order samplers, while the last three rows correspond to the first-order samplers.

using the approximation $s_t^\star(\boldsymbol{x}) \approx (\alpha_t \boldsymbol{\mu}_\theta(\boldsymbol{x}, t) - \boldsymbol{x})/\sigma_t^2$ and substituting it into (3) yields the following diffusion ODE associated with a data prediction model:

$$\dot{\boldsymbol{x}}_t = \left( f(t) + \frac{g^2(t)}{2\sigma_t^2} \right) \boldsymbol{x}_t - \frac{\alpha_t g^2(t)}{2\sigma_t^2} \boldsymbol{\mu}_\theta(\boldsymbol{x}_t, t), \quad \boldsymbol{x}_T \sim \mathcal{N}(0, \tilde{\sigma}^2 \boldsymbol{I}_d). \tag{20}$$

Solving this ODE from $T$ to $0$ again produces a sample $\boldsymbol{x}_0$. Under the same change of variables $\lambda(t)$, the solution admits the integral form: for $t < s$,

$$\boldsymbol{x}_t = \frac{\sigma_t}{\sigma_s} \boldsymbol{x}_s + \sigma_t \int_{\lambda(s)}^{\lambda(t)} e^\lambda \boldsymbol{\mu}_\theta \big( \boldsymbol{x}_{t(\lambda)}, t(\lambda) \big) \, \mathrm{d}\lambda. \tag{21}$$

### B.1. Proof of Theorem 3

**Proof for the first-order forward-value discretization scheme.** By the diffusion ODE with the data prediction model (see (21)), for each $i \in [M]$, we can express $\boldsymbol{x}_{t_i}^\star$ in terms of $\boldsymbol{x}_{t_{i-1}}^\star$ as

$$\boldsymbol{x}_{t_i}^\star = \frac{\sigma_{t_i}}{\sigma_{t_{i-1}}} \boldsymbol{x}_{t_{i-1}}^\star + \sigma_{t_i} \int_{\lambda_{t_{i-1}}}^{\lambda_{t_i}} e^\lambda \boldsymbol{\mu}_\theta \big( \boldsymbol{x}_{t(\lambda)}^\star, t(\lambda) \big) \, \mathrm{d}\lambda. \tag{22}$$

Combining this identity with the forward-value discretization (14), we can decompose the distance between the iterates $\boldsymbol{x}_{t_i}^{\text{for}}$ and $\boldsymbol{x}_{t_i}^{\star}$ as

$$
\begin{aligned}
\boldsymbol{x}_{t_i}^{\text{for}} - \boldsymbol{x}_{t_i}^{\star} &= \frac{\sigma_{t_i}}{\sigma_{t_{i-1}}}\left(\boldsymbol{x}_{t_{i-1}}^{\text{for}} - \boldsymbol{x}_{t_{i-1}}^{\star}\right) + \sigma_{t_i} \int_{\lambda_{t_{i-1}}}^{\lambda_{t_i}} \mathrm{e}^{\lambda}\left(\boldsymbol{\mu}_\theta\left(\boldsymbol{x}_{t_i}^{\text{for}}, t_i\right) - \boldsymbol{\mu}_\theta\left(\boldsymbol{x}_{t(\lambda)}^{\star}, t(\lambda)\right)\right)\mathrm{d}\lambda \\
&= \frac{\sigma_{t_i}}{\sigma_{t_{i-1}}}\left(\boldsymbol{x}_{t_{i-1}}^{\text{for}} - \boldsymbol{x}_{t_{i-1}}^{\star}\right) + \sigma_{t_i} \int_{\lambda_{t_{i-1}}}^{\lambda_{t_i}} \mathrm{e}^{\lambda}\,\mathrm{d}\lambda \cdot \left(\boldsymbol{\mu}_\theta\left(\boldsymbol{x}_{t_i}^{\text{for}}, t_i\right) - \boldsymbol{\mu}_\theta\left(\boldsymbol{x}_{t_i}^{\star}, t_i\right)\right) \\
&\quad + \sigma_{t_i} \int_{\lambda_{t_{i-1}}}^{\lambda_{t_i}} \mathrm{e}^{\lambda}\left(\boldsymbol{\mu}_\theta\left(\boldsymbol{x}_{t_i}^{\star}, t_i\right) - \boldsymbol{\mu}_\theta\left(\boldsymbol{x}_{t(\lambda)}^{\star}, t(\lambda)\right)\right)\mathrm{d}\lambda.
\end{aligned}
\tag{23}
$$

Let us control the last two terms on the right-hand-side of (23) separately. For the second term, we can leverage the Lipschitz property of $\boldsymbol{\mu}_\theta(\boldsymbol{x}, t)$ with respect to $\boldsymbol{x}$ (see Assumption 1 (A1)) to derive

$$
\begin{aligned}
\left\|\sigma_{t_i}\int_{\lambda_{t_{i-1}}}^{\lambda_{t_i}} \mathrm{e}^{\lambda}\,\mathrm{d}\lambda \cdot \left(\boldsymbol{\mu}_\theta\left(\boldsymbol{x}_{t_i}^{\text{for}}, t_i\right) - \boldsymbol{\mu}_\theta\left(\boldsymbol{x}_{t_i}^{\star}, t_i\right)\right)\right\| &\leq \sigma_{t_i}\mathrm{e}^{\lambda_{t_i}}\delta_{t_i}\frac{L_x}{\alpha_{t_i}}\left\|\boldsymbol{x}_{t_i}^{\text{for}} - \boldsymbol{x}_{t_i}^{\star}\right\| \\
&= \delta_{t_i}L_x\left\|\boldsymbol{x}_{t_i}^{\text{for}} - \boldsymbol{x}_{t_i}^{\star}\right\|,
\end{aligned}
\tag{24}
$$

where the last inequality holds due to the fact that $\lambda_{t_i} \leq \lambda_{t_{i-1}}$ and $\lambda_t = \log(\alpha_t/\sigma_t)$. Similarly, we can control the third term by the Lipschitz property of $\boldsymbol{\mu}_\theta(\boldsymbol{x}, t)$ with respect to $t$ (see Assumption 1 (A3)):

$$
\begin{aligned}
\left\|\sigma_{t_i}\int_{\lambda_{t_{i-1}}}^{\lambda_{t_i}} \mathrm{e}^{\lambda}\left(\boldsymbol{\mu}_\theta\left(\boldsymbol{x}_{t_i}^{\star}, t_i\right) - \boldsymbol{\mu}_\theta\left(\boldsymbol{x}_{t(\lambda)}^{\star}, t(\lambda)\right)\right)\mathrm{d}\lambda\right\| &\leq \sigma_{t_i}\mathrm{e}^{\lambda_{t_i}}\int_{\lambda_{t_{i-1}}}^{\lambda_{t_i}} L_t(\lambda_{t_i} - \lambda)\,\mathrm{d}\lambda = \frac{1}{2}\sigma_{t_i}\mathrm{e}^{\lambda_{t_i}}L_t\delta_{t_i}^2 \\
&= \frac{1}{2}\alpha_{t_i}L_t\delta_{t_i}^2 = O(M^{-2}),
\end{aligned}
\tag{25}
$$

where the last step arises from the condition that $\delta_{t_i} = O(M^{-1})$. Substituting (24) and (25) into (23), we find that for each $i \in [M]$,

$$
\left\|\boldsymbol{x}_{t_i}^{\text{for}} - \boldsymbol{x}_{t_i}^{\star}\right\| \leq \frac{\sigma_{t_i}}{\sigma_{t_{i-1}}}\left\|\boldsymbol{x}_{t_{i-1}}^{\text{for}} - \boldsymbol{x}_{t_{i-1}}^{\star}\right\| + L_x\delta_{t_i}\left\|\boldsymbol{x}_{t_i}^{\text{for}} - \boldsymbol{x}_{t_i}^{\star}\right\| + O(M^{-2}),
\tag{26}
$$

or equivalently,

$$
\left\|\boldsymbol{x}_{t_i}^{\text{for}} - \boldsymbol{x}_{t_i}^{\star}\right\| \leq \frac{\sigma_{t_i}}{\sigma_{t_{i-1}}(1 - L_x\delta_{t_i})}\left\|\boldsymbol{x}_{t_{i-1}}^{\text{for}} - \boldsymbol{x}_{t_{i-1}}^{\star}\right\| + O(M^{-2}),
\tag{27}
$$

where the last step arises from the fact that $1 - L_x\delta_{t_i} = 1 - O(1/M) \asymp 1$.

Applying (27) recursively allows us to bound the final error as

$$
\left\|\boldsymbol{x}_{t_M}^{\text{for}} - \boldsymbol{x}_{t_M}^{\star}\right\| \leq O(M^{-2})\sum_{i=1}^{M}\prod_{j=i+1}^{M}\frac{\sigma_{t_j}}{\sigma_{t_{j-1}}(1 - L_x\delta_{t_j})} = O(M^{-1}).
\tag{28}
$$

Here, the first step holds as $\boldsymbol{x}_{t_0}^{\text{for}} = \boldsymbol{x}_{t_0}^{\star}$ and the last step arises from the following bound:

$$
\begin{aligned}
\sum_{i=1}^{M}\prod_{j=i+1}^{M}\frac{\sigma_{t_j}}{\sigma_{t_{j-1}}(1 - L_x\delta_{t_j})} &\overset{(a)}{\leq} \sum_{i=1}^{M}\frac{\sigma_{t_M}}{\sigma_{t_i}}\exp\left(2L_x\sum_{j=i+1}^{M}\delta_{t_j}\right) = \exp(2L_x\lambda_{t_M})\sum_{i=1}^{M}\frac{\sigma_{t_M}}{\sigma_{t_i}}\exp(-2L_x\lambda_{t_i}) \\
&\overset{(b)}{\leq} \exp(2L_x\lambda_{t_M})\sum_{i=1}^{M}\frac{\sigma_{t_M}}{\alpha_{t_i}} \overset{(c)}{\leq} M\exp(2L_x\lambda_{t_M})\frac{\sigma_{t_M}}{\alpha_{t_1}} = O(M),
\end{aligned}
$$

where (a) is true as $1/(1 - x) \leq \exp(2x)$ for $0 \leq x \leq 1/2$ and $\delta_{t_j}L_x = O(M^{-1}) \leq 1/2$ for all $j$; (b) holds as long as $2L_x \geq 1$; (c) follows from the fact that $\alpha_{t_i}$ is increasing in $i$.

This justifies that the convergence order of the first-order, forward-value discretization scheme is equal to 1.

**Proof of Claim** (16). Let us fix an arbitrary $i \in [M]$. Combining the expression of $\boldsymbol{x}_{t_i}^\star$ from (22) with the update rules for the deterministic DDIM $\boldsymbol{x}_{t_i}^{\text{bck}}$ (see (7)) and the first-order forward-value discretization $\boldsymbol{x}_{t_i}^{\text{for}}$ (see (14)), we can express the target difference as

$$\boldsymbol{x}_{t_i}^{\text{bck}} - \boldsymbol{x}_{t_i}^\star + \boldsymbol{x}_{t_i}^{\text{for}} - \boldsymbol{x}_{t_i}^\star$$

$$= \frac{\sigma_{t_i}}{\sigma_{t_{i-1}}} \big( \boldsymbol{x}_{t_{i-1}}^{\text{bck}} + \boldsymbol{x}_{t_{i-1}}^{\text{for}} - 2\boldsymbol{x}_{t_{i-1}}^\star \big) + \sigma_{t_i} \left( \int_{\lambda_{t_{i-1}}}^{\lambda_{t_i}} e^\lambda \Big( \boldsymbol{\mu}_\theta\big(\boldsymbol{x}_{t_{i-1}}^{\text{bck}}, t_{i-1}\big) + \boldsymbol{\mu}_\theta\big(\boldsymbol{x}_{t_i}^{\text{for}}, t_i\big) - 2\boldsymbol{\mu}_\theta\big(\boldsymbol{x}_{t(\lambda)}^\star, t(\lambda)\big) \Big) \, d\lambda \right)$$

$$= \frac{\sigma_{t_i}}{\sigma_{t_{i-1}}} \big( \boldsymbol{x}_{t_{i-1}}^{\text{bck}} + \boldsymbol{x}_{t_{i-1}}^{\text{for}} - 2\boldsymbol{x}_{t_{i-1}}^\star \big) + \sigma_{t_i} \underbrace{\left( \int_{\lambda_{t_{i-1}}}^{\lambda_{t_i}} e^\lambda \Big( \boldsymbol{\mu}_\theta\big(\boldsymbol{x}_{t_{i-1}}^\star, t_{i-1}\big) + \boldsymbol{\mu}_\theta\big(\boldsymbol{x}_{t_i}^\star, t_i\big) - 2\boldsymbol{\mu}_\theta\big(\boldsymbol{x}_{t(\lambda)}^\star, t(\lambda)\big) \Big) \, d\lambda \right)}_{=:\chi_1}$$

$$+ \sigma_{t_i} \underbrace{\int_{\lambda_{t_{i-1}}}^{\lambda_{t_i}} e^\lambda \Big( \boldsymbol{\mu}_\theta\big(\boldsymbol{x}_{t_{i-1}}^{\text{bck}}, t_{i-1}\big) + \boldsymbol{\mu}_\theta\big(\boldsymbol{x}_{t_{i-1}}^{\text{for}}, t_{i-1}\big) - 2\boldsymbol{\mu}_\theta\big(\boldsymbol{x}_{t_{i-1}}^\star, t_{i-1}\big) \Big) \, d\lambda}_{=:\chi_2}$$

$$+ \sigma_{t_i} \underbrace{\int_{\lambda_{t_{i-1}}}^{\lambda_{t_i}} e^\lambda \Big( \boldsymbol{\mu}_\theta\big(\boldsymbol{x}_{t_i}^{\text{for}}, t_i\big) - \boldsymbol{\mu}_\theta\big(\boldsymbol{x}_{t_i}^\star, t_i\big) - \boldsymbol{\mu}_\theta\big(\boldsymbol{x}_{t_{i-1}}^{\text{for}}, t_{i-1}\big) + \boldsymbol{\mu}_\theta\big(\boldsymbol{x}_{t_{i-1}}^\star, t_{i-1}\big) \Big) \, d\lambda}_{=:\chi_3} \qquad (29)$$

In what follows, we shall analyze $\chi_1$, $\chi_2$, and $\chi_3$ separately.

- **Controlling $\chi_1$.** For simplicity of notation, let us denote $\boldsymbol{\mu}(\lambda) := \boldsymbol{\mu}_\theta\big(\boldsymbol{x}_{t(\lambda)}^\star, t(\lambda)\big)$. We begin with decomposing $\chi_1$ as

$$\chi_1 = e^{\lambda_{t_i}} \int_{\lambda_{t_{i-1}}}^{\lambda_{t_i}} \Big( \boldsymbol{\mu}_\theta\big(\boldsymbol{x}_{t_{i-1}}^\star, t_{i-1}\big) + \boldsymbol{\mu}_\theta\big(\boldsymbol{x}_{t_i}^\star, t_i\big) - 2\boldsymbol{\mu}_\theta\big(\boldsymbol{x}_{t(\lambda)}^\star, t(\lambda)\big) \Big) \, d\lambda$$

$$+ \int_{\lambda_{t_{i-1}}}^{\lambda_{t_i}} \big( e^\lambda - e^{\lambda_{t_i}} \big) \Big( \boldsymbol{\mu}_\theta\big(\boldsymbol{x}_{t_{i-1}}^\star, t_{i-1}\big) + \boldsymbol{\mu}_\theta\big(\boldsymbol{x}_{t_i}^\star, t_i\big) - 2\boldsymbol{\mu}_\theta\big(\boldsymbol{x}_{t(\lambda)}^\star, t(\lambda)\big) \Big) \, d\lambda, \qquad (30)$$

  and control these two terms individually.

  - For the first term, we first rewrite it as

$$\int_{\lambda_{t_{i-1}}}^{\lambda_{t_i}} \Big( \boldsymbol{\mu}_\theta\big(\boldsymbol{x}_{t_{i-1}}^\star, t_{i-1}\big) + \boldsymbol{\mu}_\theta\big(\boldsymbol{x}_{t_i}^\star, t_i\big) - 2\boldsymbol{\mu}_\theta\big(\boldsymbol{x}_{t(\lambda)}^\star, t(\lambda)\big) \Big) \, d\lambda$$

$$= \int_{\lambda_{t_{i-1}}}^{\lambda_{t_i}} \Big( \boldsymbol{\mu}(\lambda_{t_{i-1}}) + \boldsymbol{\mu}(\lambda_{t_i}) - 2\boldsymbol{\mu}(\lambda) \Big) \, d\lambda$$

$$= 2 \int_{\lambda_{t_{i-1}}}^{\lambda_{t_i}} \left( \frac{\boldsymbol{\mu}(\lambda_{t_i}) - \boldsymbol{\mu}(\lambda_{t_{i-1}})}{\lambda_{t_i} - \lambda_{t_{i-1}}} (\lambda - \lambda_{t_{i-1}}) + \boldsymbol{\mu}(\lambda_{t_{i-1}}) - \boldsymbol{\mu}(\lambda) \right) d\lambda$$

$$+ \int_{\lambda_{t_{i-1}}}^{\lambda_{t_i}} \left( \boldsymbol{\mu}(\lambda_{t_i}) - \boldsymbol{\mu}(\lambda_{t_{i-1}}) - \frac{2\big(\boldsymbol{\mu}(\lambda_{t_i}) - \boldsymbol{\mu}(\lambda_{t_{i-1}})\big)}{\lambda_{t_i} - \lambda_{t_{i-1}}} (\lambda - \lambda_{t_{i-1}}) \right) d\lambda$$

$$= 2 \int_{\lambda_{t_{i-1}}}^{\lambda_{t_i}} \left( \frac{\boldsymbol{\mu}(\lambda_{t_i}) - \boldsymbol{\mu}(\lambda_{t_{i-1}})}{\lambda_{t_i} - \lambda_{t_{i-1}}} (\lambda - \lambda_{t_{i-1}}) + \boldsymbol{\mu}(\lambda_{t_{i-1}}) - \boldsymbol{\mu}(\lambda) \right) d\lambda, \qquad (31)$$

  where the last identity holds because

$$\int_{\lambda_{t_{i-1}}}^{\lambda_{t_i}} \left( \boldsymbol{\mu}(\lambda_{t_i}) - \boldsymbol{\mu}(\lambda_{t_{i-1}}) - \frac{2\big(\boldsymbol{\mu}(\lambda_{t_i}) - \boldsymbol{\mu}(\lambda_{t_{i-1}})\big)}{\lambda_{t_i} - \lambda_{t_{i-1}}} (\lambda - \lambda_{t_{i-1}}) \right) d\lambda$$

$$= \big( \boldsymbol{\mu}(\lambda_{t_i}) - \boldsymbol{\mu}(\lambda_{t_{i-1}}) \big) \delta_{t_i} - \frac{2\big(\boldsymbol{\mu}(\lambda_{t_i}) - \boldsymbol{\mu}(\lambda_{t_{i-1}})\big)}{\lambda_{t_i} - \lambda_{t_{i-1}}} \int_{\lambda_{t_{i-1}}}^{\lambda_{t_i}} (\lambda - \lambda_{t_{i-1}}) \, d\lambda$$

$$= \big( \boldsymbol{\mu}(\lambda_{t_i}) - \boldsymbol{\mu}(\lambda_{t_{i-1}}) \big) \delta_{t_i} - \frac{2\big(\boldsymbol{\mu}(\lambda_{t_i}) - \boldsymbol{\mu}(\lambda_{t_{i-1}})\big)}{\delta_{t_i}} \cdot \frac{1}{2} \delta_{t_i}^2 = 0.$$

Given the expression in (31), combining Taylor's theorem with Assumption 1 (A4), we can derive

$$\left\| \int_{\lambda_{t_{i-1}}}^{\lambda_{t_i}} \left( \frac{\boldsymbol{\mu}(\lambda_{t_i}) - \boldsymbol{\mu}(\lambda_{t_{i-1}})}{\lambda_{t_i} - \lambda_{t_{i-1}}} (\lambda - \lambda_{t_{i-1}}) + \boldsymbol{\mu}(\lambda_{t_{i-1}}) - \boldsymbol{\mu}(\lambda) \right) d\lambda \right\|$$

$$\leq \left\| \int_{\lambda_{t_{i-1}}}^{\lambda_{t_i}} \left( \frac{\boldsymbol{\mu}(\lambda_{t_i}) - \boldsymbol{\mu}(\lambda_{t_{i-1}})}{\lambda_{t_i} - \lambda_{t_{i-1}}} (\lambda - \lambda_{t_{i-1}}) - \frac{\partial}{\partial \lambda} \boldsymbol{\mu}(\lambda_{t_{i-1}})(\lambda - \lambda_{t_{i-1}}) \right) d\lambda \right\|$$

$$+ H_t \int_{\lambda_{t_{i-1}}}^{\lambda_{t_i}} (\lambda - \lambda_{t_{i-1}})^2 \, d\lambda$$

$$\leq \left\| \frac{\boldsymbol{\mu}(\lambda_{t_i}) - \boldsymbol{\mu}(\lambda_{t_{i-1}})}{\lambda_{t_i} - \lambda_{t_{i-1}}} - \frac{\partial}{\partial \lambda} \boldsymbol{\mu}(\lambda_{t_{i-1}}) \right\| \int_{\lambda_{t_{i-1}}}^{\lambda_{t_i}} (\lambda - \lambda_{t_{i-1}}) \, d\lambda + \frac{H_t}{3} \delta_{t_i}^3. \tag{32}$$

Applying Assumption 1 (A4) again, we can further bound the first term on the right-hand side of (32) as

$$\left\| \frac{\boldsymbol{\mu}(\lambda_{t_i}) - \boldsymbol{\mu}(\lambda_{t_{i-1}})}{\lambda_{t_i} - \lambda_{t_{i-1}}} - \frac{\partial}{\partial \lambda} \boldsymbol{\mu}(\lambda_{t_{i-1}}) \right\| \leq \left\| \frac{\partial}{\partial \lambda} \boldsymbol{\mu}(\lambda_{t_{i-1}}) \frac{\lambda_{t_i} - \lambda_{t_{i-1}}}{\lambda_{t_i} - \lambda_{t_{i-1}}} - \frac{\partial}{\partial \lambda} \boldsymbol{\mu}(\lambda_{t_{i-1}}) \right\| + H_t \delta_{t_i}^2 = H_t \delta_{t_i}^2.$$

Substituted into (32), this yields

$$\left\| \int_{\lambda_{t_{i-1}}}^{\lambda_{t_i}} \left( \frac{\boldsymbol{\mu}(\lambda_{t_i}) - \boldsymbol{\mu}(\lambda_{t_{i-1}})}{\lambda_{t_i} - \lambda_{t_{i-1}}} (\lambda - \lambda_{t_{i-1}}) + \boldsymbol{\mu}(\lambda_{t_{i-1}}) - \boldsymbol{\mu}(\lambda) \right) d\lambda \right\|$$

$$\leq \frac{1}{2} H_t \delta_{t_i}^3 + \frac{1}{3} H_t \delta_{t_i}^3 = O(M^{-3}). \tag{33}$$

Finally, plugging (33) into (31), the first term on the right-hand side of (30) can be bounded as

$$\int_{\lambda_{t_{i-1}}}^{\lambda_{t_i}} \left( \boldsymbol{\mu}_\theta\left(\boldsymbol{x}_{t_{i-1}}^\star, t_{i-1}\right) + \boldsymbol{\mu}_\theta\left(\boldsymbol{x}_{t_i}^\star, t_i\right) - 2\boldsymbol{\mu}_\theta\left(\boldsymbol{x}_{t(\lambda)}^\star, t(\lambda)\right) \right) d\lambda = O(M^{-3}). \tag{34}$$

– Turning to the second term, one can derive

$$\left\| \int_{\lambda_{t_{i-1}}}^{\lambda_{t_i}} (e^\lambda - e^{\lambda_{t_i}}) \left( \boldsymbol{\mu}_\theta\left(\boldsymbol{x}_{t_{i-1}}^\star, t_{i-1}\right) + \boldsymbol{\mu}_\theta\left(\boldsymbol{x}_{t_i}^\star, t_i\right) - 2\boldsymbol{\mu}_\theta\left(\boldsymbol{x}_{t(\lambda)}^\star, t(\lambda)\right) \right) d\lambda \right\|$$

$$\leq \left| e^{\lambda_{t_{i-1}}} - e^{\lambda_{t_i}} \right| \int_{\lambda_{t_{i-1}}}^{\lambda_{t_i}} \left\| \boldsymbol{\mu}_\theta\left(\boldsymbol{x}_{t_{i-1}}^\star, t_{i-1}\right) + \boldsymbol{\mu}_\theta\left(\boldsymbol{x}_{t_i}^\star, t_i\right) - 2\boldsymbol{\mu}\left(\boldsymbol{x}_{t(\lambda)}^\star, t(\lambda)\right) \right\| d\lambda$$

$$\leq e^{\lambda_{t_i}} \delta_{t_i} \int_{\lambda_{t_{i-1}}}^{\lambda_{t_i}} L_t (\lambda - \lambda_{t_{i-1}} + \lambda_{t_i} - \lambda) \, d\lambda$$

$$= e^{\lambda_{t_i}} L_t \delta_{t_i}^3 = O(M^{-3}). \tag{35}$$

where the second inequality applies the Lipschitz property of $\boldsymbol{\mu}\left(\boldsymbol{x}_{t(\lambda)}^\star, t(\lambda)\right)$ from Assumption 1 (A3).

– Putting (34) and (35) together, we conclude that

$$\|\boldsymbol{\chi}_1\| = O(M^{-3}). \tag{36}$$

• **Controlling $\boldsymbol{\chi}_2$.** From Taylor's theorem and Assumption 1 (A2), we can bound

$$\left\| \boldsymbol{\mu}_\theta\left(\boldsymbol{x}_{t_{i-1}}^{\mathsf{bck}}, t_{i-1}\right) + \boldsymbol{\mu}_\theta\left(\boldsymbol{x}_{t_{i-1}}^{\mathsf{for}}, t_{i-1}\right) - 2\boldsymbol{\mu}_\theta\left(\boldsymbol{x}_{t_{i-1}}^\star, t_{i-1}\right) - \nabla\boldsymbol{\mu}_\theta\left(\boldsymbol{x}_{t_{i-1}}^\star, t_{i-1}\right)\left(\boldsymbol{x}_{t_{i-1}}^{\mathsf{bck}} + \boldsymbol{x}_{t_{i-1}}^{\mathsf{for}} - 2\boldsymbol{x}_{t_{i-1}}^\star\right) \right\|$$

$$\leq H_x \left( \left\| \boldsymbol{x}_{t_{i-1}}^{\mathsf{bck}} - \boldsymbol{x}_{t_{i-1}}^\star \right\|^2 + \left\| \boldsymbol{x}_{t_{i-1}}^{\mathsf{for}} - \boldsymbol{x}_{t_{i-1}}^\star \right\|^2 \right) = O(M^{-2}),$$

where the last step results from Theorem 1 and (28). This allows us to bound $\chi_2$ as

$$\begin{aligned}
\|\chi_2\| &\leq \int_{\lambda_{t_{i-1}}}^{\lambda_{t_i}} e^\lambda \Big( \big\| \nabla \boldsymbol{\mu}_\theta \big( \boldsymbol{x}_{t_{i-1}}^\star, t_{i-1} \big) \big( \boldsymbol{x}_{t_{i-1}}^{\mathsf{bck}} + \boldsymbol{x}_{t_{i-1}}^{\mathsf{for}} - 2\boldsymbol{x}_{t_{i-1}}^\star \big) \big\| + O(M^{-2}) \Big) \, d\lambda \\
&\leq \big( e^{\lambda_{t_i}} - e^{\lambda_{t_{i-1}}} \big) \left( \frac{L_x}{\alpha_{t_{i-1}}} \big\| \boldsymbol{x}_{t_{i-1}}^{\mathsf{bck}} + \boldsymbol{x}_{t_{i-1}}^{\mathsf{for}} - 2\boldsymbol{x}_{t_{i-1}}^\star \big\| + O(M^{-2}) \right) \\
&\leq e^{\lambda_{t_i}} \delta_{t_i} \frac{L_x}{\alpha_{t_{i-1}}} \big\| \boldsymbol{x}_{t_{i-1}}^{\mathsf{bck}} + \boldsymbol{x}_{t_{i-1}}^{\mathsf{for}} - 2\boldsymbol{x}_{t_{i-1}}^\star \big\| + e^{\lambda_{t_i}} \delta_{t_i} O(M^{-2}) \\
&= \frac{L_x \delta_{t_i}}{\sigma_{t_{i-1}}} \big\| \boldsymbol{x}_{t_{i-1}}^{\mathsf{bck}} + \boldsymbol{x}_{t_{i-1}}^{\mathsf{for}} - 2\boldsymbol{x}_{t_{i-1}}^\star \big\| + O(M^{-3}),
\end{aligned} \tag{37}$$

where the second line applies Assumption 1 (A1) and the last step holds as $\lambda_t = \log(\alpha_t/\sigma_t)$ and $\delta_{t_i} = O(M^{-1})$.

- **Controlling $\chi_3$.** By Taylor's theorem and Assumption 1 (A2), we can bound

$$\begin{aligned}
\big\| \boldsymbol{\mu}_\theta \big( &\boldsymbol{x}_{t_i}^{\mathsf{for}}, t_i \big) - \boldsymbol{\mu}_\theta \big( \boldsymbol{x}_{t_i}^\star, t_i \big) - \boldsymbol{\mu}_\theta \big( \boldsymbol{x}_{t_{i-1}}^{\mathsf{for}}, t_{i-1} \big) + \boldsymbol{\mu}_\theta \big( \boldsymbol{x}_{t_{i-1}}^\star, t_{i-1} \big) \big\| \\
&\leq \big\| \nabla \boldsymbol{\mu}_\theta \big( \boldsymbol{x}_{t_i}^\star, t_i \big) \big( \boldsymbol{x}_{t_i}^{\mathsf{for}} - \boldsymbol{x}_{t_i}^\star \big) - \nabla \boldsymbol{\mu}_\theta \big( \boldsymbol{x}_{t_{i-1}}^\star, t_{i-1} \big) \big( \boldsymbol{x}_{t_{i-1}}^{\mathsf{for}} - \boldsymbol{x}_{t_{i-1}}^\star \big) \big\| \\
&\quad + H_x \big\| \boldsymbol{x}_{t_i}^{\mathsf{for}} - \boldsymbol{x}_{t_i}^\star \big\|^2 + H_x \big\| \boldsymbol{x}_{t_{i-1}}^{\mathsf{for}} - \boldsymbol{x}_{t_{i-1}}^\star \big\|^2 \\
&\leq \big\| \big( \nabla \boldsymbol{\mu}_\theta \big( \boldsymbol{x}_{t_i}^\star, t_i \big) - \nabla \boldsymbol{\mu}_\theta \big( \boldsymbol{x}_{t_{i-1}}^\star, t_{i-1} \big) \big) \big( \boldsymbol{x}_{t_i}^{\mathsf{for}} - \boldsymbol{x}_{t_i}^\star \big) \big\| \\
&\quad + \big\| \nabla \boldsymbol{\mu}_\theta \big( \boldsymbol{x}_{t_{i-1}}^\star, t_{i-1} \big) \big\| \big\| \boldsymbol{x}_{t_i}^{\mathsf{for}} - \boldsymbol{x}_{t_i}^\star - \boldsymbol{x}_{t_{i-1}}^{\mathsf{for}} + \boldsymbol{x}_{t_{i-1}}^\star \big\| + O(M^{-2}),
\end{aligned} \tag{38}$$

where the last step applies (28).

To control the first term on the right-hand-side of (38), one knows from Assumption 1 (A5) that

$$\big\| \nabla \boldsymbol{\mu}_\theta \big( \boldsymbol{x}_{t_i}^\star, t_i \big) - \nabla \boldsymbol{\mu}_\theta \big( \boldsymbol{x}_{t_{i-1}}^\star, t_{i-1} \big) \big\| \leq H \big\| \boldsymbol{x}_{t_i}^{\mathsf{for}} - \boldsymbol{x}_{t_i}^\star \big\| = O(M^{-1}). \tag{39}$$

Therefore the first term is bounded by

$$\begin{aligned}
\big\| \big( \nabla \boldsymbol{\mu}_\theta &\big( \boldsymbol{x}_{t_i}^\star, t_i \big) - \nabla \boldsymbol{\mu}_\theta \big( \boldsymbol{x}_{t_{i-1}}^\star, t_{i-1} \big) \big) \big( \boldsymbol{x}_{t_i}^{\mathsf{for}} - \boldsymbol{x}_{t_i}^\star \big) \big\| \\
&\leq \big\| \nabla \boldsymbol{\mu}_\theta \big( \boldsymbol{x}_{t_i}^\star, t_i \big) - \nabla \boldsymbol{\mu}_\theta \big( \boldsymbol{x}_{t_{i-1}}^\star, t_{i-1} \big) \big\| \big\| \boldsymbol{x}_{t_i}^{\mathsf{for}} - \boldsymbol{x}_{t_i}^\star \big\| = O(M^{-2}),
\end{aligned} \tag{40}$$

where the last equation applies (28) again.

In addition, putting collectively what we have shown in (23), (24), and (25), one knows that

$$\big\| \boldsymbol{x}_{t_i}^{\mathsf{for}} - \boldsymbol{x}_{t_i}^\star - \boldsymbol{x}_{t_{i-1}}^{\mathsf{for}} + \boldsymbol{x}_{t_{i-1}}^\star \big\| \leq \left| \frac{\sigma_{t_i}}{\sigma_{t_{i-1}}} - 1 \right| \big\| \boldsymbol{x}_{t_{i-1}}^{\mathsf{for}} - \boldsymbol{x}_{t_{i-1}}^\star \big\| + L_x \delta_{t_i} \big\| \boldsymbol{x}_{t_i}^{\mathsf{for}} - \boldsymbol{x}_{t_i}^\star \big\| + O(M^{-2}).$$

Since $\sigma_{t_i} \leq \sigma_{t_{i-1}}$, we have

$$\left| \frac{\sigma_{t_i}}{\sigma_{t_{i-1}}} - 1 \right| = 1 - \frac{\sigma_{t_i}}{\sigma_{t_{i-1}}} = 1 - \frac{\alpha_{t_i}}{\alpha_{t_{i-1}}} e^{-(\lambda_{t_i} - \lambda_{t_{i-1}})} \leq 1 - e^{-\delta_{t_i}} \leq \delta_{t_i},$$

which together with (27) leads to

$$\big\| \boldsymbol{x}_{t_i}^{\mathsf{for}} - \boldsymbol{x}_{t_i}^\star - \boldsymbol{x}_{t_{i-1}}^{\mathsf{for}} + \boldsymbol{x}_{t_{i-1}}^\star \big\| \leq \delta_{t_i} \big\| \boldsymbol{x}_{t_{i-1}}^{\mathsf{for}} - \boldsymbol{x}_{t_{i-1}}^\star \big\| + L_x \delta_{t_i} \big\| \boldsymbol{x}_{t_i}^{\mathsf{for}} - \boldsymbol{x}_{t_i}^\star \big\| + O(M^{-2}) = O(M^{-2}). \tag{41}$$

Therefore the second term in the right-hand-side of (38) is bounded by

$$\big\| \nabla \boldsymbol{\mu}_\theta \big( \boldsymbol{x}_{t_{i-1}}^\star, t_{i-1} \big) \big\| \big\| \boldsymbol{x}_{t_i}^{\mathsf{for}} - \boldsymbol{x}_{t_i}^\star - \boldsymbol{x}_{t_{i-1}}^{\mathsf{for}} + \boldsymbol{x}_{t_{i-1}}^\star \big\| \leq \frac{L_x}{\alpha_{t_i}} O(M^{-2}) \overset{(a)}{\leq} \frac{L_x}{\alpha_{t_M}} O(M^{-2}) = O(M^{-2}), \tag{42}$$

where (a) uses the assumption that $\alpha_{t_i} \leq \alpha_{t_M}$ for all $i \leq M$.

Finally, plugging the above bound into the definition of $\chi_3$, we conclude that

$$\begin{aligned}
\|\chi_3\| &\leq \int_{\lambda_{t_{i-1}}}^{\lambda_{t_i}} e^\lambda \, d\lambda \cdot \big\| \boldsymbol{\mu}_\theta \big( \boldsymbol{x}_{t_i}^{\mathsf{for}}, t_i \big) - \boldsymbol{\mu}_\theta \big( \boldsymbol{x}_{t_i}^\star, t_i \big) - \boldsymbol{\mu}_\theta \big( \boldsymbol{x}_{t_{i-1}}^{\mathsf{for}}, t_{i-1} \big) + \boldsymbol{\mu}_\theta \big( \boldsymbol{x}_{t_{i-1}}^\star, t_{i-1} \big) \big\| \\
&\leq e^{\lambda_{t_i}} \delta_{t_i} \big\| \boldsymbol{\mu}_\theta \big( \boldsymbol{x}_{t_i}^{\mathsf{for}}, t_i \big) - \boldsymbol{\mu}_\theta \big( \boldsymbol{x}_{t_i}^\star, t_i \big) - \boldsymbol{\mu}_\theta \big( \boldsymbol{x}_{t_{i-1}}^{\mathsf{for}}, t_{i-1} \big) + \boldsymbol{\mu}_\theta \big( \boldsymbol{x}_{t_{i-1}}^\star, t_{i-1} \big) \big\| = O(M^{-3}).
\end{aligned} \tag{43}$$

- **Putting bounds for $\chi_1$, $\chi_2$, and $\chi_3$ together.** Finally, combining the bounds in (36), (37), and (43), and then plugging it into (29), we find that for any $i \geq 1$,

$$\left\| \boldsymbol{x}_{t_i}^{\mathsf{bck}} - \boldsymbol{x}_{t_i}^{\star} + \boldsymbol{x}_{t_i}^{\mathsf{for}} - \boldsymbol{x}_{t_i}^{\star} \right\| \leq \frac{\sigma_{t_i}}{\sigma_{t_{i-1}}} (1 + L_x \delta_{t_i}) \left\| \boldsymbol{x}_{t_{i-1}}^{\mathsf{bck}} + \boldsymbol{x}_{t_{i-1}}^{\mathsf{for}} - 2\boldsymbol{x}_{t_{i-1}}^{\star} \right\| + \sigma_{t_i} O(M^{-3}).$$

Applying this inequality recursively from $i = 1$ to $i = M$, and noting that $\boldsymbol{x}_{t_0}^{\mathsf{bck}} + \boldsymbol{x}_{t_0}^{\mathsf{for}} - 2\boldsymbol{x}_{t_0}^{\star} = 0$, we obtain

$$\begin{aligned}
\left\| \boldsymbol{x}_{t_i}^{\mathsf{bck}} - \boldsymbol{x}_{t_i}^{\star} + \boldsymbol{x}_{t_i}^{\mathsf{for}} - \boldsymbol{x}_{t_i}^{\star} \right\| &\leq \sum_{i=1}^{M} \prod_{j=i+1}^{M} \frac{\sigma_{t_j}}{\sigma_{t_{j-1}}} (1 + L_x \delta_{t_j}) \cdot \sigma_{t_i} O(M^{-3}) \\
&\leq \sum_{i=1}^{M} \prod_{j=i+1}^{M} \frac{\sigma_{t_j}}{\sigma_{t_{j-1}}} \exp(L_x \delta_{t_j}) \cdot \sigma_{t_i} O(M^{-3}) \\
&= \sum_{i=1}^{M} \sigma_{t_M} \exp(L_x(\lambda_{t_M} - \lambda_{t_i})) \cdot O(M^{-3}) \\
&\leq M \sigma_{t_M} \exp(L_x(\lambda_{t_M} - \lambda_{t_0})) \cdot O(M^{-3}) = O(M^{-2}),
\end{aligned}$$

where the second step is true because $1 + x \leq \mathrm{e}^x$ for any $x \in \mathbb{R}$, and the last line holds as $\lambda_{t_M} - \lambda_{t_i} \leq \lambda_{t_M} - \lambda_{t_0}$ for all $0 \leq i \leq M$.

### B.2. Proof of Theorem 4

Comparing the update rules of our sampler (12) and that of the idealized forward-value discretization (14), one can express the difference between their iterates as

$$\boldsymbol{x}_{t_i} - \boldsymbol{x}_{t_i}^{\mathsf{for}} = \frac{\sigma_{t_i}}{\sigma_{t_{i-1}}} \left( \boldsymbol{x}_{t_{i-1}} - \boldsymbol{x}_{t_{i-1}}^{\mathsf{for}} \right) + \sigma_{t_i} \int_{\lambda_{t_{i-1}}}^{\lambda_{t_i}} \mathrm{e}^{\lambda} \, \mathrm{d}\lambda \cdot \left( \boldsymbol{\mu}_\theta(\widehat{\boldsymbol{x}}_{t_i}, t_i) - \boldsymbol{\mu}_\theta(\boldsymbol{x}_{t_i}^{\mathsf{for}}, t_i) \right),$$

for any $i \geq 1$. Leveraging the Lipschitz property of $\boldsymbol{\mu}_\theta(\boldsymbol{x}_{t_i}, t_i)$, we can bound

$$\begin{aligned}
\left\| \boldsymbol{x}_{t_i} - \boldsymbol{x}_{t_i}^{\mathsf{for}} \right\| &\leq \frac{\sigma_{t_i}}{\sigma_{t_{i-1}}} \left\| \boldsymbol{x}_{t_{i-1}} - \boldsymbol{x}_{t_{i-1}}^{\mathsf{for}} \right\| + \sigma_{t_i} \int_{\lambda_{t_{i-1}}}^{\lambda_{t_i}} \mathrm{e}^{\lambda} \, \mathrm{d}\lambda \cdot \left\| \boldsymbol{\mu}_\theta(\boldsymbol{x}_{t_i}^{\mathsf{for}}, t_i) - \boldsymbol{\mu}_\theta(\widehat{\boldsymbol{x}}_{t_i}, t_i) \right\| \\
&\leq \frac{\sigma_{t_i}}{\sigma_{t_{i-1}}} \left\| \boldsymbol{x}_{t_{i-1}} - \boldsymbol{x}_{t_{i-1}}^{\mathsf{for}} \right\| + \sigma_{t_i} \mathrm{e}^{\lambda_{t_i}} \delta_{t_i} \frac{L_x}{\alpha_{t_i}} \left\| \boldsymbol{x}_{t_i}^{\mathsf{for}} - \widehat{\boldsymbol{x}}_{t_i} \right\| \\
&\leq \frac{\sigma_{t_i}}{\sigma_{t_{i-1}}} \left\| \boldsymbol{x}_{t_{i-1}} - \boldsymbol{x}_{t_{i-1}}^{\mathsf{for}} \right\| + \delta_{t_i} L_x \left\| \boldsymbol{x}_{t_i}^{\mathsf{for}} - \boldsymbol{x}_{t_i}^{\star}(\boldsymbol{x}_{t_{i-1}}, t_{i-1}) \right\| \\
&\quad + \delta_{t_i} L_x \left\| \boldsymbol{x}_{t_i}^{\star}(\boldsymbol{x}_{t_{i-1}}, t_{i-1}) - \widehat{\boldsymbol{x}}_{t_i} \right\|,
\end{aligned} \tag{44}$$

where the last equation uses $\sigma_{t_i} \mathrm{e}^{\lambda_{t_i}} / \alpha_{t_i} = \mathrm{e}^{\lambda_{t_i}} \mathrm{e}^{-\lambda_{t_i}} = 1$.

We claim that the second term on the right-hand-side of (44) satisfies the following bound:

$$\left\| \boldsymbol{x}_{t_i}^{\mathsf{for}} - \boldsymbol{x}_{t_i}^{\star}(\boldsymbol{x}_{t_{i-1}}, t_{i-1}) \right\| \leq \frac{\sigma_{t_i}}{\sigma_{t_{i-1}}(1 - \delta_{t_i} L_x)} \left\| \boldsymbol{x}_{t_{i-1}}^{\mathsf{for}} - \boldsymbol{x}_{t_{i-1}} \right\| + \frac{\sigma_{t_i} \mathrm{e}^{\lambda_{t_i}} L_t \delta_{t_i}^2}{2(1 - \delta_{t_i} L_x)}. \tag{45}$$

The proof is deferred to the end of this section.

Suppose (45) holds temporarily. We can plug it into (44) to obtain the following key relationship regarding the difference

between our sampler $\boldsymbol{x}_{t_i}$ and the idealized forward-value discretization $\boldsymbol{x}_{t_i}^{\text{for}}$:

$$
\begin{aligned}
\left\|\boldsymbol{x}_{t_i} - \boldsymbol{x}_{t_i}^{\text{for}}\right\| &\leq \frac{\sigma_{t_i}}{\sigma_{t_{i-1}}}\left\|\boldsymbol{x}_{t_{i-1}} - \boldsymbol{x}_{t_{i-1}}^{\text{for}}\right\| + \delta_{t_i} L_x\left(\frac{\sigma_{t_i}\left\|\boldsymbol{x}_{t_{i-1}}^{\text{for}} - \boldsymbol{x}_{t_{i-1}}\right\|}{\sigma_{t_{i-1}}(1 - \delta_{t_i} L_x)} + \frac{\sigma_{t_i} \mathrm{e}^{\lambda_{t_i}} L_t \delta_{t_i}^2}{2(1 - \delta_{t_i} L_x)}\right) \\
&\quad + \delta_{t_i} L_x \left\|\boldsymbol{x}_{t_i}^{\star}(\boldsymbol{x}_{t_{i-1}}, t_{i-1}) - \widehat{\boldsymbol{x}}_{t_i}\right\| \\
&= \frac{\sigma_{t_i}\left\|\boldsymbol{x}_{t_{i-1}}^{\text{for}} - \boldsymbol{x}_{t_{i-1}}\right\|}{\sigma_{t_{i-1}}(1 - \delta_{t_i} L_x)} + \frac{\sigma_{t_i} \mathrm{e}^{\lambda_{t_i}} L_x L_t \delta_{t_i}^3}{2(1 - \delta_{t_i} L_x)} + \delta_{t_i} L_x \left\|\boldsymbol{x}_{t_i}^{\star}(\boldsymbol{x}_{t_{i-1}}, t_{i-1}) - \widehat{\boldsymbol{x}}_{t_i}\right\| \\
&= \frac{\sigma_{t_i}\left\|\boldsymbol{x}_{t_{i-1}}^{\text{for}} - \boldsymbol{x}_{t_{i-1}}\right\|}{\sigma_{t_{i-1}}(1 - \delta_{t_i} L_x)} + o(M^{-2}),
\end{aligned}
\tag{46}
$$

where the last equation uses the assumption that $\left\|\boldsymbol{x}_{t_i}^{\star}(\boldsymbol{x}_{t_{i-1}}, t_{i-1}) - \widehat{\boldsymbol{x}}_{t_i}\right\| = o(M^{-1})$ and $\delta_{t_i} = O(M^{-1})$. Applying this relationship recursively, we can bound the final error at $t_M$ as

$$
\left\|\boldsymbol{x}_{t_M} - \boldsymbol{x}_{t_M}^{\text{for}}\right\| \leq \sum_{i=1}^{M} \prod_{j=i+1}^{M} \frac{\sigma_{t_j}}{\sigma_{t_{j-1}}(1 - \delta_{t_j} L_x)} \cdot o(M^{-2}).
\tag{47}
$$

To finish up, note that

$$
\begin{aligned}
\prod_{j=i+1}^{M} \frac{\sigma_{t_j}}{\sigma_{t_{j-1}}(1 - \delta_{t_j} L_x)} &\overset{(a)}{\leq} \prod_{j=i+1}^{M} \frac{\sigma_{t_j}}{\sigma_{t_{j-1}}} \exp\left(2\delta_{t_j} L_x\right) = \frac{\sigma_{t_M}}{\sigma_{t_i}} \exp\left(2L_x \sum_{j=i+1}^{M} \delta_{t_j}\right) \\
&= \frac{\sigma_{t_M}}{\sigma_{t_i}} \exp\left(2L_x(\lambda_{t_M} - \lambda_{t_i})\right) \\
&\overset{(b)}{\leq} \exp\left(2L_x(\lambda_{t_M} - \lambda_{t_0})\right),
\end{aligned}
$$

where (a) holds because $1/(1 - x) \leq \exp(2x)$ for $0 \leq x \leq 1/2$ and $\delta_{t_j} L_x = O(M^{-1}) \leq 1/2$; (b) is true because of the assumption $\sigma_{t_i} \geq \sigma_{t_M}$ and $\lambda_{t_M} - \lambda_{t_i} \leq \lambda_{t_M} - \lambda_{t_0}$ for all $0 \leq i \leq M$. Summing over $i$ from 1 to $M$, we reach the advertised result:

$$
\left\|\boldsymbol{x}_{t_i} - \boldsymbol{x}_{t_i}^{\text{for}}\right\| \leq M \exp\left(2L_x(\lambda_{t_M} - \lambda_{t_0})\right) \cdot o(M^{-2}) = o(M^{-1}).
\tag{48}
$$

It remains to prove the claim (45). Towards this, note that by the diffusion ODE (21), we can rewrite $\boldsymbol{x}_{t_i}^{\star}(\boldsymbol{x}_{t_{i-1}}, t_{i-1})$ defined in (17) in terms of the data predictor $\boldsymbol{\mu}_\theta$ as

$$
\boldsymbol{x}_t^{\star}(\boldsymbol{x}_{t_{i-1}}, t_{i-1}) = \frac{\sigma_{t_i}}{\sigma_{t_{i-1}}}\boldsymbol{x}_{t_{i-1}} + \sigma_{t_i} \int_{\lambda_{t_{i-1}}}^{\lambda_{t_i}} \mathrm{e}^{\lambda} \boldsymbol{\mu}_\theta\left(\boldsymbol{x}_{t(\lambda)}^{\star}(\boldsymbol{x}_{t_{i-1}}, t_{i-1}), t(\lambda)\right) \mathrm{d}\lambda.
$$

Combined with the definition of $\boldsymbol{x}_{t_i}^{\text{for}}$ (cf. (14)), the above representation allows us to decompose their difference as

$$
\begin{aligned}
\boldsymbol{x}_{t_i}^{\text{for}} - \boldsymbol{x}_{t_i}^{\star}(\boldsymbol{x}_{t_{i-1}}, t_{i-1}) &= \frac{\sigma_{t_i}}{\sigma_{t_{i-1}}}\left(\boldsymbol{x}_{t_{i-1}}^{\text{for}} - \boldsymbol{x}_{t_{i-1}}\right) + \sigma_{t_i} \int_{\lambda_{t_{i-1}}}^{\lambda_{t_i}} \mathrm{e}^{\lambda}\left(\boldsymbol{\mu}_\theta\left(\boldsymbol{x}_{t_i}^{\text{for}}, t_i\right) - \boldsymbol{\mu}_\theta(\boldsymbol{x}_{t(\lambda)}^{\star}(\boldsymbol{x}_{t_{i-1}}, t_{i-1}), t(\lambda))\right) \mathrm{d}\lambda \\
&= \frac{\sigma_{t_i}}{\sigma_{t_{i-1}}}\left(\boldsymbol{x}_{t_{i-1}}^{\text{for}} - \boldsymbol{x}_{t_{i-1}}\right) + \sigma_{t_i} \int_{\lambda_{t_{i-1}}}^{\lambda_{t_i}} \mathrm{e}^{\lambda} \mathrm{d}\lambda \cdot \left(\boldsymbol{\mu}_\theta\left(\boldsymbol{x}_{t_i}^{\text{for}}, t_i\right) - \boldsymbol{\mu}_\theta(\boldsymbol{x}_{t_i}^{\star}(\boldsymbol{x}_{t_{i-1}}, t_{i-1}), t_i)\right) \\
&\quad + \sigma_{t_i} \int_{\lambda_{t_{i-1}}}^{\lambda_{t_i}} \mathrm{e}^{\lambda}\left(\boldsymbol{\mu}_\theta\left(\boldsymbol{x}_{t_i}^{\star}(\boldsymbol{x}_{t_{i-1}}, t_{i-1}), t_i\right) - \boldsymbol{\mu}_\theta\left(\boldsymbol{x}_{t(\lambda)}^{\star}(\boldsymbol{x}_{t_{i-1}}, t_{i-1}), t(\lambda)\right)\right) \mathrm{d}\lambda.
\end{aligned}
\tag{49}
$$

Let us control the second and third quantities on the right-hand-side of (49) separately. Regarding the second term, we can invoke a similar argument as in (24) to bound it as

$$
\begin{aligned}
\left\|\sigma_{t_i} \int_{\lambda_{t_{i-1}}}^{\lambda_{t_i}} \mathrm{e}^{\lambda} \mathrm{d}\lambda \cdot \left(\boldsymbol{\mu}_\theta\left(\boldsymbol{x}_{t_i}^{\text{for}}, t_i\right) - \boldsymbol{\mu}_\theta(\boldsymbol{x}_{t_i}^{\star}(\boldsymbol{x}_{t_{i-1}}, t_{i-1}), t_i)\right)\right\| &\leq \sigma_{t_i} \mathrm{e}^{\lambda_{t_i}} \delta_{t_i} \frac{L_x}{\alpha_{t_i}}\left\|\boldsymbol{x}_{t_i}^{\text{for}} - \boldsymbol{x}_{t_i}^{\star}(\boldsymbol{x}_{t_{i-1}}, t_{i-1})\right\| \\
&= \delta_{t_i} L_x\left\|\boldsymbol{x}_{t_i}^{\text{for}} - \boldsymbol{x}_{t_i}^{\star}(\boldsymbol{x}_{t_{i-1}}, t_{i-1})\right\|,
\end{aligned}
\tag{50}
$$

where the first inequality uses $\lambda_{t_i} \leq \lambda_{t_{i-1}}$ and the Lipschitz condition of $\boldsymbol{\mu}_\theta$ from Assumption 1 (A1), and the last equation uses $\lambda_{t_i} = \log(\alpha_{t_i}/\sigma_{t_i})$. As for the third quantity, repeating a similar argument for (25), one can bound it as

$$\sigma_{t_i} \left\| \int_{\lambda_{t_{i-1}}}^{\lambda_{t_i}} e^\lambda \Big( \boldsymbol{\mu}_\theta(\boldsymbol{x}_{t_i}^\star(\boldsymbol{x}_{t_{i-1}}, t_{i-1}), t_i) - \boldsymbol{\mu}_\theta(\boldsymbol{x}_{t(\lambda)}^\star(\boldsymbol{x}_{t_{i-1}}, t_{i-1}), t(\lambda)) \Big) \, d\lambda \right\| \leq \frac{1}{2} \sigma_{t_i} e^{\lambda_{t_i}} L_t \delta_{t_i}^2.$$

Plugging these two bounds into (49) and rearranging the inequality, we finish the proof of the claim (45).

## C. Proof of convergence order lower bound (Theorem 2)

Let us consider the case where the target distribution is an isotropic Gaussian distribution $q_0 = \mathcal{N}(0, \gamma I_d)$ for some constant $\gamma \geq 0$.

**Preliminaries.**  Before proceeding to the main proof, we first introduce some preliminary results that will be frequently used in the following analysis.

By the choice of the forward process (1), the distribution $q_t$ of $\boldsymbol{x}_t$ at any time $t \geq 0$ satisfies

$$\boldsymbol{x}_t \sim q_t = \mathcal{N}\big(0, (\alpha_t^2 \gamma^2 + \sigma_t^2)\boldsymbol{I}_d\big),$$

and its score function $\boldsymbol{s}_t^\star(\cdot)$ can be computed in closed form as

$$\boldsymbol{s}_t^\star(\boldsymbol{x}) = \nabla_{\boldsymbol{x}} \log q_t(\boldsymbol{x}) = \nabla_{\boldsymbol{x}} \left( -\frac{\|\boldsymbol{x}\|^2}{\alpha_t^2 \gamma^2 + \sigma_t^2} \right) = -\frac{\boldsymbol{x}}{\alpha_t^2 \gamma^2 + \sigma_t^2}.$$

Combining this with the relationship between the noise predictor and score function $\boldsymbol{\varepsilon}_\theta(\boldsymbol{x}_t, t) = -\sigma_t \boldsymbol{s}_t^\star(\boldsymbol{x}_t)$, we can derive the noise predictor at noise level $t(\lambda)$ for any $\lambda \in \mathbb{R}$ as

$$\boldsymbol{\varepsilon}_\theta\big(\boldsymbol{x}_{t(\lambda)}, t(\lambda)\big) = -\sigma_{t(\lambda)} \boldsymbol{s}_{t(\lambda)}^\star(\boldsymbol{x}_{t(\lambda)}) = \frac{\sigma_{t(\lambda)} \boldsymbol{x}_{t(\lambda)}}{\alpha_{t(\lambda)}^2 \gamma^2 + \sigma_{t(\lambda)}^2} = \frac{e^{-\lambda} \boldsymbol{x}_{t(\lambda)}}{\alpha_{t(\lambda)}(\gamma^2 + e^{-2\lambda})}, \tag{51}$$

where the last line holds due to the fact that $\sigma_{t(\lambda)} = \alpha_{t(\lambda)} e^{-\lambda}$.

We claim that in the isotropic Gaussian case, the exact solution $\boldsymbol{x}_t^\star$ of the diffusion ODE (5) remains colinear with the initial point $\boldsymbol{x}_{t_0}^\star$. Specifically, for any $\lambda \geq 0$, we have

$$\boldsymbol{x}_{t(\lambda)}^\star = \kappa_{t(\lambda)}^\star \boldsymbol{x}_{t_0}^\star \quad \text{with} \quad \kappa_{t(\lambda)}^\star = \frac{\alpha_{t(\lambda)}}{\alpha_{t_0}} \sqrt{\frac{\gamma^2 + e^{-2\lambda}}{\gamma^2 + e^{-2\lambda(t_0)}}}. \tag{52}$$

To verify this, let us substitute the expression (52) into the right-hand-side of the diffusion ODE (5). For arbitrary $t, s \geq 0$, straightforward calculation yields

$$\frac{\alpha_t}{\alpha_s} \boldsymbol{x}_s^\star - \alpha_t \int_{\lambda(s)}^{\lambda(t)} e^{-\lambda} \boldsymbol{\varepsilon}_\theta\big(\boldsymbol{x}_{t(\lambda)}^\star, t(\lambda)\big) \, d\lambda$$

$$\overset{(a)}{=} \frac{\alpha_t}{\alpha_s} \boldsymbol{x}_s^\star - \int_{\lambda(s)}^{\lambda(t)} \frac{e^{-2\lambda}}{\gamma^2 + e^{-2\lambda}} \boldsymbol{x}_{t(\lambda)}^\star \, d\lambda$$

$$\overset{(b)}{=} \boldsymbol{x}_s^\star \left( \frac{\alpha_t}{\alpha_s} - \int_{\lambda(s)}^{\lambda(t)} \frac{e^{-2\lambda} \kappa_{t(\lambda)}^\star}{(\gamma^2 + e^{-2\lambda}) \kappa_s^\star} \, d\lambda \right)$$

$$\overset{(c)}{=} \boldsymbol{x}_s^\star \left( \frac{\alpha_t}{\alpha_s} - \frac{\alpha_t}{\alpha_s \sqrt{\gamma^2 + e^{-2\lambda(s)}}} \int_{\lambda(s)}^{\lambda(t)} \frac{e^{-2\lambda}}{\sqrt{\gamma^2 + e^{-2\lambda}}} \, d\lambda \right)$$

$$= \frac{\alpha_t \boldsymbol{x}_s^\star}{\alpha_s} \left( 1 - \frac{1}{\sqrt{\gamma^2 + e^{-2\lambda(s)}}} \sqrt{\gamma^2 + e^{-2\lambda}} \Big|_{\lambda(t)}^{\lambda(s)} \right)$$

$$= \frac{\alpha_t \boldsymbol{x}_s^\star}{\alpha_s} \sqrt{\frac{\gamma^2 + e^{-2\lambda(t)}}{\gamma^2 + e^{-2\lambda(s)}}} = \frac{\kappa_t^\star}{\kappa_s^\star} \boldsymbol{x}_s^\star = \boldsymbol{x}_t^\star, \tag{53}$$

where (a) applies (51); (b) and (c) uses the expression of $x^\star_{t(\lambda)}$ from (52); and the penultimate line holds because $\frac{\mathrm{d}}{\mathrm{d}\lambda}\sqrt{\gamma^2 + \mathrm{e}^{-2\lambda}} = -\frac{\mathrm{e}^{-2\lambda}}{\sqrt{\gamma^2 + \mathrm{e}^{-2\lambda}}}$. Setting $s = t_0$ establishes the claim.

Moreover, the calculation in (53) implies the following identity regarding $\kappa_{t_i}$ for any $i \in [M]$:

$$\kappa^\star_{t_i} = \frac{\alpha_{t_i}}{\alpha_{t_{i-1}}}\left(1 - \frac{1}{\sqrt{\gamma^2 + \mathrm{e}^{-2\lambda_{t_{i-1}}}}}\int_{\lambda_{t_{i-1}}}^{\lambda_{t_i}}\frac{\mathrm{e}^{-2\lambda}}{\sqrt{\gamma^2 + \mathrm{e}^{-2\lambda}}}\,\mathrm{d}\lambda\right)\kappa^\star_{t_{i-1}}, \tag{54}$$

which will be frequently used in the following proof.

With the above preparation in hand, we shall establish the lower bound for deterministic DDIM (7) and second-order ODE solver (18) separately.

**Lower bound for deterministic DDIM.** In this paragraph, let $x_{t_i}$ denote the iterates produced by the deterministic DDIM sampler.

As shown in (51), the noise predictor $\varepsilon_\theta(x, t)$ is colinear with $x$. Together with the DDIM update rule (7), this implies that the $i$-th iterate $x_{t_i}$ is aligned with the initial point $x_{t_0}$ for all $i \in [M]$. Therefore, we can write

$$x_{t_i} = \kappa_{t_i} x_{t_0}$$

where $\kappa_{t_i} \in \mathbb{R}$ denotes the scalar coefficient.

Substituting $x_{t_i} = \kappa_{t_i} x_{t_0}$ and the expression (51) for $\varepsilon_\theta$ into the DDIM update (7) yields the following recurrence for $\kappa_{t_i}$:

$$\kappa_{t_0} = 1, \quad\text{and}\quad \kappa_{t_i} = \frac{\alpha_{t_i}}{\alpha_{t_{i-1}}}\left(1 - \frac{\mathrm{e}^{-\lambda_{t_{i-1}}}}{\gamma^2 + \mathrm{e}^{-2\lambda_{t_{i-1}}}}\int_{\lambda_{t_{i-1}}}^{\lambda_{t_i}}\mathrm{e}^{-\lambda}\,\mathrm{d}\lambda\right)\kappa_{t_{i-1}}, \quad i \geq 1. \tag{55}$$

Comparing (55) with the corresponding identity for $\kappa^\star_{t_i}$ in (54), and using $\kappa^\star_{t_0} = \kappa_{t_0} = 1$, we obtain that for any $1 \leq i \leq M$,

$$\kappa^\star_{t_i} - \kappa_{t_i} = a_i(\kappa^\star_{t_{i-1}} - \kappa_{t_{i-1}}) + \kappa^\star_{t_{i-1}}\frac{\alpha_{t_i}}{\alpha_{t_{i-1}}\sqrt{\gamma^2 + \mathrm{e}^{-2\lambda_{t_{i-1}}}}}\int_{\lambda_{t_{i-1}}}^{\lambda_{t_i}}\mathrm{e}^{-\lambda}\left(\frac{\mathrm{e}^{-\lambda_{t_{i-1}}}}{\sqrt{\gamma^2 + \mathrm{e}^{-2\lambda_{t_{i-1}}}}} - \frac{\mathrm{e}^{-\lambda}}{\sqrt{\gamma^2 + \mathrm{e}^{-2\lambda}}}\right)\mathrm{d}\lambda$$

$$= a_i(\kappa^\star_{t_{i-1}} - \kappa_{t_{i-1}}) + \frac{\alpha_{t_i}}{\alpha_{t_0}\sqrt{\gamma^2 + \mathrm{e}^{-2\lambda_{t_0}}}}\int_{\lambda_{t_{i-1}}}^{\lambda_{t_i}}\mathrm{e}^{-\lambda}\left(\frac{\mathrm{e}^{-\lambda_{t_{i-1}}}}{\sqrt{\gamma^2 + \mathrm{e}^{-2\lambda_{t_{i-1}}}}} - \frac{\mathrm{e}^{-\lambda}}{\sqrt{\gamma^2 + \mathrm{e}^{-2\lambda}}}\right)\mathrm{d}\lambda, \tag{56}$$

where the coefficient $a_i$ is defined as

$$a_i := \frac{\alpha_{t_i}}{\alpha_{t_{i-1}}}\left(1 - \frac{\mathrm{e}^{-\lambda_{t_{i-1}}}}{\gamma^2 + \mathrm{e}^{-2\lambda_{t_{i-1}}}}\int_{\lambda_{t_{i-1}}}^{\lambda_{t_i}}\mathrm{e}^{-\lambda}\,\mathrm{d}\lambda\right), \quad 1 \leq i \leq M, \tag{57}$$

and the second line follows from the expression of $\kappa^\star_{t_{i-1}}$ from (52).

Notice that for any $\gamma > 0$,

$$\frac{\mathrm{e}^{-\lambda_{t_{i-1}}}}{\sqrt{\gamma^2 + \mathrm{e}^{-2\lambda_{t_{i-1}}}}} - \frac{\mathrm{e}^{-\lambda}}{\sqrt{\gamma^2 + \mathrm{e}^{-2\lambda}}} \geq 0, \quad \forall \lambda \in [\lambda_{t_{i-1}}, \lambda_{t_i}].$$

Plugging this into (56) leads to $\kappa^\star_{t_i} \geq \kappa_{t_i}$ for all $0 \leq i \leq M$. Thus, the difference between the diffusion ODE and the deterministic DDIM trajectories at step $M$ satisfies

$$\left\|x_{t_M} - x^\star_{t_M}\right\| = (\kappa^\star_{t_M} - \kappa_{t_M})\|x_{t_0}\|.$$

Therefore, to establish that the deterministic DDIM has convergence order at most one, it is sufficient to show that $\kappa^\star_{t_M} - \kappa_{t_M} = \Omega(1/M)$.

To this end, note that the integral on the right-hand-side of (56) can be lower bounded by

$$\int_{\lambda_{t_{i-1}}}^{\lambda_{t_i}} e^{-\lambda} \left( \frac{e^{-\lambda_{t_{i-1}}}}{\sqrt{\gamma^2 + e^{-2\lambda_{t_{i-1}}}}} - \frac{e^{-\lambda}}{\sqrt{\gamma^2 + e^{-2\lambda}}} \right) d\lambda$$

$$= \int_{\lambda_{t_{i-1}}}^{\lambda_{t_i}} e^{-\lambda'} \int_{\lambda_{t_{i-1}}}^{\lambda'} -\frac{d}{d\lambda} \frac{e^{-\lambda}}{\sqrt{\gamma^2 + e^{-2\lambda}}} \, d\lambda \, d\lambda'$$

$$= \int_{\lambda_{t_{i-1}}}^{\lambda_{t_i}} \left( e^{-\lambda} - e^{-\lambda_{t_i}} \right) \frac{e^{2\lambda}}{(\gamma^2 e^{2\lambda} + 1)^{\frac{3}{2}}} \, d\lambda$$

$$\overset{(a)}{\geq} \frac{\delta_{t_i}^2 e^{-\lambda_{t_i}} e^{2\lambda_{t_{i-1}}}}{2(\gamma^2 e^{2\lambda_{t_i}} + 1)^{\frac{3}{2}}}$$

$$\overset{(b)}{\geq} \frac{\delta_{t_i}^2 e^{\lambda_{t_i}}}{2(\gamma^2 e^{2\lambda_{t_i}} + 1)^{\frac{3}{2}}}, \tag{58}$$

where (a) holds because $\lambda_{t_i}$ is decreasing in $i$ and thus

$$\int_{\lambda_{t_{i-1}}}^{\lambda_{t_i}} \left( e^{-\lambda} - e^{-\lambda_{t_i}} \right) \frac{e^{2\lambda}}{(\gamma^2 e^{2\lambda} + 1)^{\frac{3}{2}}} \, d\lambda \geq \frac{e^{2\lambda_{t_{i-1}}}}{(\gamma^2 e^{2\lambda_{t_i}} + 1)^{\frac{3}{2}}} \int_{\lambda_{t_{i-1}}}^{\lambda_{t_i}} \left( e^{-\lambda} - e^{-\lambda_{t_i}} \right) \, d\lambda$$

$$= \frac{e^{2\lambda_{t_{i-1}}} e^{-\lambda_{t_i}}}{(\gamma^2 e^{2\lambda_{t_i}} + 1)^{\frac{3}{2}}} \left( e^{\delta_{t_i}} - 1 - \delta_{t_i} \right) \geq \frac{e^{2\lambda_{t_{i-1}}} e^{-\lambda_{t_i}} \delta_{t_i}^2}{2(\gamma^2 e^{2\lambda_{t_i}} + 1)^{\frac{3}{2}}},$$

and (b) is true because $e^{2\lambda_{t_{i-1}} - \lambda_{t_i}} = e^{\lambda_{t_i}} e^{-2\delta_{t_i}} \geq e^{\lambda_{t_i}}/2$ provided that $\delta_{t_i} = O(1/M) \leq \log(2)/2$. Substituting (58) into (56), we obtain

$$\kappa_{t_i}^\star - \kappa_{t_i} \geq a_i(\kappa_{t_{i-1}}^\star - \kappa_{t_{i-1}}) + \frac{1}{\alpha_{t_0} \sqrt{\gamma^2 + e^{-2\lambda_{t_0}}}} \frac{\delta_{t_i}^2 \alpha_{t_i} e^{\lambda_{t_i}}}{2(\gamma^2 e^{2\lambda_{t_i}} + 1)^{\frac{3}{2}}}, \quad \forall 1 \leq i \leq M. \tag{59}$$

Applying (59) recursively allows us to bound the final error as

$$\kappa_{t_M}^\star - \kappa_{t_M} \geq \frac{1}{\alpha_{t_0} \sqrt{\gamma^2 + e^{-2\lambda_{t_0}}}} \sum_{i=1}^{M} \prod_{j=i+1}^{M} a_j \frac{\delta_{t_i}^2 \alpha_{t_i} e^{\lambda_{t_i}}}{2(\gamma^2 e^{2\lambda_{t_i}} + 1)^{\frac{3}{2}}}. \tag{60}$$

To control the right-hand-side of (60), let us introduce a subset of steps:

$$\mathcal{S} := \{0 \leq i \leq M : c_1 \leq e^{\lambda_{t_i}} \leq c_2\}, \tag{61}$$

where $c_1$, $c_2$ are two positive constants. Note that $e^{-\lambda_{t_i}} \leq e^{-\lambda}$ and $\gamma^2 + e^{-2\lambda_{t_i}} \leq (\gamma^2 + e^{-2\lambda}) e^{-2\delta_{t_i}}$ for all $\lambda \in [\lambda_{t_{i-1}}, \lambda_{t_i}]$. This together with the definition of $a_i$ in (57) imply the following lower bound

$$a_i = \frac{\alpha_{t_i}}{\alpha_{t_{i-1}}} \left( 1 - \int_{\lambda_{t_{i-1}}}^{\lambda_{t_i}} \frac{e^{-\lambda_{t_i}}}{\gamma^2 + e^{-2\lambda_{t_i}}} e^{-\lambda} \, d\lambda \right)$$

$$\geq \frac{\alpha_{t_i}}{\alpha_{t_{i-1}}} \left( 1 - e^{2\delta_{t_{i-1}}} \int_{\lambda_{t_{i-1}}}^{\lambda_{t_i}} \frac{e^{-2\lambda}}{\gamma^2 + e^{-2\lambda}} \, d\lambda \right) \geq \frac{\alpha_{t_i}}{\alpha_{t_{i-1}}} \left( 1 - \frac{5}{4} \int_{\lambda_{t_{i-1}}}^{\lambda_{t_i}} \frac{e^{-2\lambda}}{\gamma^2 + e^{-2\lambda}} \, d\lambda \right)$$

provided that $2\delta_{t_i} = O(1/M) \leq \log(5/4)$. This further allows us to lower bound the cumulative product $\prod_{j=i+1}^{M} a_j$ for

any $i \in \mathcal{S}$:

$$
\prod_{j=i+1}^{M} a_j \geq \frac{\alpha_{t_M}}{\alpha_{t_i}} \prod_{j=i+1}^{M} \left( 1 - \frac{5}{4} \int_{\lambda_{t_{j-1}}}^{\lambda_{t_j}} \frac{e^{-2\lambda}}{\gamma^2 + e^{-2\lambda}} \, d\lambda \right)
$$

$$
\geq \frac{\alpha_{t_M}}{\alpha_{t_i}} \left( 1 - \frac{5}{4} \sum_{j=i+1}^{M} \int_{\lambda_{t_{j-1}}}^{\lambda_{t_j}} \frac{e^{-2\lambda}}{\gamma^2 + e^{-2\lambda}} \, d\lambda \right)
$$

$$
\geq \frac{\alpha_{t_M}}{\alpha_{t_i}} \left( 1 - \frac{5}{4} \sum_{e^{\lambda_{t_j}} \geq c_1} \int_{\lambda_{t_{j-1}}}^{\lambda_{t_j}} \frac{e^{-2\lambda}}{\gamma^2 + e^{-2\lambda}} \, d\lambda \right) \overset{(a)}{\geq} \frac{\alpha_{t_M}}{\alpha_{t_i}} \left( 1 - \frac{5}{8} \log(1 + \gamma^{-2} c_1^{-2}) \right) \geq \frac{\alpha_{t_M}}{8\alpha_{t_i}}, \qquad (62)
$$

provided that $\log(1 + \gamma^{-2} c_1^{-2}) \leq 7/5$, where (a) holds because

$$
\sum_{e^{\lambda_{t_j}} \geq c_1} \int_{\lambda_{t_{j-1}}}^{\lambda_{t_j}} \frac{e^{-2\lambda}}{\gamma^2 + e^{-2\lambda}} \, d\lambda \leq \int_{\log c_1}^{\infty} \frac{e^{-2\lambda}}{\gamma^2 + e^{-2\lambda}} \, d\lambda = \frac{1}{2} \log(\gamma^2 + e^{-2\lambda}) \Big|_{\infty}^{\log c_1} = \frac{1}{2} \log(1 + \gamma^{-2} c_1^{-2}).
$$

To finish up, substituting (62) into (60) yields

$$
\kappa_{t_M}^{\star} - \kappa_{t_M} \geq \frac{1}{\alpha_{t_0} \sqrt{\gamma^2 + e^{-2\lambda_{t_0}}}} \sum_{i \in \mathcal{S}} \prod_{j=i+1}^{M} a_j \frac{\delta_{t_i}^2 \alpha_{t_i} e^{\lambda_{t_i}}}{2(\gamma^2 e^{2\lambda_{t_i}} + 1)^{\frac{3}{2}}}
$$

$$
\overset{(a)}{\geq} \frac{\alpha_{t_M}}{\alpha_{t_0} \sqrt{\gamma^2 + e^{-2\lambda_{t_0}}}} \frac{c_1}{16(\gamma^2 c_2^2 + 1)^{\frac{3}{2}}} \sum_{i \in \mathcal{S}} \delta_{t_i}^2
$$

$$
\overset{(b)}{\geq} \frac{\alpha_{t_M}}{\alpha_{t_0} \sqrt{\gamma^2 + e^{-2\lambda_{t_0}}}} \frac{c_1}{16(\gamma^2 c_2^2 + 1)^{\frac{3}{2}}} \frac{(\log c_2 - \log c_1)^2}{M} = \Omega(M^{-1}), \qquad (63)
$$

where (a) holds because $e^{\lambda_{t_i}} / (\gamma^2 e^{2\lambda_{t_i}} + 1)^{3/2} \geq c_1 / (\gamma^2 c_2^2 + 1)^{3/2}$, and (b) is true due to the Cauchy-Schwarz inequality $(\sum_{i \in \mathcal{S}} \delta_{t_i})^2 \leq |\mathcal{S}| \sum_{i \in \mathcal{S}} \delta_{t_i}^2$ and the fact that $\sum_{i \in \mathcal{S}} \delta_{t_i} = \log c_2 - \log c_1$. This completes the proof for the lower bound of deterministic DDIM.

**Lower bound for second-order ODE solver.** In this paragraph, we use $\boldsymbol{x}_{t_i}$ to represent the iterates generated by the second-order ODE solver (18). Similar to the analysis for the deterministic DDIM, we know that $\boldsymbol{\varepsilon}_\theta(\boldsymbol{x}, t)$ is colinear with $\boldsymbol{x}$. In particular, we can express the $i$-th iterate as

$$
\boldsymbol{x}_{t_i} = \kappa_{t_i} \boldsymbol{x}_{t_0},
$$

where the coefficient $\kappa_{t_i}$ satisfies that for any $i \geq 1$,

$$
\kappa_{t_i} = \frac{\alpha_{t_i}}{\alpha_{t_{i-1}}} \kappa_{t_{i-1}} - \alpha_{t_i} \int_{\lambda_{t_{i-1}}}^{\lambda_{t_i}} e^{-\lambda} \, d\lambda \frac{e^{-\lambda_{t_{i-1}}} \kappa_{t_{i-1}}}{\alpha_{t_{i-1}}(\gamma^2 + e^{-2\lambda_{t_{i-1}}})}
$$

$$
- \alpha_{t_i} \int_{\lambda_{t_{i-1}}}^{\lambda_{t_i}} (\lambda - \lambda_{t_{i-1}}) e^{-\lambda} \, d\lambda \frac{\frac{e^{-\lambda_{t_{i-1}}} \kappa_{t_{i-1}}}{\alpha_{t_{i-1}}(\gamma^2 + e^{-2\lambda_{t_{i-1}}})} - \frac{e^{-\lambda_{t_{i-2}}} \kappa_{t_{i-2}}}{\alpha_{t_{i-2}}(\gamma^2 + e^{-2\lambda_{t_{i-2}}})}}{\lambda_{t_{i-1}} - \lambda_{t_{i-2}}}. \qquad (64)
$$

Combining this with the recurrence for $\kappa_{t_i}^{\star}$ given in (54), we can decompose the difference $\kappa_{t_i}^{\star} - \kappa_{t_i}$ as

$$
\kappa_{t_i}^{\star} - \kappa_{t_i} = a_i(\kappa_{t_{i-1}}^{\star} - \kappa_{t_{i-1}}) + b_i + c_i, \qquad (65)
$$

where $a_i$ is define in (57), and $b_i$ and $c_i$ are given by

$$b_i := -\alpha_{t_i} \int_{\lambda_{t_{i-1}}}^{\lambda_{t_i}} (\lambda - \lambda_{t_{i-1}}) e^{-\lambda} \, d\lambda \cdot \frac{1}{\lambda_{t_{i-1}} - \lambda_{t_{i-2}}} \left( \frac{e^{-\lambda_{t_{i-1}}}(\kappa_{t_{i-1}}^\star - \kappa_{t_{i-1}})}{\alpha_{t_{i-1}}(\gamma^2 + e^{-2\lambda_{t_{i-1}}})} - \frac{e^{-\lambda_{t_{i-2}}}(\kappa_{t_{i-2}}^\star - \kappa_{t_{i-2}})}{\alpha_{t_{i-2}}(\gamma^2 + e^{-2\lambda_{t_{i-2}}})} \right), \tag{66}$$

$$c_i := -\alpha_{t_i} \int_{\lambda_{t_{i-1}}}^{\lambda_{t_i}} \left( \frac{e^{-2\lambda}}{\sqrt{\gamma^2 + e^{-2\lambda}}} \frac{\kappa_{t_{i-1}}^\star}{\alpha_{t_{i-1}}\sqrt{\gamma^2 + e^{-2\lambda_{t_{i-1}}}}} - \frac{e^{-\lambda_{t_{i-1}} - \lambda}\kappa_{t_{i-1}}^\star}{\alpha_{t_{i-1}}(\gamma^2 + e^{-2\lambda_{t_{i-1}}})} \right.$$

$$\left. - (\lambda - \lambda_{t_{i-1}})e^{-\lambda} \frac{1}{\lambda_{t_{i-1}} - \lambda_{t_{i-2}}} \left( \frac{e^{-\lambda_{t_{i-1}}}\kappa_{t_{i-1}}^\star}{\alpha_{t_{i-1}}(\gamma^2 + e^{-2\lambda_{t_{i-1}}})} - \frac{e^{-\lambda_{t_{i-2}}}\kappa_{t_{i-2}}^\star}{\alpha_{t_{i-2}}(\gamma^2 + e^{-2\lambda_{t_{i-2}}})} \right) \right) d\lambda. \tag{67}$$

Applying recursion to (65) yields

$$\kappa_{t_M}^\star - \kappa_{t_M} = \sum_{i=1}^{M} \prod_{j=i+1}^{M} a_j(b_i + c_i). \tag{68}$$

In what follows, we shall analyze $c_i$, $b_i$, and $a_i$ separately.

- Controlling $c_i$. For simplicity of notation, define the auxiliary function

$$f(\lambda) := \frac{e^{-\lambda}}{\sqrt{\gamma^2 + e^{-2\lambda}}}.$$

Using the expression of $\kappa_{t_j}$ in (52) , the following factors appearing in the definition of $c_i$ can be rewritten in terms of $f(\cdot)$ for any $i \geq 1$:

$$\frac{e^{-\lambda_{t_j}}\kappa_{t_j}^\star}{\alpha_{t_j}(\gamma^2 + e^{-2\lambda_{t_j}})} = \frac{1}{\alpha_{t_0}\sqrt{\gamma^2 + e^{-2\lambda_{t_0}}}} \frac{e^{-\lambda_{t_j}}}{\sqrt{\gamma^2 + e^{-2\lambda_{t_j}}}} = \frac{f(\lambda_{t_j})}{\alpha_{t_0}\sqrt{\gamma^2 + e^{-2\lambda_{t_0}}}}, \quad j = i, i-1, i-2,$$

$$\frac{e^{-2\lambda}}{\sqrt{\gamma^2 + e^{-2\lambda}}} \frac{\kappa_{t_{i-1}}^\star}{\alpha_{t_{i-1}}\sqrt{\gamma^2 + e^{-2\lambda_{t_{i-1}}}}} = \frac{e^{-\lambda}f(\lambda)}{\alpha_{t_0}\sqrt{\gamma^2 + e^{-2\lambda_{t_0}}}}. \tag{69}$$

Consequently, the term $c_i$ can be expressed as:

$$c_i = -\frac{\alpha_{t_i}}{\alpha_{t_0}\sqrt{\gamma^2 + e^{-2\lambda_{t_0}}}} \int_{\lambda_{t_{i-1}}}^{\lambda_{t_i}} e^{-\lambda}\left( f(\lambda) - f(\lambda_{t_{i-1}}) - (\lambda - \lambda_{t_{i-1}})\frac{f(\lambda_{t_{i-1}}) - f(\lambda_{t_{i-2}})}{\lambda_{t_{i-1}} - \lambda_{t_{i-2}}} \right) d\lambda. \tag{70}$$

The derivative of $f(\lambda)$ can be computed as

$$f'(\lambda) = -\frac{\gamma^2 e^{2\lambda}}{(\gamma^2 e^{2\lambda} + 1)^{\frac{3}{2}}}, \qquad f''(\lambda) = -\frac{\gamma^2 e^{2\lambda}(2 - \gamma^2 e^{2\lambda})}{(\gamma^2 e^{2\lambda} + 1)^{\frac{5}{2}}}. \tag{71}$$

By a second-order Taylor's expansion and the fact that $\delta_{t_{i-1}} = O(M^{-1})$ and $\delta_{t_i} = O(M^{-1})$, we obtain

$$f(\lambda) - f(\lambda_{t_{i-1}}) - (\lambda - \lambda_{t_{i-1}})\frac{f(\lambda_{t_{i-1}}) - f(\lambda_{t_{i-2}})}{\lambda_{t_{i-1}} - \lambda_{t_{i-2}}}$$

$$\overset{(a)}{=} (\lambda - \lambda_{t_{i-1}})\left( f'(\lambda_{t_{i-1}}) - \frac{f(\lambda_{t_{i-1}}) - f(\lambda_{t_{i-2}})}{\lambda_{t_{i-1}} - \lambda_{t_{i-2}}} \right) + \frac{1}{2}f''(\lambda_{t_{i-1}})(\lambda - \lambda_{t_{i-1}})^2 + O(M^{-3})$$

$$\overset{(b)}{=} \frac{1}{2}f''(\lambda_{t_{i-1}})(\lambda_{t_{i-1}} - \lambda_{t_{i-2}})(\lambda - \lambda_{t_{i-1}}) + \frac{1}{2}f''(\lambda_{t_{i-1}})(\lambda - \lambda_{t_{i-1}})^2 + O(M^{-3})$$

$$= \frac{1}{2}f''(\lambda_{t_{i-1}})(\lambda - \lambda_{t_{i-2}})(\lambda - \lambda_{t_{i-1}}) + O(M^{-3}).$$

where (a) holds because $f(\lambda) - f(\lambda_{t_{i-1}}) = f'(\lambda_{t_{i-1}})(\lambda - \lambda_{t_{i-1}}) + \frac{1}{2}f''(\lambda_{t_{i-1}})(\lambda - \lambda_{t_{i-1}})^2 + O(M^{-3})$, and (b) is true because $f(\lambda_{t_{i-1}}) - f(\lambda_{t_{i-2}}) = f'(\lambda_{t_{i-1}})(\lambda_{t_{i-1}} - \lambda_{t_{i-2}}) - \frac{1}{2}f''(\lambda_{t_{i-1}})(\lambda_{t_{i-1}} - \lambda_{t_{i-2}})^2 + O(M^{-3})$. Substituting this into (70) results in the following bound for $c_i$:

$$
\begin{aligned}
c_i &\overset{(a)}{=} -\frac{\alpha_{t_i} f''(\lambda_{t_{i-1}})}{2\alpha_{t_0}\sqrt{\gamma^2 + e^{-2\lambda_{t_0}}}} \int_{\lambda_{t_{i-1}}}^{\lambda_{t_i}} e^{-\lambda}(\lambda - \lambda_{t_{i-2}})(\lambda - \lambda_{t_{i-1}})\, d\lambda + O(M^{-4}) \\
&\overset{(b)}{=} -\frac{\alpha_{t_i} e^{-\lambda_{t_{i-1}}} f''(\lambda_{t_{i-1}})}{2\alpha_{t_0}\sqrt{\gamma^2 + e^{-2\lambda_{t_0}}}} \int_{\lambda_{t_{i-1}}}^{\lambda_{t_i}} (\lambda - \lambda_{t_{i-2}})(\lambda - \lambda_{t_{i-1}})\, d\lambda + O(M^{-4}) \\
&= -\frac{\alpha_{t_i} e^{-\lambda_{t_{i-1}}} f''(\lambda_{t_{i-1}})}{2\alpha_{t_0}\sqrt{\gamma^2 + e^{-2\lambda_{t_0}}}} \frac{1}{6}\delta_{t_i}^2\left(2\delta_{t_i} + 3\delta_{t_{i-1}}\right) + O(M^{-4}) \\
&= -\frac{\alpha_{t_i} e^{-\lambda_{t_{i-1}}} f''(\lambda_{t_{i-1}})}{12\alpha_{t_0}\sqrt{\gamma^2 + e^{-2\lambda_{t_0}}}} \delta_{t_i}^2\left(2\delta_{t_i} + 3\delta_{t_{i-1}}\right) + O(M^{-4}),
\end{aligned}
\tag{72}
$$

where (a) holds because

$$
\left| \frac{\alpha_{t_i}}{\alpha_{t_0}\sqrt{\gamma^2 + e^{-2\lambda_{t_0}}}} \int_{\lambda_{t_{i-1}}}^{\lambda_{t_i}} e^{-\lambda} O(M^{-3})\, d\lambda \right| \leq \frac{\alpha_{t_i} e^{-\lambda_{t_i}}}{\alpha_{t_0}\sqrt{\gamma^2 + e^{-2\lambda_{t_0}}}} \delta_{t_i} \cdot O(M^{-3}) = O(M^{-4}),
$$

and (b) is true because

$$
\begin{aligned}
\left| \int_{\lambda_{t_{i-1}}}^{\lambda_{t_i}} (e^{-\lambda} - e^{-\lambda_{t_i}})(\lambda - \lambda_{t_{i-2}})(\lambda - \lambda_{t_{i-1}})\, d\lambda \right| &\leq (e^{-\lambda_{t_{i-1}}} - e^{-\lambda_{t_i}}) \int_{\lambda_{t_{i-1}}}^{\lambda_{t_i}} (\lambda - \lambda_{t_{i-2}})(\lambda - \lambda_{t_{i-1}})\, d\lambda \\
&\leq e^{-\lambda_{t_{i-1}}} \delta_{t_i} \frac{1}{6}\delta_{t_i}^2\left(2\delta_{t_i} + 3\delta_{t_{i-1}}\right) = O(M^{-4}).
\end{aligned}
$$

- Controlling $b_i$. For simplicity of notation, we introduce a sequence

$$
g_{t_i} := \frac{e^{-\lambda_{t_i}}}{\gamma^2 + e^{-2\lambda_{t_i}}} \frac{\kappa_{t_i}^\star - \kappa_{t_i}}{\alpha_{t_i}}.
$$

By definition, $b_i$ can be expressed in terms of $g_{t_{i-1}}$ and $g_{t_{i-2}}$ as:

$$
b_i = -\alpha_{t_i} \int_{\lambda_{t_{i-1}}}^{\lambda_{t_i}} (\lambda - \lambda_{t_{i-1}}) e^{-\lambda}\, d\lambda \frac{g_{t_{i-1}} - g_{t_{i-2}}}{\lambda_{t_{i-1}} - \lambda_{t_{i-2}}}.
\tag{73}
$$

Therefore it suffices to analyze the term $\frac{g_{t_{i-1}} - g_{t_{i-2}}}{\lambda_{t_{i-1}} - \lambda_{t_{i-2}}}$. Leveraging that $\kappa_{t_i}^\star - \kappa_{t_i} = \alpha_{t_i} \frac{\gamma^2 + e^{-2\lambda_{t_i}}}{e^{-\lambda_{t_i}}} g_{t_i}$, recurrence for $\kappa_{t_i}^\star - \kappa_{t_i}$ in (65) also establishes a recurrence for $g_{t_i}$ as:

$$
g_{t_i} = \frac{\alpha_{t_{i-1}} e^{-\lambda_{t_i}}(\gamma^2 + e^{-2\lambda_{t_{i-1}}})}{\alpha_{t_i}(\gamma^2 + e^{-2\lambda_{t_i}}) e^{-\lambda_{t_{i-1}}}} a_i g_{t_{i-1}} - \alpha_{t_i} \int_{\lambda_{t_{i-1}}}^{\lambda_{t_i}} (\lambda - \lambda_{t_{i-1}}) e^{-\lambda}\, d\lambda \frac{g_{t_{i-1}} - g_{t_{i-2}}}{\lambda_{t_{i-1}} - \lambda_{t_{i-2}}} + c_i.
\tag{74}
$$

We further have

$$
\frac{g_{t_i} - g_{t_{i-1}}}{\lambda_{t_i} - \lambda_{t_{i-1}}} = \left( \frac{\alpha_{t_{i-1}} e^{-\lambda_{t_i}}(\gamma^2 + e^{-2\lambda_{t_{i-1}}})}{\alpha_{t_i}(\gamma^2 + e^{-2\lambda_{t_i}}) e^{-\lambda_{t_{i-1}}}} a_i - 1 \right) \frac{g_{t_{i-1}}}{\delta_{t_i}} - \frac{\alpha_{t_i}}{\delta_{t_i}} \int_{\lambda_{t_{i-1}}}^{\lambda_{t_i}} (\lambda - \lambda_{t_{i-1}}) e^{-\lambda}\, d\lambda \frac{g_{t_{i-1}} - g_{t_{i-2}}}{\lambda_{t_{i-1}} - \lambda_{t_{i-2}}} + \frac{c_i}{\delta_{t_i}}.
\tag{75}
$$

Let us control three quantities on the right-hand-side of (75) separately. Regarding the third term, identity (72) gives

$$
\frac{|c_i|}{\delta_{t_i}} \leq \frac{\alpha_{t_i} e^{-\lambda_{t_{i-1}}} |f''(\lambda_{t_{i-1}})|}{12\alpha_{t_0}\sqrt{\gamma^2 + e^{-2\lambda_{t_0}}}} \delta_{t_i}\left(2\delta_{t_i} + 3\delta_{t_{i-1}}\right) + O(M^{-3}) = O(M^{-2}).
\tag{76}
$$

Regarding the second term, provided that $\delta_{t_i} = O(M^{-1}) \leq \log 2$, the coefficient before $\frac{g_{t_{i-1}} - g_{t_{i-2}}}{\lambda_{t_{i-1}} - \lambda_{t_{i-2}}}$ is bounded by:

$$\frac{\alpha_{t_i}}{\delta_{t_i}} \int_{\lambda_{t_{i-1}}}^{\lambda_{t_i}} (\lambda - \lambda_{t_{i-1}}) e^{-\lambda} \, d\lambda \leq \frac{1}{2} \alpha_{t_i} e^{-\lambda_{t_{i-1}}} \delta_{t_i} = \frac{1}{2} \sigma_{t_i} e^{\delta_{t_i}} \delta_{t_i} \leq \sigma_{t_i} \delta_{t_i}. \tag{77}$$

Moreover, by the definition of $a_i$ in (57), the first term is simplified as

$$\frac{1}{\delta_{t_i}} \left( \frac{\alpha_{t_{i-1}} e^{-\lambda_{t_i}} (\gamma^2 + e^{-2\lambda_{t_{i-1}}})}{\alpha_{t_i} (\gamma^2 + e^{-2\lambda_{t_i}}) e^{-\lambda_{t_{i-1}}}} a_i - 1 \right)$$

$$= \frac{1}{\delta_{t_i}} \left( \frac{e^{-\lambda_{t_i}} (\gamma^2 + e^{-2\lambda_{t_{i-1}}})}{(\gamma^2 + e^{-2\lambda_{t_i}}) e^{-\lambda_{t_{i-1}}}} - \frac{e^{-\lambda_{t_i}}}{\gamma^2 + e^{-2\lambda_{t_i}}} \int_{\lambda_{t_{i-1}}}^{\lambda_{t_i}} e^{-\lambda} \, d\lambda - 1 \right)$$

$$= \frac{1}{\delta_{t_i}} \left( \frac{e^{-\lambda_{t_{i-1}}} (1 - e^{-\delta_{t_i}})(e^{-\lambda_{t_{i-1}} - \lambda_{t_i}} - \gamma^2)}{(\gamma^2 + e^{-2\lambda_{t_i}}) e^{-\lambda_{t_{i-1}}}} - \frac{e^{-\lambda_{t_i}}}{\gamma^2 + e^{-2\lambda_{t_i}}} \int_{\lambda_{t_{i-1}}}^{\lambda_{t_i}} e^{-\lambda} \, d\lambda \right)$$

$$= -\frac{1}{\delta_{t_i}} \frac{\gamma^2 (1 - e^{-\delta_{t_i}})}{\gamma^2 + e^{-2\lambda_{t_i}}}, \tag{78}$$

where the second line holds because

$$\frac{e^{-\lambda_{t_i}} (\gamma^2 + e^{-2\lambda_{t_{i-1}}})}{(\gamma^2 + e^{-2\lambda_{t_i}}) e^{-\lambda_{t_{i-1}}}} - 1 = \frac{e^{-\lambda_{t_i}} (\gamma^2 + e^{-2\lambda_{t_{i-1}}}) - (\gamma^2 + e^{-2\lambda_{t_i}}) e^{-\lambda_{t_{i-1}}}}{(\gamma^2 + e^{-2\lambda_{t_i}}) e^{-\lambda_{t_{i-1}}}}$$

$$= \frac{\gamma^2 (e^{-\lambda_{t_i}} - e^{-\lambda_{t_{i-1}}}) + e^{-\lambda_{t_i}} e^{-\lambda_{t_{i-1}}} (e^{-\lambda_{t_{i-1}}} - e^{-\lambda_{t_i}})}{(\gamma^2 + e^{-2\lambda_{t_i}}) e^{-\lambda_{t_{i-1}}}}$$

$$= \frac{(e^{-\lambda_{t_{i-1}}} - e^{-\lambda_{t_i}})(e^{-\lambda_{t_i}} e^{-\lambda_{t_{i-1}}} - \gamma^2)}{(\gamma^2 + e^{-2\lambda_{t_i}}) e^{-\lambda_{t_{i-1}}}}$$

$$= \frac{e^{-\lambda_{t_{i-1}}} (1 - e^{-\delta_{t_i}})(e^{-\lambda_{t_i}} e^{-\lambda_{t_{i-1}}} - \gamma^2)}{(\gamma^2 + e^{-2\lambda_{t_i}}) e^{-\lambda_{t_{i-1}}}}, \tag{79}$$

and the last identity holds since

$$\frac{e^{-\lambda_{t_{i-1}}} (1 - e^{-\delta_{t_i}})(e^{-\lambda_{t_{i-1}} - \lambda_{t_i}} - \gamma^2)}{(\gamma^2 + e^{-2\lambda_{t_i}}) e^{-\lambda_{t_{i-1}}}} - \frac{e^{-\lambda_{t_i}}}{\gamma^2 + e^{-2\lambda_{t_i}}} \int_{\lambda_{t_{i-1}}}^{\lambda_{t_i}} e^{-\lambda} \, d\lambda$$

$$= \frac{(1 - e^{-\delta_{t_i}})(e^{-\lambda_{t_{i-1}} - \lambda_{t_i}} - \gamma^2)}{\gamma^2 + e^{-2\lambda_{t_i}}} - \frac{e^{-\lambda_{t_i}}}{\gamma^2 + e^{-2\lambda_{t_i}}} (e^{-\lambda_{t_{i-1}}} - e^{-\lambda_{t_i}})$$

$$= \frac{-\gamma^2 (1 - e^{-\delta_{t_i}})}{\gamma^2 + e^{-2\lambda_{t_i}}} + \frac{e^{-\lambda_{t_{i-1}} - \lambda_{t_i}} - e^{-2\lambda_{t_i}} - e^{-\lambda_{t_i}} (e^{-\lambda_{t_{i-1}}} - e^{-\lambda_{t_i}})}{\gamma^2 + e^{-2\lambda_{t_i}}}$$

$$= \frac{-\gamma^2 (1 - e^{-\delta_{t_i}})}{\gamma^2 + e^{-2\lambda_{t_i}}}.$$

Equation (78) can be further simplified by leveraging the fact that $1 - e^{-\delta_{t_i}} \leq \delta_{t_i}$:

$$\left| \frac{1}{\delta_{t_i}} \left( \frac{\alpha_{t_{i-1}} e^{-\lambda_{t_i}} (\gamma^2 + e^{-2\lambda_{t_{i-1}}})}{\alpha_{t_i} (\gamma^2 + e^{-2\lambda_{t_i}}) e^{-\lambda_{t_{i-1}}}} a_i - 1 \right) \right| \leq \frac{\gamma^2}{\gamma^2 + e^{-2\lambda_{t_i}}} \leq 1. \tag{80}$$

Finally, plugging the above bounds (77), (76), and (78) into (75), we find that the absolute value of $\frac{g_{t_i} - g_{t_{i-1}}}{\lambda_{t_i} - \lambda_{t_{i-1}}}$ is bounded by

$$\left| \frac{g_{t_i} - g_{t_{i-1}}}{\lambda_{t_i} - \lambda_{t_{i-1}}} \right| \leq \delta_{t_i} \sigma_{t_i} \left| \frac{g_{t_{i-1}} - g_{t_{i-2}}}{\lambda_{t_{i-1}} - \lambda_{t_{i-2}}} \right| + g_{t_{i-1}} + O(M^{-2}).$$

According to Theorem 1, we have $g_{t_{i-1}} = O(M^{-2})$. Hence, for $\sigma_{t_i}\delta_{t_i} = (M^{-1}) \leq 1/2$, we have

$$\left|\frac{g_{t_i} - g_{t_{i-1}}}{\lambda_{t_i} - \lambda_{t_{i-1}}}\right| \leq 2 \max_{0 \leq i \leq M} |g_{t_i}| + O(M^{-2}) = O(M^{-2}). \tag{81}$$

Substituting (81) into (73), we bound the term $|b_i|$ by

$$\begin{aligned}
|b_i| &\leq \alpha_{t_i} \int_{\lambda_{t_{i-1}}}^{\lambda_{t_i}} (\lambda - \lambda_{t_{i-1}}) e^{-\lambda} \, d\lambda \left|\frac{g_{t_{i-1}} - g_{t_{i-2}}}{\lambda_{t_{i-1}} - \lambda_{t_{i-2}}}\right| \\
&\leq \frac{1}{2} \alpha_{t_i} e^{-\lambda_{t_{i-1}}} \delta_{t_i}^2 \left|\frac{g_{t_{i-1}} - g_{t_{i-2}}}{\lambda_{t_{i-1}} - \lambda_{t_{i-2}}}\right| = O\left(M^{-4}\right).
\end{aligned} \tag{82}$$

- Controlling $a_i$. For simplicity of notation, let us denote

$$\tilde{\delta}_i := \frac{e^{-\lambda_{t_{i-1}}}}{\gamma^2 + e^{-2\lambda_{t_{i-1}}}} \int_{\lambda_{t_{i-1}}}^{\lambda_{t_i}} e^{-\lambda} \, d\lambda.$$

By the definition of $a_i$ in (57), we have

$$\prod_{j=i+1}^{M} a_j = \frac{\alpha_{t_M}}{\alpha_{t_i}} \prod_{j=i+1}^{M} (1 - \tilde{\delta}_j) \stackrel{(a)}{=} \frac{\alpha_{t_M}}{\alpha_{t_i}} \exp\left(-\sum_{j=i+1}^{M} \tilde{\delta}_j + O\left(\sum_{j=i+1}^{M} \tilde{\delta}_j^2\right)\right), \tag{83}$$

where (a) holds because

$$\log(1 - \delta_{t_j}) = -\delta_{t_j} + O(\delta_{t_j}^2), \tag{84}$$

We can compute the sum of $\tilde{\delta}_j$ and $\tilde{\delta}_j^2$ as

$$\begin{aligned}
\sum_{j=i+1}^{M} \tilde{\delta}_j &= \sum_{j=i+1}^{M} \frac{e^{-\lambda_{t_{j-1}}}}{\gamma^2 + e^{-2\lambda_{t_{j-1}}}} \int_{\lambda_{t_{j-1}}}^{\lambda_{t_j}} e^{-\lambda} \, d\lambda \stackrel{(a)}{=} \sum_{j=i+1}^{M} \int_{\lambda_{t_{j-1}}}^{\lambda_{t_j}} \frac{e^{-2\lambda}}{\gamma^2 + e^{-2\lambda}} \, d\lambda + O(M^{-1}) \\
&= \int_{\lambda_{t_i}}^{\lambda_{t_M}} \frac{e^{-2\lambda}}{\gamma^2 + e^{-2\lambda}} \, d\lambda + O(M^{-1}) \\
&\stackrel{(b)}{=} \frac{1}{2} \log\left(\frac{\gamma^2 + e^{-2\lambda_{t_i}}}{\gamma^2 + e^{-2\lambda_{t_M}}}\right) + O(M^{-1}),
\end{aligned} \tag{85}$$

where (a) holds because

$$\begin{aligned}
\int_{\lambda_{t_{j-1}}}^{\lambda_{t_j}} \left|\frac{e^{-\lambda_{t_{j-1}}}}{\gamma^2 + e^{-2\lambda_{t_{j-1}}}} - \frac{e^{-\lambda}}{\gamma^2 + e^{-2\lambda}}\right| e^{-\lambda} \, d\lambda &\leq e^{-\lambda_{t_{i-1}}} \max_{\lambda \in [\lambda_{t_{i-1}}, \lambda_{t_i}]} \left|\frac{d}{d\lambda} \frac{e^{-\lambda}}{\gamma^2 + e^{-2\lambda}}\right| \int_{\lambda_{t_{j-1}}}^{\lambda_{t_j}} (\lambda - \lambda_{t_{i-1}}) \, d\lambda \\
&= \frac{\delta_{t_i}^2}{2} e^{-\lambda_{t_{i-1}}} \max_{\lambda \in [\lambda_{t_{i-1}}, \lambda_{t_i}]} \frac{e^{-\lambda}|\gamma^2 - e^{-2\lambda}|}{(\gamma^2 + e^{-2\lambda})^2} \leq \frac{e^{\delta_{t_i}}}{2} \delta_{t_i}^2 = O(M^{-2}),
\end{aligned}$$

and (b) arises from

$$\int_{\lambda_{t_i}}^{\lambda_{t_M}} \frac{e^{-2\lambda}}{\gamma^2 + e^{-2\lambda}} \, d\lambda = \frac{1}{2} \log(\gamma^2 + e^{-2\lambda}) \Big|_{\lambda_{t_M}}^{\lambda_{t_i}} = \frac{1}{2} \log\left(\frac{\gamma^2 + e^{-2\lambda_{t_i}}}{\gamma^2 + e^{-2\lambda_{t_M}}}\right).$$

In addition, the sum of $\tilde{\delta}_j^2$ is computed as

$$\sum_{j=i+1}^{M} \tilde{\delta}_j^2 \leq \sum_{i=1}^{M} \frac{e^{-2\lambda_{t_{i-1}}}}{(\gamma^2 + e^{-2\lambda_{t_{i-1}}})^2} \left(\int_{\lambda_{t_{i-1}}}^{\lambda_{t_i}} e^{-\lambda} \, d\lambda\right)^2 \leq \sum_{i=1}^{M} \frac{\delta_{t_i}^2 e^{-4\lambda_{t_{i-1}}}}{(\gamma^2 + e^{-2\lambda_{t_{i-1}}})^2} \leq \sum_{i=1}^{M} \delta_{t_i}^2 = O(M^{-1}). \tag{86}$$

Substituting (85) and (86) into (83), we have

$$\prod_{j=i+1}^{M} a_j = \frac{\alpha_{t_M}}{\alpha_{t_i}} \sqrt{\frac{\gamma^2 + e^{-2\lambda_{t_M}}}{\gamma^2 + e^{-2\lambda_{t_i}}}} \left(1 + O(M^{-1})\right). \tag{87}$$

Finally, plugging the above bounds (87), (82), and (72) into (68), the discretization error at the last step is given by

$$\kappa_{t_M}^{\star} - \kappa_{t_M} = -\frac{\alpha_{t_M}\sqrt{\gamma^2 + e^{-2\lambda_{t_M}}}}{12\alpha_{t_0}\sqrt{\gamma^2 + e^{-2\lambda_{t_0}}}} \sum_{i=1}^{M} \frac{e^{-\lambda_{t_{i-1}}} f''(\lambda_{t_{i-1}})}{\sqrt{\gamma^2 + e^{-2\lambda_{t_i}}}} \delta_{t_i}^2 \big(2\delta_{t_i} + 3\delta_{t_{i-1}}\big) + O(M^{-3})$$

$$= \frac{\alpha_{t_M}\sqrt{\gamma^2 + e^{-2\lambda_{t_M}}}}{12\alpha_{t_0}\sqrt{\gamma^2 + e^{-2\lambda_{t_0}}}} \sum_{i=1}^{M} \frac{e^{-\lambda_{t_{i-1}}}}{\sqrt{\gamma^2 + e^{-2\lambda_{t_i}}}} \frac{\gamma^2 e^{2\lambda_{t_{i-1}}}(2 - \gamma^2 e^{2\lambda_{t_{i-1}}})}{(\gamma^2 e^{2\lambda_{t_{i-1}}} + 1)^{\frac{5}{2}}} \delta_{t_i}^2 \big(2\delta_{t_i} + 3\delta_{t_{i-1}}\big) + O(M^{-3})$$

$$= \frac{\alpha_{t_M}\sqrt{\gamma^2 + e^{-2\lambda_{t_M}}}}{12\alpha_{t_0}\sqrt{\gamma^2 + e^{-2\lambda_{t_0}}}} \sum_{i=1}^{M} \frac{\gamma^2 e^{2\lambda_{t_{i-1}}}(2 - \gamma^2 e^{2\lambda_{t_{i-1}}})}{(\gamma^2 e^{2\lambda_{t_{i-1}}} + 1)^3} \delta_{t_i}^2 \big(2\delta_{t_i} + 3\delta_{t_{i-1}}\big) + O(M^{-3}),$$

where the second line inserts the second-order derivative of $f$ from (71). Taking $\gamma = e^{-\lambda_{t_M}}$, we have $2 - \gamma^2 e^{2\lambda_{t_{i-1}}} \geq 0$ for all $0 \leq i - 1 \leq M$. Therefore, we arrive at the desired result:

$$\kappa_{t_M}^{\star} - \kappa_{t_M} \geq \frac{\alpha_{t_M}\sqrt{\gamma^2 + e^{-2\lambda_{t_M}}}}{12\alpha_{t_0}\sqrt{\gamma^2 + e^{-2\lambda_{t_0}}}} \sum_{i \in \mathcal{S}} \frac{\gamma^2 e^{2\lambda_{t_{i-1}}}(2 - \gamma^2 e^{2\lambda_{t_{i-1}}})}{(\gamma^2 e^{2\lambda_{t_{i-1}}} + 1)^3} \delta_{t_i}^2 \bigg(2\delta_{t_i} + 3\delta_{t_{i-1}}\bigg) + O(M^{-3})$$

$$\geq \frac{2\alpha_{t_M}\sqrt{\gamma^2 + e^{-2\lambda_{t_M}}}\gamma^2 c_1^2 (2 - \gamma^2 c_2^2)}{12\alpha_{t_0}\sqrt{\gamma^2 + e^{-2\lambda_{t_0}}}(\gamma^2 c_2^2 + 1)^3} \sum_{i \in \mathcal{S}} \delta_{t_i}^3 + O(M^{-3})$$

$$\geq \frac{2\alpha_{t_M}\sqrt{\gamma^2 + e^{-2\lambda_{t_M}}}\gamma^2 c_1^2 (2 - \gamma^2 c_2^2)}{12\alpha_{t_0}\sqrt{\gamma^2 + e^{-2\lambda_{t_0}}}(\gamma^2 c_2^2 + 1)^3} \frac{(\log c_2 - \log c_1)^3}{M^2} + O(M^{-3}) = \Omega(M^{-2}),$$

where $\mathcal{S}$ is defined in (61), and the last inequality uses Holder inequality that

$$\log c_2 - \log c_1 = \sum_{i \in \mathcal{S}} \delta_{t_i} \leq \bigg(\sum_{i \in \mathcal{S}} \delta_{t_i}^3\bigg)^{\frac{1}{3}} \bigg(\sum_{i \in \mathcal{S}} 1\bigg)^{\frac{2}{3}} \leq \bigg(\sum_{i \in \mathcal{S}} \delta_{t_i}^3\bigg)^{\frac{1}{3}} M^{\frac{2}{3}}.$$

This finishes the proof.

