# OpenReview forum: "Are First-Order Diffusion Samplers Really Slower? A Fast Forward-Value Approach"
_ICML.cc/2026/Conference — ICML 2026 regular_

### Official Review · Reviewer_JagV · 2026-03-07

**Soundness:** 2
**Presentation:** 2
**Significance:** 3
**Originality:** 2
**Overall Recommendation:** 4
**Confidence:** 3

**Summary:**

The paper presents a novel first-order solver for diffusion ODEs. The authors' key claim is that the approximation error of the solver and the resulting quality of generation rely **not** only on the solver's order, but also on the points, in which the velocity field is evaluated. To verify this claim, the authors build a solver that consists of two steps: 1) calculate a "lookahead" approximation of the next point; 2) make a step using it for evaluation of the velocity field instead of the current point. Experimental results suggest better performance of the solver compared to DDIM and the higher-order DPMSolver(2, 3) and UniPC.

**Compliance With Llm Reviewing Policy:**

Affirmed.

**Final Justification:**

The paper proposes a new family of solvers that perform the next step prediction via constructing a cheap lookahead estimate and evaluating the velocity at the new point instead of the current point. While keeping the number of diffusion model evaluations equal to the number of steps, the method significantly improves generative quality over base solver (e.g. DDIM) and produces comparable or superior quality to other competitive solvers (e.g. IPNDM, UniPC). This illustrates the authors' idea that the order of the solver is not a unique property that defines the resulting generative qualities, which is a promising idea for constructing new base solvers that do not require additional training.

While it would be great to see a more thorough evaluation on newer/bigger models, including Flow Matching-based ODEs, the authors have addressed my major concerns during the rebuttal period. They clarified that the method's NFE is equal to the number of steps, added/acknowledged comparisons with UniPC/iPNDM, and provided evaluation on Stable Diffusion. Thus, I raise my score to "Weak accept".

**Key Questions For Authors:**

* Though I am not an expert in the classical numerical methods, the proposed idea looks quite elegant and similar to some ideas in the basic ODE solvers. Computing lookahead points are the base of Runge-Kutta methods. The difference is the authors' method (seemingly) suggests computing the lookahead point without additional function evaluations and moving along the direction, predicted in the lookahead point, instead of moving along the weighted average of directions. Are there any classical numerical methods that try to eliminate this extra NFE while still making the scheme closer to "implicit" instead of "explicit"? Discussing the connection to the classical numerical methods that share the similar ideas would be a great addition.

* The resulting method could greatly benefit from a more direct comparison with higher-order methods. If I understand correctly and the lookahead step uses the previously calculated velocities, the proposed update step has the same signature as the linear multi-step methods: the next point is the linear combination of the current point with the previously calculated velocities. How would the coefficients of this linear combination compare with the coeffcients in the higher-order methods, e.g. DPMSolver-2?

**Limitations:**

yes

**Strengths And Weaknesses:**

## Significance / Originality

### Strengths

 * Designing reliable and high-quality base solvers that could potentially replace well-established methods as DDIM or DPM-Solver remains a relevant task, since they are still used even in large text-to-image/text-to-video models, and even the advanced solver learning methods rely on this fundament;
 * The paper proposes an interesting perspective on improving efficiency of the solver: changing the evaluation point could be at least as beneficial as improving its order. This opens opportunities for the further design of high-quality lower-order solvers.

### Weaknesses

* The paper would greatly benefit from including more direct comparisons of the derived method with the baselines (e.g. formula by formula, please see questions for the details) and discussing connections with the classical numerical methods.
* The experimental evaluation is quite limited. The method does not require training, thus I would suggest to include larger-scale text-to-image models for a more practically relevant setup.

## Soundness / presentation

### Strengths

* The writing is clear, except one key detail, mentioned in weaknesses;
* If the empirical results presented by the authors are obtained under a fair comparison (see details in weaknesses and questions), then the derived method offers a solid alternative for the well-established base solvers, obtaining higher quality under the fixed budget;
* The empirical claims are supported by theory.

### Weaknesses

* There is one key detail that is not explicitly written anywhere in the paper: how is the lookahead sample produced? Different phrases suggest different interpretations:
	1) Algorithm 1: "Construct a lookahead estimate, using only information up to step $i − 1$ (e.g., one-step DDIM from $(x_{t_{i - 1}} , t_{i−1})$";
	2) In the experiments, the algorithm is called F-DDIM, which suggests the authors use DDIM to produce the lookahead sample;
	3) NFE and the number of discretization steps are considered synonyms;

	The first phrase itself contains a contradiction: if one interprets "information up to step $i - 1$" as a synonym to "without additional neural network evaluations", then one *cannot* perform DDIM step, since it involves calculating $\varepsilon(x_{t_{i - 1}}, t_{i - 1})$. If the authors perform DDIM step for the lookahead, then the NFE multiplies by 2, which means the comparisons are incorrect due to the drastically different resource constraints between F-DDIM and the baselines.

	The attached implementation suggests (please tell if I am wrong) the authors do not use additional neural network evaluations during the lookahead step. If so, this procedure should not be called DDIM and should be specified by writing the exact update formula. If this is not a straightforward DDIM step, but is called DDIM, then what is it: DDIM update but with reusing $\varepsilon_{i - 2}$ instead of calculating $\varepsilon_{i - 1}$? This detail needs to be clarified and specified in the paper to avoid such confusion.
* Even under the assumption that F-DDIM does not use additional neural net evaluation during the lookahead step, the results seem quite unreliable for me.
	1) The authors change the design choice of the baselines: `'We ignore certain implementation-specific tricks, such as reducing the solver order in the final iterations"`. This detail makes the comparison unfair: if the lower solver order in the last step is a part of the method's design, then removing it means making a baseline (one of the strongest) strictly worse. If this replacement was made e.g. for the purpose of comparing robustness to hyperparameters, such replacement would be justified by the experimental setup. However, in the case of directly comparing performance of the methods, the strongest variants should be chosen, if this does not change the setup. I would suggest the authors to perform comparison with the original UniPC and the other baselines, if they are changed;
	 2) The paper misses the important baseline, iPNDM [1]. In the LD3 [2] paper, the authors claim that iPNDM achieves FID equal to [11.93, 6.38, 5.08] on NFEs [4, 5, 6] on LSUN-bedroom. At the same time, F-DDIM achieves [12.18, 6.98, 5.57] on NFEs [4, 5, 6], which is strictly worse. Though the difference is not that big, including comparison with iPNDM, where F-DDIM could sometimes perform worse, would make the experiments more fair and ensure the baselines are competitive enough.
* Not particularly a weakness, but a suggestion: due to the recent rise of popularity of the Flow Matching [3] parameterization (e.g. SD3 [4], FLUX, video models), it would also be very beneficial to evaluate the proposed solver on some Flow Matching model (not necessarily large scale), since the prominent solver here is order-1 Euler (equal to DDIM in this case). Since F-DDIM seems to significantly outperform DDIM, it would be interesting to see, whether it holds for Flow Matching and could be a valid replacement.

[1] Fast sampling of diffusion models with exponential integrator

[2] Learning to Discretize Denoising Diffusion ODEs

[3] Flow Matching for Generative Modeling

[4] Scaling Rectified Flow Transformers for High-Resolution Image Synthesis

---

> ### Author Rebuttal · Authors · 2026-03-31
>
> **Calculation of one-step lookahead and NFE**
>
> We would like to clarify that the lookahead step in F-DDIM does not require an additional model evaluation, so the NFE does not double.
>
> More precisely, NFE and the number of discretization steps are synonymous, and **NFE is not multiplied by 2 relative to the number of discretization steps**. This is because we **reuse** the mu predictor calculated in the previous iteration, $\mu(\hat{x}_ {t_ {i-1}},t_ {i-1})$, to construct the eps predictor used in DDIM, i.e., $\epsilon_ {t_ {i-1}}:=(\hat{x}_ {t_ {i-1}}-\alpha_ {t_ {i-1}}\mu(\hat{x}_ {t_ {i-1}},t_ {i-1}))/\sigma_{t_ {i-1}}$.
>
> In the original version, we presented Algorithm 1 in a general form to cover all cases, including F-DDIM and F-DPMSolver2. To make it clearer, we will add the formula of one-step look-ahead predictor for F-DDIM: $\hat{x}_ {t_i} = \alpha_ {t_i}/\alpha_ {t_ {i-1}}+(\alpha_ {t_i}-\frac{\sigma_ {t_ i}}{\sigma_ {t_ {i-1}}}\alpha_ {t_{i-1}}) \epsilon_ {t_ {i-1}}$. In addition, we will also present the formula of one-step look-ahead predictor for F-DPMSolver in the appendix:
> $$
> \hat{x}_ {t_i} = \alpha_ {t_i}/\alpha_ {t_ {i-1}} + \alpha_ {t_i}[ ({\rm e}^{-\lambda_ {t_i}} - {\rm e}^{-\lambda_ {t_{i-1}}}) \epsilon_ {t_ {i-1}} + \phi_1D_1],
> $$
> where $D_1 = \frac{\epsilon_ {t_ {i=1}}-\epsilon_ {t_ {i-2}}}{\lambda_ {t_{i-1}}-\lambda_ {t_ {i-2}}}$, $\phi_1$ is given by (18b), and $\epsilon_ {t_ {i-1}} := (\hat{x}_ {t_ {i-1}} - \alpha_ {t_ {i-1}}\mu(\hat{x}_ {t_ {i-1}},t_ {i-1})) / \sigma_ {t_ {i-1}}$, $\epsilon_ {t_ {i-2}} := (\hat{x}_ {t_ {i-2}} - \alpha_ {t_ {i-2}}\mu(\hat{x}_ {t_ {i-2}},t_ {i-2})) / \sigma_ {t_ {i-2}}$.
>
> **Additional experiments**
>
> We would like to first clarify the goal of our work. Due to space limits, see response to the third reviewer (Reviewer hyjX) on "Position" for details.
>
> * **Implementation of UniPC**
> For UniPC, we have incorporated the implementation-specific trick of reducing the solver order in the final iterations.
> This revision does affect the numerical results, but it does not change the overall conclusion. For example, on ImageNet64 with EDM2-S, the results change to 50.43, 25.88, 13.88, 5.07, 2.79 at NFE = 4,5,6,8,10 (Table 3).
>
> * **Experiments on larger-scale text-to-image models**
> In prior work, experiments are primarily conducted on class-conditional or unconditional generation tasks, and quantitative evaluation for text-to-image models is less standard. For example, UniPC does not report quantitative metrics, while DPMSolver reports MSE rather than perceptual metrics such as FID.
> Following this line of work, our original submission did not include experiments on text-to-image models. In response to this comment, we have conducted additional experiments on a Stable Diffusion. As is common in this setting, we will provide visualization comparisons.
>
> * **Comparison with iPNDM on LSUN bedroom** We have included comparisons with iPNDM in the revised version. As reported, iPNDM achieves slightly better performance than our sampler on LSUN Bedroom in Table 4. The results are 50.43, 25.88, 13.88, 5.07, 2.79 at NFE = 4,5,6,8,10.
>
> * **Test on Flow Matching** See response to Reviewer yiQS on "Applications on low-matching approaches" for details.
>
> **Discussion about the classical numerical methods**
>
> We agree that the look-ahead point has been used in many existing methods. The main difference is that these methods typically use a future predictor to increase the solver order. For example, in UniPC, the corrector order is always higher than the predictor order. By contrast, our sampler does not use the predictor for order enhancement and remains a first-order method. We will revise the paper to highlight this comparison and to discuss the connections and differences more clearly.
>
> In addition, we would like to remark that our sampler does not require extra NFE for the one-step look-ahead prediction, since we reuse the mu predictor calculated in the previous iteration to obtain eps prediction used for computing $\hat{x}_{t_i}$ as explained in the beginning.
>
> **Comparison with higher-order methods**
>
> While our update can be written in a similar linear form, its coefficients are fundamentally different from those in higher-order solvers such as DPMSolver-2. In our method, the update uses the first-order forward-value discretization coefficients, i.e., the same DDIM-type coefficients but with the model evaluated at a lookahead estimate of the next step, rather than at the current step. By contrast, DPMSolver-2 chooses its coefficients to achieve second-order accuracy via a higher-order approximation of the diffusion ODE integral. Although the algebraic form looks similar, the design principle is different: DPMSolver-2 improves accuracy through higher-order quadrature, whereas our method keeps a first-order coefficient structure and improves accuracy through evaluation placement. We will clarify this distinction and add a more direct comparison in the revision.

---

> > ### Author Rebuttal · Reviewer_JagV · 2026-04-03
> >
> > I thank the authors for their rebuttal. My main concern about the number of model evaluations, as well as the ideological questions about method's philosophy, are resolved.
> >
> > I have one question unresolved: could the authors demonstrate their results on Stable Diffusion in terms of metrics and visual performance (e.g. via anonymous links)? I have not found the results in the answers despite the experiment being mentioned.

---

> > > ### Author Response · Authors · 2026-04-05
> > >
> > > Thank you again for your time and for acknowledging our rebuttal. The remaining concern is regarding the experiments on Stable Diffusion.
> > >
> > >
> > > We have completed the experiments for Stable Diffusion, and the visual results are available at https://anonymous.4open.science/r/F-DPMSolver_rebuttal-01A1/.
> > >
> > > For qualitative comparisons, the results are shown in Figures 1--2. In Figure 1, for the prompt “a desk and chair in an office cubicle,” our forward-value sampler successfully generates a chair, whereas the backward-value samplers fail to do so. In Figure 2, for the prompt “Four tennis players with rackets on a court,” our forward-value sampler generates four players, while the backward-value samplers produce only three.
> > >
> > >
> > > Regarding quantitative comparisons, as noted in our rebuttal, we choose not to report them for two main reasons. First, the FID changes only marginally as NFE increases, a phenomenon also observed in prior work. For example, the authors in [1] note
> > > “We find that all the solvers can achieve an FID around 15.0–16.0 even with only 10 steps, which is very close to the FID computed by the converged samples reported on the official Stable Diffusion page.”
> > > This suggests that the FID is not sufficiently sensitive to reflect quality differences in this regime. Since there is no better perceptual metric, prior works often report the convergence rate of the MSE in latent space relative to DDIM at NFE = 999.
> > > However, our main finding is that the convergence rate is not the only factor governing generation quality. In particular, we show in Theorem 4 that the convergence rate of F-DDIM is the same as that of DDIM. Therefore, MSE is also not an appropriate metric for evaluating our method. We will add these discussions into our paper.
> > >
> > > We hope these clarifications address your concerns. Please feel free to let us know if you have any further questions or comments.
> > >
> > >
> > > [1] Lu, C., Zhou, Y., Bao, F., Chen, J., Li, C., and Zhu, J. Dpmsolver++: Fast solver for guided sampling of diffusion probabilistic models. arXiv: 2211.01095, 2022.

---

### Official Review · Reviewer_hyjX · 2026-03-12

**Soundness:** 3
**Presentation:** 3
**Significance:** 2
**Originality:** 3
**Overall Recommendation:** 4
**Confidence:** 4

**Summary:**

This paper presents training-free diffusion sampling in a few-step generative process. It points out that, in addition to using higher-order approximations, the evaluation position itself matters. An important insight is that, during the few steps (low NFE), using the forward evaluation is more empirically accurate, with a leading discretization error that has the opposite sign to DDIM. The paper provides theoretical results and experiments to support the proposed training-free acceleration framework.

**Compliance With Llm Reviewing Policy:**

Affirmed.

**Final Justification:**

While the rebuttal clarified some experimental aspects, my main concern about the precise motivation and practical implications of forward evaluation was partially resolved. However, the authors also clarified that the theoretical foundation remains limited at this stage.

All in all, I appreciate the exploration of first-order samplers as an alternative to increasingly complex higher-order methods. Considering the overall contribution and potential value of this direction, I have slightly increased my score to a positive value.

**Key Questions For Authors:**

I list below what I believe are the questions, but I would be happy to be corrected if I misunderstood any part of the work.

1. based on (6), the $t_{0}>t_{1}>...>t_{M}=0$. Thus, the $\sigma_{t_{0}}$ should be the largest $\sigma$, which means $\sigma_{t_{0}}\geq\sigma_{t_{i}}$; then line 1292 in the proof of Theorem 4 is not right.

2. What is the goal of Theorem 3? The theorem shows that the leading discretization errors of forward and backward discretization are opposite. However, it is not clear to me what the practical implications of this are, or why it is implied that using forward evaluation is better. Moreover, does this theorem give us a direct way to suggest that using $\frac{x_{t_{M}}^{\text{bck}}+x_{t_{M}}^{\text{for}}}{2}$ is naively higher-order approximation which means it should be more precise and will this leads a better quality?

3. As one main objective of this paper is a training-free acceleration method, do you have any run-time comparison, like speed up, or a specific time? NFE comparison is reasonable, but I would still like to understand whether the proposed solver is actually faster in practice. Also, the paper is restricted to NFE <= 10. What will happen to both quality and runtimes as the number of steps increases? Furthermore, it is important to clarify the paper's position: if we want a few-step model, we can use either the meanflow-style model or the consistency model. If we want to pursue the quality, we can simply add the sampling steps. But for more steps, the steps are close and forward evaluation may not be that important. All in all, my current impression is that the benefit of forward evaluation may be most pronounced in the low-NFE regime, whereas at higher NFE its role may diminish. I would appreciate a clearer discussion of the intended use case and the paper's practical positioning.

**Limitations:**

Yes. However, as I mentioned in my previous question, I would like to better understand the paper's positioning.

**Strengths And Weaknesses:**

## strength
1. The paper challenges a common belief that a higher order of approximation leads to better results.
2. The paper provides theories of order support, and most of the order results look reasonable to me.
3. The main idea is clear, and I find it easy to follow. The idea of considering evaluation placement, rather than only solver order, is conceptually interesting.

## weakness
1. I understand the technical construction, but the position of this paper and the motivation for using forward evaluation are not sufficiently clear for me to see Questions 2-3. **While this is my main concern, if you can address this concern, I would like to increase my score.**
2. Some derivation steps are not fully clear to me, see question 1.

3. The latent diffusion is the main flow nowadays, while the results in 4 for the latent diffusion model are not that clear an advantage, which weakens the practical impact.

---

> ### Author Rebuttal · Authors · 2026-03-31
>
> **Position of this work**
>
> Our goal is not to argue that first-order samplers universally dominate higher-order methods. Rather, the main message of the paper is that solver order is not the only important factor in low-NFE diffusion sampling: the location at which the DPM is evaluated can also have a substantial effect on sample quality. In this sense, our work is complementary to the literature on higher-order solvers. It identifies evaluation placement as an additional and largely independent design axis for training-free acceleration.
>
> In addition, we have conducted experiments to validate the efficiency of the proposed sampler. Experimental results demonstrate that in most cases, our first-order sampler improves the performance over existing first-order methods and is comparable to several higher-order samplers. Hence, we believe that the current experiments are sufficient to support our main message, and we are also happy to add more according to the reviewer's suggestion in the revision.
>
> **Goal of Theorem 3**
>
> Theorem 3 serves two purposes. First, it establishes that the proposed forward-evaluation scheme remains a first-order method. Second, it gives a more refined comparison with DDIM by showing that the leading discretization terms of the forward and backward evaluations are of the same order and norm scale, but point in opposite directions. However, empirical results show that our sampler significantly outperforms DDIM. This suggests that even when first-order error is present, the higher-order terms may still play an important role in sample quality.
>
> **Sum of backward and forward outputs**
>
> Following your suggestion, we have included additional experiments using the averaged estimator $(x_ {t_ M}^{\mathsf{bck}}+x_{t_ M} ^{\mathsf{for}})/2$, which becomes a higher-order sampler.
> For example, on ImageNet64 with EDM2-S, the resulting FIDs are $27.80, 16.37, 11.31, 6.43, 4.61$ for NFE=$4,5,6,8,10$, respectively. This combination improves over DDIM and achieves performance comparable to higher-order samplers such as DPMSolver-2, but is worse than our first-order forward-value F-DDIM (see Table 3). This once again supports the main messages of our paper: a higher-order solver does not necessarily lead to higher sampling accuracy, and higher-order error may play an important role even when the first-order error is present. We will add these results to the revised paper
>
> **Results with higher NFE**
>
> Our original experiments follow closely related works (Lu et al., 2022a; Zheng et al., 2023) and focus on the low-NFE regime, where discretization effects are most pronounced. In response to this comment, we have added results at higher NFEs and will include them in the revised version. In particular, we consider a large model, as it typically requires more sampling steps. We report FIDs on ImageNet512 using the EDM2-XXL model for NFE $=10,12,15,20$, as shown in the table below. As NFE increases, our method remains comparable to higher-order samplers such as DPMSolver and UniPC across this range, further supporting our claims.
>
> |NFE|DDIM|DPMSolver-2|DPMSolver-3|UniPC-3|F-DDIM|
> |-|-|-|-|-|-|
> |10|21.48|6.59|3.60|3.30|**3.06**|
> |12|15.02|4.44|2.80|2.58|**2.48**|
> |15|9.76|3.19|2.39|2.26|**2.23**|
> |20|5.98|2.52|2.16|**2.11**|**2.11**|
>
> **Results on latent diffusion models**
>
> We appreciate this comment. Table 2 (ImageNet512) reports results for conditional latent diffusion models, and Table 4 (FFHQ and LSUN) reports results for unconditional latent diffusion models. On ImageNet512, our sampler outperforms all reference samplers across all NFEs from 4 to 10 for both the EDM2-XS model and EDM2-XXL model. On FFHQ, our sampler achieves the **best** performance except at NFE = 4. On LSUN bedroom, our sampler is occasionally slightly worse than some baselines, but the gap is small and the overall performance remains comparable. This is consistent with our claim that F-DDIM obtains comparable performance to high-order algorithms.
>
> **Running time comparison**
>
> Thank you for this suggestion. We have added runtime comparisons in the revision. To provide a clear comparison across methods, we normalize the runtime by taking DDIM with NFE = $10$ as a reference (set to $10$), and report other methods using the normalized metric
> $$\frac{\text{runtime}}{\text{runtime of DDIM-10}}\times 10.$$
> The above results are reported in the revised paper.
> |NFE|4|5|6|8|10|
> |-|-|-|-|-|-|
> |DDIM|4 x 1.18|5 x 1.12|6 x 1.32|8 x 1.04|10x 1.00|
> |DPMSolver-2|4x 1.17|5 x 1.10|6x 1.07|8x 1.02|10x 1.00|
> |DPMSolver-3|4 x 1.59|5 x 1.47|6 x 1.16|8 x 1.09|10x 1.05|
> |UniPC-3|4 x 1.18|5 x 1.16|6 x1.38|8 x 1.08|10 x 1.05|
> |F-DDIM (ours)|4 x 1.15|5 x 1.08|6 x 1.06|8 x 1.03|10 x 0.99|
>
> **Typo in line 1292**
>
> Thank you for catching this typo. We have corrected it in the revision. This correction does not affect the result in (48), i.e., $\\|x_ {t_ i} - x_ {t_i} ^{\mathsf{for}}\\|=o(1/M)$ or any subsequent derivations and conclusions.

---

> > ### Author Rebuttal · Reviewer_hyjX · 2026-04-03
> >
> > While I still do not fully understand the precise motivation for using forward evaluation beyond the empirical results (and would welcome further clarification), I appreciate the paper’s attempt to challenge the common emphasis on higher-order solvers. The exploration of first-order samplers and evaluation placement is an interesting and worthwhile direction.
> >
> > Given this, I am inclined to support the paper and will increase my score to a positive one.

---

> > > ### Author Response · Authors · 2026-04-04
> > >
> > > Thank you again for your constructive feedback and for helping improve our manuscript. We truly appreciate your support.
> > >
> > > For the motivation, it is primarily discussed in Section 3.1 and Theorem 4. In Section 3.1, we show that with a perfect look-ahead estimator, strong performance can be achieved even with a small number of iterations. In Theorem 4, we further show that even a first-order estimator such as DDIM can provide a sufficiently accurate look-ahead estimate. Motivated by these observations, we propose the forward value sampler. We agree that the method still lacks a more comprehensive theoretical foundation, which is an important direction for future work. We hope this helps clarify your question.
> > >
> > > We would be glad to address any additional questions or comments you may have, and we will ensure that your suggestions are incorporated into the revised manuscript.

---

### Official Review · Reviewer_MiH1 · 2026-03-13

**Soundness:** 2
**Presentation:** 3
**Significance:** 2
**Originality:** 2
**Overall Recommendation:** 4
**Confidence:** 3

**Summary:**

This paper challenges the common belief that higher-order ODE solvers are necessary for fast diffusion sampling, and proposes Forward DPMSolver, which evaluates at the predicted future denoising state rather than the current state, corresponding to a forward-value discretization of the diffusion ODE. The paper provides theoretical analysis showing that the proposed algorithm achieves the same first-order convergence rate under certain assumptions. In the experimental perspective, the method is evaluated on image generation tasks and compared against higher-order samplers and DDIM. Results show that this method achieves improved FID scores in low-NFE regimes against other baselines.

**Compliance With Llm Reviewing Policy:**

Affirmed.

**Final Justification:**

I weighed more on the strengths and weaknesses on soundness, as that's why and how they developed the method they proposed. And I weighed in the order of significance, originality and clarity.
The rebuttal has addressed my main concerns, so I have increased my final score.

**Key Questions For Authors:**

Given the weaknesses discussed above:
1. Can you clarify whether the assumption that the predictor error satisfying $o(1/M)$ instead of $O(1/M)$ is expected to hold in practice, or whether weaker assumption would still guarantee the same convergence result?
2. For the off-trajectory states concern: how sensitive is the method to predictor quality? In particular, could inaccurate predictions lead to worse performance compared to DDIM?
3. can you provide additional evaluation metrics for the image generation experiments to offer a more comprehensive evaluation of sampling quality?

**Limitations:**

Limitation is not exactly discussed, but this paper itself notes that further theoretical understanding of the forward-value discretization is needed to explain the observed practical gains beyond standard first-order schemes.

**Strengths And Weaknesses:**

**Soundness**:

strengths: This paper provides a theoretical analysis to connect forward-value discretization with diffusion ODE sampling in DDIM. The algorithm is training-free and simple to implement.

weaknesses:

Theorem 4 assumes the predictor satisfies $o(1/M)$, which is stronger than the $O(1/M)$ accuracy conventionally achieved by first-order solvers such as DDIM. It is unclear whether assumption holds in practice, so the theoretical guarantee may not apply to the implemented algorithm.

The algorithm also heavily relies on the quality of the look-ahead predictor. If the one-step denoised estimate is inaccurate, the predicted next state may deviate from the true trajectory and lie off the data manifold. Moreover, such errors may accumulate during the sampling process and lead to further deviation of the sampling trajectory.

**Presentation**:

strengths: This paper is easy to follow and understand. The motivation behind the forward-value discretization is clearly explained. The algorithm is straightforward to implement.

weaknesses: The experimental evaluation mainly reports FID scores. Including more metrics could provide more insight into why the method works. For example, measuring divergence between intermediate sampler distribution and the true diffusion marginals at corresponding noise levels could help examine how the one-step prediction would affect the accuracy of the constructed noisy samples.

**Significance**:

strengths: This paper challenges the belief that higher-order solvers are necessary for fast diffusion sampling. The proposed method also improves sampling quality in low-NFE regimes.

weaknesses: The experiments are limited to image generation tasks, which may restrict the demonstrated impact of the method across other real world application such as inverse problem or molecule generation.

**Originality**:

strengths: This paper proposes an interesting perspective that the forward-value discretization strategy achieves significant empirical improvement over standard first-order samplers, and also competes with higher-order samplers.

weaknesses: The idea of using a denoiser to estimate a future state resembles implicit Euler-type approximation schemes. In particular, the proposed method can be interpreted as a predictor-corrector strategy: the next denoising state is first predicted using a one-step estimate from the denoiser, and then an Euler-type update is applied as the corrector step. Therefore, the methodology novelty may be incremental from a numerical analysis perspective. It would therefore be helpful to include comparisons with existing predictor-corrector samplers as baselines, which is also discussed in the related work.

---

> ### Author Rebuttal · Authors · 2026-03-31
>
> **Clarify the misunderstanding of the condition in Theorem 4**
>
> We would like to clarify that the condition in Theorem 4 concerns the **one-step** lookahead error from $t_ {t-1}$ to $t_i$, rather than the **cumulative** error over the full trajectory. Specifically, the assumption requires that the one-step prediction error $\|\hat{x}_ {t_i} - x_ {t_i}^{\star}(x_ {t_ {i-1}}, t_ {i-1}) \|$ is $o(1/M)$, where $x_{t_i}^{\star}(x_{t_{i-1}}, t_{i-1})$ denotes the exact ODE solution at time $t_i$ **initialized from** $(x_ {t_ {i-1}}, t_ {i-1})$ **instead of** $(x_ {t_ {0}}, t_ {0})$.
>
> Moreover, F-DDIM satisfies this condition naturally. Indeed, by the decomposition
> $$\\| \hat{x}_ {t_ i} - x_ {t_ i}^{\star}\\|_ 2 \le \\|\hat{x}_ {t_ i} - x_ {t_ i} ^{\mathsf{ddim}}\\|_ 2+\\|x_ {t_ i} ^{\mathsf{ddim}} - x_ {t_ i} ^{\star}\\|_ 2,$$
> where $x_ {t_i} ^{\mathsf{ddim}}$ denotes the one-step DDIM update from $(x_{t_{i-1}}, t_{i-1})$ using $\varepsilon_\theta(x_{t_{i-1}}, t_{i-1})$, the second term is $O(1/M^2)$ by Theorem 1. For the first term, we have
> $$\\|\hat{x}_ {t_ i} - x_ {t_ i} ^{\mathsf{ddim}}\\|_ 2=O(\delta_ {t_ i}\\|\hat{x}_ {t_ {i-1}} - x_ {t_ {i-1}}\\|_ 2)=O(1/M ^2).$$
> In addition, we can observe that the performance of our sampler is not overly sensitive to the one-step prediction error characterized by the factor $\delta_{t_i}$
>
> The similar argument applies to F-DPMSolver2. We will revise the discussion of the theorem to clarify the distinction between one-step and global errors.
>
> **Sensitivity to predictor error**
>
> According to previous discussion, our requirement on the one-step look-ahead error is mild, only $o(1/M)$ is sufficient for single step. Moreover, even for the first-order sampler such as DDIM, the one-step estimate error is $O(1/M^2)$.
>
> **Measure divergence of intermediate steps**
>
> Almost all related works in this direction report only FID as the performance metric, including Zhao et al. (2023), Lu et al. (2022a), and Zheng et al. (2023).
> For the divergence at intermediate steps, we are not aware of any established evaluation method, especially in the high-dimensional setting. To address your comment, we have provided example samples at intermediate steps for illustration.
> In addition, we have also reported FID scores at intermediate steps to provide a quantitative perspective.
> For example, on ImageNet64 dataset with the EDM2-S model and NFE$=6$, the FIDs at steps $t=2,4,6$ are $0.035$, $5.79$, and $6.67$, respectively.
> As expected, the FID increases with $t$, due to the accumulation of approximation error along the sampling trajectory.
>
> In addition, if there is a method for measuring the divergence between the intermediate sampler distribution and the true diffusion marginals, we would greatly appreciate a reference.
>
> **Extension to inverse problem or molecule generation**
>
> Since diffusion models have been successfully applied to these tasks, we expect our sampler to be effective in these settings as well. However, as this direction lies beyond the scope of the current work, we leave it for future investigation.
>
> **Comparison to predictor-corrector samplers**
>
> We agree that our method can be interpreted from a predictor-corrector perspective. In fact, we have already included predictor-corrector samplers, such as UniPC, as baselines in our experiments. However, we would like to emphasize an important distinction in design and objective.
> Classical predictor-corrector samplers are primarily constructed to increase the numerical order of the solver; for example, in methods such as UniPC, the corrector is explicitly designed to achieve a higher-order approximation than the predictor. In contrast, our method does not aim to improve the order of the solver and remains first-order overall. Instead, it explores a different design axis --- namely, the choice of evaluation location of the denoising model --- and analyzes how this affects discretization error.
> We will further clarify this connection and distinction in the revised version.

---

> > ### Author Rebuttal · Reviewer_MiH1 · 2026-04-03
> >
> > I thank for the response. I will increase my score.

---

> > > ### Author Response · Authors · 2026-04-04
> > >
> > > Thank you again for your valuable feedback and for raising your score! We will ensure that your comments are incorporated into the revision.

---

### Official Review · Reviewer_yiQS · 2026-03-19

**Soundness:** 2
**Presentation:** 2
**Significance:** 3
**Originality:** 3
**Overall Recommendation:** 4
**Confidence:** 2

**Summary:**

- This paper introduces a novel training-free first order sampler called F-DPMSolver, which challenges the common belief – that the first-order diffusion probabilistic sampling is slower than higher order ODE solvers. The authors approximate the forward diffusion process via the next-time-step predictor for faster sampling, and also provide a theoretical analysis which guarantees the ideal forward trajectory without hurting first-order convergence. Also, the proposed method is tested on various benchmarks in the low-NFE regime, demonstrating that higher-order methods are not the only viable approach for accelerating diffusion model sampling.

**Compliance With Llm Reviewing Policy:**

Affirmed.

**Final Justification:**

The authors sufficiently addressed my concerns, and I thus maintain my score.

**Key Questions For Authors:**

Please refer to the above weaknesses.

**Limitations:**

Yes.

**Strengths And Weaknesses:**

Disclaimer: This work is somewhat outside my primary area of expertise.

Strengths
- The paper tackles a timely and practically-relevant problem supported by a fair amount of experiments.
- The paper proposes a novel perspective viewpoint of efficient diffusion modeling along with the theoretical analysis.
- Overall, the paper is clearly written and easy to follow.

Weaknesses
- One of my main questions is whether the experimental evaluation regarding NFE is fairly executed between the original method (F-DDIM) and the hybrid method (F-DPMSolver2). For example, there are the main results comparing the trade-off between NFE and FID in Tables 1, 2 for F-DDIM and Tables 5, 6 for F-DPMSolver2. I wonder whether those two methods are evaluated under the same NFE budget. Are we caching all prior forward values?
- As mentioned in the literature review section, I would like to ask why the authors did not compare the training-free paralleling sampling methods such as (Gupta et al., 2024).
- From a practical standpoint, although the paper focuses on “training-free” acceleration, it would be useful to examine whether the proposed method can be extended to other frameworks such as consistency models or progressively distilled models.
- Along similar lines, some discussion on flow-matching approaches (e.g., FLUX) would strengthen the paper, particularly regarding whether the proposed method is applicable or effective in that setting.

---

> ### Author Rebuttal · Authors · 2026-03-31
>
> We thank the reviewer for the thoughtful and constructive comments. Below, we provide a point-by-point response and clarify how these issues will be addressed in the revision.
>
> **NFE budget of F-DPMSolver2**
>
> We would like to clarify that the NFE budgets of F-DDIM and F-DPMSolver2 are the same. For both F-DDIM and F-DPMSolver2, we perform only **one DPM evaluation** per iteration, i.e., $\mu _ {\theta}(\hat{x} _ {t_ i}, t_ i)$. The difference lies only in how the one-step lookahead estimate $\hat{x}_ {t_i}$ is formed from previously computed quantities.
> Specifically, F-DDIM constructs $\hat{x}_ {t_i}$ by reusing $x_ {t_{i-1}}$ and the previous DPM evaluation $\mu_ {\theta}(\hat{x}_ {t_ {i-1}}, t_ {i-1})$, while F-DPMSolver2 additionally reuses $\mu_ {\theta}(\hat{x}_ {t_{i-2}}, t_ {i-2})$. Thus, F-DPMSolver2 requires slightly more caching, but not additional model evaluations. We will revise the paper to state this more explicitly and avoid any ambiguity.
>
> **Comparison with (Gupta et al., 2024)**
>
> We did not include Gupta et al. (2024) as an experimental baseline because that work is primarily theoretical and does not provide a practical implementation or empirical results for direct comparison. In addition, its parallel strategy involves multiple DPM evaluations within an iteration, so the computational budget is not directly comparable to our single-evaluation-per-step setting.
>
> In the revision, we will add a discussion to better position our work relative to this line of theoretical acceleration results:
> "We note that several theoretical works (Gupta et al., 2024) study acceleration through randomized or intermediate evaluation strategies. However, they focus on improving the theoretical convergence rate, whereas our focus is on practical sampling performance.''
>
> **Extension to consistency models or progressively distilled models**
>
> This is an interesting direction. In the training of consistency models or progressively distilled models, it is required to generate data pairs along the diffusion ODE. Our proposed method could potentially be used in these fields for generating such training pairs more efficiently, which will be left as our future work.
>
> **Applications on flow-matching approaches**
>
> We agree that this is an important point. Our method is motivated by the placement of field evaluations along the trajectory, and this idea is not restricted to the score-based parameterization. From this perspective, our forward-evaluation strategy can also be applied in the flow-matching setting by evaluating the velocity field at a forward-looking point along the trajectory. We expect that this will yield similar acceleration benefits as in the score-based setting, and we will explore this important direction in future work.

---

> > ### Author Rebuttal · Reviewer_yiQS · 2026-04-03
> >
> > The authors sufficiently addressed my concerns, and I thus maintain my score.

---

> > > ### Author Response · Authors · 2026-04-04
> > >
> > > Thank you again for your time and effort in handling our paper. We would be happy to address any further questions or comments you may have, and make sure to incorporate the suggestions in the revision.

---

### Decision · Program_Chairs · 2026-04-30

**Decision:**

Accept (regular)

**Comment:**

This paper challenges a notion that first-order samplers require more NFEs than higher-order ones, to achieve the same level of sample quality. After the rebuttal, the reviewers unanimously agreed this paper should be accepted.